# Generalization Guarantee of SGD for Pairwise Learning

**Yunwen Lei**[1]    **Mingrui Liu**[2]    **Yiming Ying**[3]

[1]School of Computer Science, University of Birmingham, Birmingham B15 2TT, United Kingdom
[2]Department of Computer Science, George Mason University, Fairfax, VA 22030, USA
[3]Department of Mathematics and Statistics, State University of New York at Albany, USA
y.lei@bham.ac.uk    mingruil@gmu.edu    yying@albany.edu

## Abstract

Recently, there is a growing interest in studying pairwise learning since it includes many important machine learning tasks as specific examples, e.g., metric learning, AUC maximization and ranking. While stochastic gradient descent (SGD) is an efficient method, there is a lacking study on its generalization behavior for pairwise learning. In this paper, we present a systematic study on the generalization analysis of SGD for pairwise learning to understand the balance between generalization and optimization. We develop a novel high-probability generalization bound for uniformly-stable algorithms to incorporate the variance information for better generalization, based on which we establish the first *nonsmooth* learning algorithm to achieve almost optimal *high-probability* and dimension-independent excess risk bounds with $O(n)$ gradient computations. We consider both convex and nonconvex pairwise learning problems. Our stability analysis for convex problems shows how the interpolation can help generalization. We establish a uniform convergence of gradients, and apply it to derive the first excess risk bounds on population gradients for nonconvex pairwise learning. Finally, we extend our stability analysis to pairwise learning with gradient-dominated problems.

## 1   Introduction

Many machine learning problems can be formulated as learning with pairwise loss functions, where the performance of the associated models needs to be quantified on a pair of training examples. Representative problems include AUC maximization [14, 25, 42, 63, 66], metric learning [8, 31], ranking [1, 13] and learning with minimum error entropy loss functions [29]. For example, in supervised metric learning we wish to find a distance function between pairs of examples so that examples within the same class are relatively close while examples from different classes are far apart from each other. In ranking, we aim to find a function to predict the ordering of examples. This motivates the recent growing interest in a unifying study of these problems, under the framework of pairwise learning [29, 32, 40, 58].

Stochastic gradient descent (SGD) is a workhorse for machine learning due to its cheap computation complexity, simplicity and efficiency [7, 20, 49, 53, 62, 65]. SGD iteratively updates the model based on stochastic gradients computed on one or several randomly selected training examples, which can achieve sample-size independent iteration complexity for a prescribed optimization accuracy. This is especially attractive for pairwise learning as the objective function involves $O(n^2)$ terms for problems with $n$ training examples. An important problem on SGD is to understand its generalization performance, i.e., how the models trained by SGD would behave on testing examples. While there are some interesting work on the generalization analysis of SGD for pointwise learning [9, 10, 28, 36, 39], there is much less work on SGD for pairwise learning. A notable difference between pairwise learning

35th Conference on Neural Information Processing Systems (NeurIPS 2021).

and pointwise learning is that the objective function in pairwise learning involves $O(n^2)$ *dependent* terms, which introduces a difficulty in handling this dependency. For example, one needs to decouple this dependency to apply concentration inequalities established for independent data. Furthermore, for algorithmic stability analysis, a perturbation of the training dataset by a single example can change $O(n)$ terms in the objective function, which is more challenging than stability analysis of pointwise learning. To our best knowledge, the only work on generalization analysis of SGD for pairwise learning are [40, 55, 59]. However, their analysis requires restrictive assumptions on convexity, smoothness and Lipschitz continuity of loss functions. Furthermore, they fail to incorporate the interpolation (low noise) assumption into their generalization guarantee.

In this paper, we initialize a systematic generalization analysis of SGD for pairwise learning under general assumptions. Our contributions are listed as follows.

1. We develop a novel high-probability generalization bound for uniformly-stable algorithms, which incorporates the variance information to improve the learning performance. We apply this result to develop the first dimension-independent high-probability bound $O(1/\sqrt{n})$ (up to a log factor) for an algorithm with $O(n)$ gradient computations to solve nonsmooth learning problems.

2. We study the stability and generalization guarantee of SGD for pairwise learning with convex loss functions, covering both smooth and nonsmooth problems. Our analysis suggests an early-stopping strategy for getting excess population risk bounds of the order $O(1/\sqrt{n})$ and $O(1/(n\sigma))$ for convex and $\sigma$-strongly convex problems, respectively. Under an interpolation or a low noise assumption, we improve our excess risk bounds to $O(1/n)$ by exploiting the smoothness assumption.

3. We provide the first generalization analysis of SGD for pairwise learning with nonconvex loss functions. We establish a uniform convergence of empirical gradients to population gradients by showing its connection to Rademacher chaos complexities. We then apply this uniform convergence to develop high-probability generalization guarantees for general nonconvex pairwise learning. Under a gradient dominance assumption, our stability analysis gives dimension-independent bounds.

The paper is organized as follows. We survey the related work in Section 2 and give the problem formulation in Section 3. We study convex and nonconvex pairwise learning in Section 4 and Section 5, respectively. Conclusion is given in Section 6. In the appendix, we present all the proofs, specific examples of pairwise learning and preliminary experimental results.

## 2 Related Work

We first review the related work on algorithmic stability. Algorithmic stability is an important concept in statistical learning theory (SLT) with close connection to learnability [52, 54]. The modern framework of stability analysis was established in a seminal paper [5], where the important uniform stability was introduced. This stability measure was extended to study randomized algorithms in [18] and motivates several concepts including argument stability [39, 43], Bayes stability [41] and on-average stability [39, 54]. Algorithmic stability has shown its remarkable effectiveness in deriving dimension-independent generalization bounds for various domains including stochastic optimization [10, 28, 36, 47, 51], structured prediction [44], transfer learning [37] and differential privacy [3, 48]. Recent progress shows a tradeoff between optimization and stability [11, 55], and its applications to yield almost optimal high-probability generalization bounds [6, 22, 34].

We now review the related work on pairwise learning. There are two popular approaches to studying the generalization performance of pairwise learning: the uniform convergence approach and the algorithmic stability approach. The idea of uniform convergence is to control the uniform deviation between training errors and testing errors over a hypothesis space. The complexity measures of function spaces play an important role in this approach, including VC dimension, covering numbers [17, 58, 60] and Rademacher complexities [8, 32]. Furthermore, one needs to use concentration inequalities to handle the associated U-statistics and U-processes [13, 16, 50]. Algorithmic stability of ranking [1] and metric learning [31, 57] was studied for strongly convex objectives [30]. High-probability bounds of the order $O(\epsilon \log n + 1/\sqrt{n})$ were recently developed for $\epsilon$-uniformly stable pairwise learning algorithms [40]. A nice property of stability analysis is its ability to yield dimension-independent bounds, while a square-root dependency on the dimension is inevitable for uniform convergence analysis when considering general problems [21]. However, algorithmic stability generally requires a convexity assumption which is not required for the uniform convergence

approach. Other than the above two approaches, several researchers have studied the generalization behavior of pairwise learning using the algorithmic robustness [4, 12] and integral operators [19, 27].

Before we move on, we add more discussions with a very related work on generalization analysis of pairwise learning [13, 50]. The work [50] considers a very general problem setting for SGD with $K$-sample U-statistic of degrees $(d_1, \ldots, d_K)$, which includes our algorithm as a special case with $K = 1$ and $d_1 = 2$. It shows the advantage of reducing variances using gradient estimates through incomplete U-statistics over that through complete U-statistics based on subsamples. We sketch the difference as follows. First, the generalization analysis in Papa et al. [50] requires smoothness and strong convexity assumptions. As a comparison, we also consider nonsmooth problems (Section 4.2) and nonconvex problems (Section 5). Second, the work [50] studies generalization via the uniform convergence approach and requires a complexity assumption. As a comparison, we also study generalization via a fundamentally different algorithmic stability approach. The classical work [13] focuses on the exact solution of empirical risk minimizer for pairwise learning from the perspective of uniform convergence. As a comparison, we study the excess risk of SGD via both an algorithmic stability approach and an uniform convergence approach, for which we also consider the tradeoff between optimization and generalization.

## 3 Problem Formulation

### 3.1 Pairwise Learning and Stochastic Gradient Descent

Let $\rho$ be a probability measure defined on $\mathcal{Z} = \mathcal{X} \times \mathcal{Y}$ with an input space $\mathcal{X}$ and an output space $\mathcal{Y}$. Let $S = \{z_1, \ldots, z_n\}$ be drawn independently according to $\rho$, from which we aim to learn a prediction function $h : \mathcal{X} \mapsto \mathbb{R}$ or $h : \mathcal{X} \times \mathcal{X} \mapsto \mathbb{R}$. We consider parametric models where the prediction function $h_{\mathbf{w}}$ can be indexed by an element $\mathbf{w} \in \mathcal{W}$, where $\mathcal{W}$ is a $d$-dimensional Hilbert space. We consider pairwise learning problems where the performance of a model $h_{\mathbf{w}}$ on an example pair $(z, z')$ can be measured by a nonnegative loss function $f(\mathbf{w}; z, z')$. This is in contrast to standard pointwise learning (e.g., classification and regression) where we can measure the quality of a model via its behavior on an individual point. Two notable examples of pairwise learning include ranking and supervised metric learning. For ranking, we build a function $h_{\mathbf{w}} : \mathcal{X} \mapsto \mathcal{Y}$ to rank instances in a way consistent with the outputs, i.e., $h_{\mathbf{w}}(x) < h_{\mathbf{w}}(x')$ if $y < y'$ for two example pairs $z = (x, y), z' = (x', y')$. Then we can formulate ranking as a pairwise learning problem with $f(\mathbf{w}; z, z') = \psi(\mathrm{sgn}(y - y')(h_{\mathbf{w}}(x) - h_{\mathbf{w}}(x')))$, where sgn is the sign function and $\psi$ can be either the hinge loss $\psi(t) = \max\{1 - t, 0\}$ or the logistic loss $\psi(t) = \log(1 + \exp(-t))$. For supervised metric learning with $\mathcal{Y} = \{-1, +1\}$, we find a distance function under which examples with the same label are similar while examples with different labels are apart from each other. A popular distance function takes the form $h_{\mathbf{w}}(x, x') = \langle \mathbf{w}, (x - x')(x - x')^\top \rangle$, where $\mathbf{w} \in \mathbb{R}^{d \times d}$ is positive definite. We can formulate supervised metric learning as pairwise learning with $f(\mathbf{w}; z, z') = \psi(\tau(y, y')(1 - h_{\mathbf{w}}(x, x')))$, where $\tau(y, y') = 1$ if $y = y'$ and $-1$ otherwise.

The population risk of $\mathbf{w}$ in pairwise learning is $F(\mathbf{w}) = \mathbb{E}_{Z,Z'}[f(\mathbf{w}; Z, Z')]$, where $\mathbb{E}_{Z,Z'}$ denotes the expectation with respect to (w.r.t.) $Z, Z' \sim \rho$. The empirical risk of $\mathbf{w}$ is

$$F_S(\mathbf{w}) = \frac{1}{n(n-1)} \sum_{i,j \in [n]: i \neq j} f(\mathbf{w}; z_i, z_j),$$

where $[n] := \{1, \ldots, n\}$. Let $\mathbf{w}_S^* = \arg\min_{\mathbf{w} \in \mathcal{W}} F_S(\mathbf{w})$ and $\mathbf{w}^* = \arg\min_{\mathbf{w} \in \mathcal{W}} F(\mathbf{w})$. For a randomized algorithm $A$, we use $A(S)$ to denote the output model produced by applying $A$ to the dataset $S$. We are interested in the excess risk $F(A(S)) - F(\mathbf{w}^*)$, which measures the relative behavior of $A(S)$ as compared to the best model. A standard approach to handle $F(A(S)) - F(\mathbf{w}^*)$ is to use the following error decomposition

$$\mathbb{E}_{S,A}\big[F(A(S)) - F(\mathbf{w}^*)\big] = \mathbb{E}_{S,A}\big[F(A(S)) - F_S(A(S))\big] + \mathbb{E}_{S,A}\big[F_S(A(S)) - F_S(\mathbf{w}^*)\big], \quad (3.1)$$

where the first term $F(A(S)) - F_S(A(S))$ is called the generalization error and the second term $F_S(A(S)) - F_S(\mathbf{w}^*)$ is the optimization error. These two errors can be handled by tools in SLT and optimization theory, respectively. We are interested in the specific SGD for pairwise learning.

**Definition 1** (SGD for Pairwise Learning). Let $\mathbf{w}_1 = 0 \in \mathbb{R}^d$ and $\{\eta_t\}_t$ be a stepsize sequence. Let $\nabla f(\mathbf{w}_t; z_{i_t}, z_{j_t})$ denote the gradient of $f$ w.r.t. the first argument. At the $t$-th iteration, we first draw $\{(i_t, j_t)\}$ from the uniform distribution over all pairs $\{(i, j) : i, j \in [n], i \neq j\}$ and then update

$$\mathbf{w}_{t+1} = \mathbf{w}_t - \eta_t \nabla f(\mathbf{w}_t; z_{i_t}, z_{j_t}). \quad (3.2)$$

Note our problem setting is totally different from some online learning setting where the streaming examples are assumed to be drawn from the true probability measure $\rho$ [15]. Indeed, we consider the offline setting where the dataset is given beforehand and during the optimization process we actually randomly draw an example from the empirical measure. This necessitates the consideration of the generalization gap which is not touched in the online learning setting [15]. An advantage of SGD is that its computational complexity to achieve an accuracy is independent of the number of pairs, which is particularly attractive for pairwise learning (gradient descent requires $O(n^2)$ gradient computations per iteration). We describe $\mathrm{SGD}(S, T, f, \{\eta_t\})$ in Algorithm 1 of SGD with dataset $S$, iteration number $T$, loss function $f$ and stepsize $\{\eta_t\}$. Algorithm 1 was also studied in [40], which however requires restrictive assumptions on convexity, smoothness and Lipschitz continuity. We significantly extend their discussions by considering either nonconvex, nonsmooth or non-Lipschitz loss. Moreover, our analysis can clarify the effect of interpolation on generalization.

---

**Algorithm 1:** $\mathrm{SGD}(S, T, f, \{\eta_t\})$

---

**Input:** initial point $\mathbf{w}_1 = 0$, learning rates $\{\eta_t\}_t$, and dataset $S = \{z_1, \ldots, z_n\}$
1 **for** $t = 1, 2, \ldots, T$ **do**
2 $\quad$ draw $(i_t, j_t)$ uniformly over all pairs $\{(i, j) : i, j \in [n], i \neq j\}$
3 $\quad$ update $\mathbf{w}_{t+1}$ according to Eq. (3.2)
4 **end**

---

Below we introduce necessary definitions and assumptions. Let $\|\cdot\|_2$ be the Euclidean norm and $\langle \cdot, \cdot \rangle$ be the associated inner product. Let $b = \sup_{z, z' \in \mathcal{Z}} f(0; z, z')$ and $b' = \sup_{z, z' \in \mathcal{Z}} \|\nabla f(0; z, z')\|_2$. We denote $B \asymp \widetilde{B}$ if there are absolute constants $c_1$ and $c_2$ such that $c_1 B \leq \widetilde{B} \leq c_2 B$. We collect the notations of this paper in Table A.1.

**Definition 2.** Let $g : \mathcal{W} \mapsto \mathbb{R}, L, G > 0, \sigma \geq 0$.

1. We say $g$ is $L$-smooth if $\|\nabla g(\mathbf{w}) - \nabla g(\mathbf{w}')\|_2 \leq L\|\mathbf{w} - \mathbf{w}'\|_2$ for all $\mathbf{w}, \mathbf{w}' \in \mathcal{W}$.

2. We say $g$ is $G$-Lipschitz continuous if $|g(\mathbf{w}) - g(\mathbf{w}')| \leq G\|\mathbf{w} - \mathbf{w}'\|_2$ for all $\mathbf{w}, \mathbf{w}' \in \mathcal{W}$.

3. We say $g$ is $\sigma$-strongly convex w.r.t. $\|\cdot\|_2$ if $g(\mathbf{w}) - \left(g(\mathbf{w}') + \langle \mathbf{w} - \mathbf{w}', \nabla g(\mathbf{w}') \rangle\right) \geq \sigma\|\mathbf{w} - \mathbf{w}'\|_2^2/2$ for all $\mathbf{w}, \mathbf{w}' \in \mathcal{W}$. We say $g$ is convex if $g$ is $\sigma$-strongly convex with $\sigma = 0$.

**Assumption 1** (Convexity). Assume for all $z, z' \in \mathcal{Z}$, the function $\mathbf{w} \mapsto f(\mathbf{w}; z, z')$ is convex.

**Assumption 2** (Boundedness of Gradients). Assume for all $z, z'$ and $\mathbf{w} \in \mathcal{W}$, $\|\nabla f(\mathbf{w}; z, z')\|_2 \leq G$.

**Assumption 3** (Smoothness). Assume for all $z, z'$, $\mathbf{w} \mapsto f(\mathbf{w}; z, z')$ is nonnegative and $L$-smooth.

### 3.2 Algorithmic Stability and Generalization

A fundamental concept in SLT is the algorithmic stability, which measures the sensitivity of an algorithm w.r.t. the perturbation of the training dataset. Various stability measures have been introduced in the literature, including uniform stability [5], hypothesis stability [5, 18], argument stability [43] and on-average stability [54]. We focus on uniform stability and on-average stability here. The following on-average loss stability was introduced in [40], while the on-average argument stability was motivated by a similar concept in pointwise learning [39]. Let $S = \{z_1, \ldots, z_n\}, S' = \{z_1', \ldots, z_n'\}$ be independently drawn from $\rho$. We denote

$$S_i = \{z_1, \ldots, z_{i-1}, z_i', z_{i+1}, \ldots, z_n\}, \quad \forall i \in [n], \tag{3.3}$$

$$S_{i,j} = \{z_1, \ldots, z_{i-1}, z_i', z_{i+1}, \ldots, z_{j-1}, z_j', z_{j+1}, \ldots, z_n\}, \quad \forall i < j \in [n]. \tag{3.4}$$

**Definition 3** (Algorithmic Stability). Let $S = \{z_1, \ldots, z_n\}$ and $S' = \{z_1', \ldots, z_n'\}$ be drawn independently from $\rho$. For any $i, j \in [n]$, denote $S_i$ as (3.3) and $S_{i,j}$ as (3.4).

1. We say a deterministic algorithm $A : \mathcal{Z}^n \mapsto \mathcal{W}$ is $\epsilon$-uniformly stable if for any datasets $S, \widetilde{S} \in \mathcal{Z}^n$ that differ by at most a single example we have $\sup_{z, \tilde{z} \in \mathcal{Z}} \left| f(A(S); z, \tilde{z}) - f(A(\widetilde{S}); z, \tilde{z}) \right| \leq \epsilon$.

2. We say a randomized algorithm $A$ is on-average (loss) $\epsilon$-stable if

$$\frac{1}{n(n-1)} \sum_{i,j \in [n]: i \neq j} \mathbb{E}_{S, S', A} \left[ f(A(S_{i,j}); z_i, z_j) - f(A(S); z_i, z_j) \right] \leq \epsilon.$$

3. We say $A$ is on-average argument $\epsilon$-stable if $\mathbb{E}_{S,\widetilde{S},A}\left[\frac{1}{n}\sum_{i=1}^{n}\|A(S)-A(S_i)\|_2^2\right]\leq\epsilon^2$.

The following theorem establishes the connection between on-average stability and generalization in expectation for pairwise learning. Part (a) was due to [40], while Part (b) was motivated by a similar result in pointwise learning [39]. The proof is given in Appendix B.

**Theorem 1.** *(a) If $A$ is on-average (loss) $\epsilon$-stable, then $\mathbb{E}[F(A(S))-F_S(A(S))]\leq\epsilon$.*

*(b) If $A$ is on-average argument $\epsilon$-stable and Assumption 3 holds, then for any $\gamma>0$ we have*

$$\mathbb{E}[F(A(S))-F_S(A(S))]\leq 2(L+\gamma)\epsilon^2+L\gamma^{-1}\mathbb{E}[F_S(A(S))].$$

**Remark 1.** We can choose $\gamma\asymp\sqrt{\mathbb{E}[F_S(A(S))]}/\epsilon$ in Part (b) and get the bound $\mathbb{E}[F(A(S))-F_S(A(S))]=O(\epsilon^2+\epsilon\sqrt{\mathbb{E}[F_S(A(S))]})$. Therefore, if $\mathbb{E}[F_S(A(S))]$ is small, Part (b) can imply bounds better than $O(\epsilon)$. In particular, if $\mathbb{E}[F_S(A(S))]=O(\epsilon^2)$ the bound in Part (b) becomes $O(\epsilon^2)$, which is much faster than $O(\epsilon)$ in Part (a).

Theorem 2 establishes the connection between uniform stability and generalization with high probability for pairwise learning. The proof is given in Section C, whose novelty is to use decoupling techniques to address the coupling among $O(n^2)$ terms in the objective function of pairwise learning.

**Theorem 2.** *Let $A$ be an $\epsilon$-uniformly stable and deterministic algorithm. Let $B:=\sup_{z,z'}\left|\mathbb{E}_S[f(A(S);z,z')]-f(\mathbf{w}^*;z,z')\right|$ and $\sigma_0^2:=\mathbb{E}_{Z,Z',S}\left[\left(f(A(S);Z,Z')-f(\mathbf{w}^*;Z,Z')\right)^2\right]$. For any $\delta\in(0,1)$, the following inequality holds with probability at least $1-\delta$*

$$F(A(S))-F_S(A(S))-F(\mathbf{w}^*)+F_S(\mathbf{w}^*)\leq 98\sqrt{2}e\epsilon\log n\log(2e/\delta)+\frac{2B\log(2/\delta)}{3\lfloor n/2\rfloor}+\sqrt{\frac{2\sigma_0^2\log(2/\delta)}{\lfloor n/2\rfloor}}.$$

**Remark 2.** Note we only impose a bounded loss assumption on $A(S)$, which can be achieved by truncating the value of the output function. Theorem 2 was motivated by the recent work [34, 40]. For pointwise learning, high-probability generalization bounds of the order $O(\epsilon\log n)$ were developed for $\epsilon$-uniformly stable algorithms under a further Bernstein condition on the variance-expectation relationship [34]. High-probability bounds $O(\epsilon\log n+1/\sqrt{n})$ were also developed for $\epsilon$-uniformly stable algorithms in pairwise learning [40]. We refine these results by developing generalization bounds $O(\epsilon\log n+\sqrt{\sigma_0^2/n})$, where $\sigma_0^2$ is the variance of the excess loss at the output model. If this variance is small, then our bounds can be much better than that in [40]. For example, if $F$ is $\sigma$-strongly convex, then one can show that this variance can be bounded by $O(\mathbb{E}[F(A(S))-F(\mathbf{w}^*)]/\sigma)$, and in this case the term $\sqrt{\sigma_0^2/n}$ in our bound would be $o(n^{-\frac{1}{2}})$ instead of $O(n^{-\frac{1}{2}})$ in [40]. As we will show, Theorem 2 can imply almost optimal excess risk bounds for an algorithm with $O(n)$ gradient computations to solve nonsmooth problems (Theorem 8), for which the existing high-probability analysis can only imply bounds of the order $O(n^{-\frac{1}{4}})$ [40].

## 4 Pairwise Learning with Convex Loss Functions

In this section, we study the generalization performance of SGD for pairwise learning with convex loss functions. We consider convex/strongly-convex and smooth/nonsmooth problems.

### 4.1 Convex and Smooth Problems

We first consider stability and risk bounds for convex and smooth pairwise learning problems. The proofs of results in this subsection can be found in Section E. Theorem 3 gives the bounds for on-average argument stability of SGD. Note we do not require the loss functions to be Lipschitz continuous. A nice property is that the upper bound involves the empirical risk of $\mathbf{w}_j$. Since we are minimizing the empirical risk by SGD, it is reasonable to assume that $F_S(\mathbf{w}_j)$ would become smaller and smaller along the learning process. It should be mentioned that a similar result was derived for pointwise learning [39]. A key difference in the stability analysis for pairwise learning is that a change of $z_i$ would influence $2(n-1)$ pairs $(z_j,z_i),(z_i,z_j)$ for $j\neq i$. We need to use the U-structure of the empirical risk to prove this result.

**Theorem 3** (Stability bound). *Let Assumptions 1, 3 hold. Let A be SGD (Algorithm 1) with $\eta_j \leq 2/L$. Then A with $t$ iterations is on-average argument $\epsilon$-stable with $\epsilon^2 \leq \frac{16L(1+2t/n)e}{n} \sum_{j=1}^{t} \eta_j^2 \mathbb{E}[F_S(\mathbf{w}_j)]$.*

Based on Theorem 3, we get error bounds for pairwise learning with convex and smooth functions.

**Theorem 4** (Excess risk bound). *Let Assumptions 1, 3 hold. Let $\{\mathbf{w}_t\}$ be the sequence produced by SGD (Algorithm 1) on a dataset of size $n$ with $\eta_t = \eta \leq 2/L$. Then for $\bar{\mathbf{w}}_T = \frac{1}{T}\sum_{t=1}^{T}\mathbf{w}_t$ and any $\gamma \geq 1$ we have the following inequality for all $\mathbf{w}$ independent of A (can depend on S)*

$$\mathbb{E}[F(\bar{\mathbf{w}}_T) - F_S(\mathbf{w})] = O\Big(\Big(\eta + \frac{1}{\gamma} + \frac{\gamma(T + T^2/n)\eta^2}{n}\Big)\mathbb{E}[F_S(\mathbf{w})]\Big)$$
$$+ O\Big(\frac{\mathbb{E}[\|\mathbf{w}\|_2^2]}{T\eta} + \frac{\gamma\eta(1 + T/n)\mathbb{E}[\|\mathbf{w}\|_2^2]}{n}\Big). \quad (4.1)$$

A notable property of the above bound is that it holds for any $\mathbf{w}$ independent of A. If $\mathbb{E}[\|\mathbf{w}_S^*\|_2^2]$ is finite, we can choose $\mathbf{w} = \mathbf{w}_S^*$ in Eq. (4.1) and get a bound involving $\mathbb{E}[F_S(\mathbf{w}_S^*)]$. Furthermore, if we are in an interpolation or overparameterized setting [45] then $\mathbb{E}[F_S(\mathbf{w}_S^*)] = o(1/\sqrt{n})$ and the generalization will improve according to (4.1). Therefore, our stability analysis provides an explanation on how interpolation/overparameterization can help in generalization. Note SGD has an implicit bias to choose a model with a small norm and therefore it is reasonable to assume $\mathbb{E}[\|\mathbf{w}_S^*\|_2^2] < \infty$. We can also choose $\mathbf{w} = \mathbf{w}^*$ in Eq. (4.1) to get optimistic bounds in the sense of involving $F(\mathbf{w}^*)$, which decay fast if $F(\mathbf{w}^*)$ is small [56, 60]. Indeed, the following corollary gives the bound $O(1/\sqrt{n})$ in the general case and improves it to $O(1/n)$ if $F(\mathbf{w}^*) = O(1/n)$. Note here we use the assumption $F(\mathbf{w}^*) = O(1/n)$ just to show that we can get improved bound under low noise conditions. The term $F(\mathbf{w}^*)$ should be independent of $n$.

**Corollary 5.** *Let Assumptions in Theorem 4 hold.*

*(a) We can choose $\eta_t = \eta \asymp 1/\sqrt{T}$ and $T \asymp n$ to get $\mathbb{E}[F(\bar{\mathbf{w}}_T)] - F(\mathbf{w}^*) = O(1/\sqrt{n})$.*

*(b) If $F(\mathbf{w}^*) = O(1/n)$, choosing $\eta_t = \eta \leq 2/L$ and $T \asymp n$ yields $\mathbb{E}[F(\bar{\mathbf{w}}_T)] - F(\mathbf{w}^*) = O(1/n)$.*

**Remark 3.** Stability and excess risk bounds of the order $O(1/\sqrt{n})$ were studied for SGD applied to pairwise learning [40, 55]. However, these discussions require the loss functions to be smooth, Lipschitz continuous and convex. As a comparison, we remove the Lipschitz continuity assumption. Furthermore, their discussion can only imply non-optimistic bounds of the order $O(1/\sqrt{n})$. As a comparison, our discussions can fully exploit the property of $F(\mathbf{w}^*)$ to imply fast bounds $O(1/n)$.

### 4.2 Convex and Nonsmooth Problems

We now turn to pairwise learning with convex and nonsmooth functions. Theorem 6 gives the argument stability bounds based on which we develop excess risk bounds in Theorem 7. The proofs of results in this subsection are given in Section F.

**Theorem 6** (Stability bounds). *Let Assumptions 1, 2 hold. Let $S = \{z_1, \ldots, z_n\}$ and $S' = \{z_1', \ldots, z_n'\}$ be two datasets that differ by a single point. Let $\{\mathbf{w}_t\}, \{\mathbf{w}_t'\}$ be the sequence produced by SGD (Algorithm 1) w.r.t. S and S' with $\eta_t = \eta$, respectively. Then*

$$\mathbb{E}\big[\|\mathbf{w}_{t+1} - \mathbf{w}_{t+1}'\|_2^2\big] \leq 4G^2 et\big(1 + 4t/n^2\big)\eta^2. \quad (4.2)$$

*For any $\delta \in (0,1)$ with probability at least $1 - \delta$ we have*

$$\|\mathbf{w}_{t+1} - \mathbf{w}_{t+1}'\|_2^2 \leq 4G^2\eta^2 e\big(t + \big(2t/n + \log(1/\delta) + \sqrt{4tn^{-1}\log(1/\delta)}\big)^2\big).$$

**Theorem 7** (Excess risk bounds). *Let Assumptions 1, 2 hold. Let $\{\mathbf{w}_t\}$ be the sequence produced by SGD with $\eta_t = \eta \asymp T^{-\frac{3}{4}}$. If $T \asymp n^2$, then $\mathbb{E}[F(\bar{\mathbf{w}}_T)] - F(\mathbf{w}^*) = O(n^{-\frac{1}{2}})$.*

**Remark 4.** As compared to the stability bounds in the smooth case (Theorem 3), the stability bounds in (4.2) are worse in the sense that we do not have a factor of $1/n$ in (4.2). Therefore, one needs to choose very small stepsizes to let the right-hand side of (4.2) vanish to 0. Indeed, Theorem 7 suggests $\eta_t \asymp T^{-\frac{3}{4}}$, which are much smaller than the $\eta_t \asymp T^{-\frac{1}{2}}$ in the smooth case. As a result, we

require to run SGD with $T \asymp n^2$ to get the optimal excess risk bounds $O(1/\sqrt{n})$. Note we also give high-probability bounds on the argument stability in Theorem 6. It should be mentioned that the stability of a variant of SGD as $\mathbf{w}_{t+1} = \mathbf{w}_t - \frac{\eta_t}{t-1} \sum_{k=1}^{t-1} \nabla f(\mathbf{w}_t; z_{i_t}, z_{i_k})$ was recently studied for pairwise learning with convex and nonsmooth loss functions [59]. Note this update requires $O(t)$ gradient computations at the $t$-th iteration, while (3.2) only requires a single gradient computation.

---

**Algorithm 2:** Iterative Localized Algorithm for Pairwise Learning

---

**Input:** initial point $\mathbf{w}_0 = 0$, parameter $\gamma > 0, k = \lceil \frac{1}{2} \log_2 n \rceil$

1 **for** $i = 1, 2, \ldots, k$ **do**

2     set $T_i \asymp n_i = \lceil \frac{n}{2^i} \rceil, \gamma_i = \frac{\gamma}{2^i}, \eta_t = \frac{\gamma_i n_i}{t+1}, t \in \mathbb{N}, \tilde{f}(\mathbf{w}; z, z') = f(\mathbf{w}; z, z') + \frac{1}{\gamma_i n_i} \|\mathbf{w} - \mathbf{w}_{i-1}\|_2^2$

3     draw a sample $S_i$ of size $n_i$ independently from $\rho$

4     apply $\text{SGD}(S_i, T_i, \tilde{f}, \{\eta_t\})$ to minimize the following problem and get $\mathbf{w}_i$

$$\widetilde{F}_{S_i}(\mathbf{w}) := \frac{1}{n_i(n_i - 1)} \sum_{z,z' \in S_i: z \neq z'} f(\mathbf{w}; z, z') + \frac{1}{\gamma_i n_i} \|\mathbf{w} - \mathbf{w}_{i-1}\|_2^2. \qquad (4.3)$$

5 **end**

---

Note SGD requires the undesirable $O(n^2)$ gradient computations to achieve the bound $O(1/\sqrt{n})$ for nonsmooth problems. The following theorem shows one can also achieve the bound $O(1/\sqrt{n})$ with $O(n)$ gradient computations by considering Algorithm 2. Algorithm 2 is motivated from the iterative localization approach established in pointwise learning [23], which was also used to develop efficient differentially private algorithms [2, 35]. Note the choice $\gamma \asymp n^{-\frac{1}{2}} \|\mathbf{w}^*\|_2$ depends on the unknown $\|\mathbf{w}^*\|_2$. However, one can get the bound $O(D/\sqrt{n})$ by choosing $\gamma \asymp D/\sqrt{n}$ if $\|\mathbf{w}^*\|_2 \leq D$.

**Theorem 8.** *Let Assumptions 1, 2 hold and $\delta \in (0, 1)$. Let $\mathbf{w}_k$ be produced by Algorithm 2 and assume $\sup_{z,z'} |\mathbb{E}[f(\mathbf{w}_i, z, z')]| \leq B$ for some $B > 0$. If we choose $\gamma \asymp n^{-\frac{1}{2}} \|\mathbf{w}^*\|_2$, then with probability at least $1 - \delta$ we have $F(\mathbf{w}_k) - F(\mathbf{w}^*) = O(\log(\log n/\delta)(\log n)\|\mathbf{w}^*\|_2/\sqrt{n})$. Moreover, Algorithm 2 requires only $O(n)$ gradient computations to achieve this excess risk bound.*

**Remark 5.** Iterative localization approach was developed in [23, 35] to develop novel algorithms with $O(n)$ gradient computations and $O(1/\sqrt{n})$ excess risk bounds for nonsmooth pointwise learning problems. We extend this technique to the pairwise learning setting. Furthermore, the excess risk bounds in [23, 35] are stated in expectation. As a comparison, we use the novel high-probability bounds for uniformly stable algorithms established in Theorem 2 to develop high-probability bounds of the order $(\log^2 n)/\sqrt{n}$, which has not been developed even for pointwise learning (the best *high-probability* excess risk bound for SGD with nonsmooth loss functions requires $O(n^2)$ gradient computations [3]). Note that the sample size $n_k \asymp \sqrt{n}$ in the $k$-th epoch, and therefore the high-probability bounds in [40] can only yield bounds $O(n_k^{-\frac{1}{2}}) = O(n^{-\frac{1}{4}})$ for Algorithm 2. As a comparison, Theorem 2 applied to the ERM of $\widetilde{F}_{S_k}$ yields the bounds $O(n_k^{-1} + \gamma_k) = O(n^{-\frac{1}{2}})$ (ERM of $\widetilde{F}_{S_k}$ is $\gamma_k$-uniformly stable and the last term of the bound in Theorem 2 is dominated). This demonstrates the advantage of our new high-probability bounds in Theorem 2 for developing almost optimal bounds with $O(n)$ gradient computations. Another difference is that we apply iterative localization framework with $k = \lceil \frac{1}{2} \log_2 n \rceil$ instead of $k' = \lceil \log_2 n \rceil$ epochs in [23, 35]. The underlying reason is that the high-probability bounds for ERM of $\widetilde{F}_{S_k}$ involve $n_k^{-1} + \gamma_k^{-1}$, while the bounds in expectation only involve $\gamma_k^{-1}$ [40]. Since $n_{k'} \asymp 1$, our high-probability analysis only implies vacuous bounds $O(1)$ if we use $k' = \lceil \log_2 n \rceil$ epochs.

### 4.3 Strongly Convex Problems

We now turn to strongly convex cases. Theorem 9 gives bounds for smooth problems, while Theorem 10 gives excess risk bounds for nonsmooth problems. Note Theorems 9 and 10 apply to any algorithm, which give a general relationship between excess risks and optimization errors. One can plug the optimization error bounds for any algorithm to immediately derive the corresponding excess risk bounds. The proofs of results in this subsection are given in Section G.

**Theorem 9** (Strongly Convex and Smooth Problems). *Let Assumption 3 hold. Assume for all $S \in \mathcal{Z}^n$, $F_S$ is $\sigma$-strongly convex w.r.t. $\|\cdot\|_2$. Let $A$ be a randomized algorithm and $\sigma n \geq 8L$. Then*

$$\mathbb{E}[F(A(S))] - F(\mathbf{w}^*) \leq 128L\Big(\frac{16L}{n^2\sigma^2} + \frac{1}{n\sigma}\Big)\mathbb{E}\big[F_S(\mathbf{w}_S^*)\big] + \frac{2L}{\sigma}\mathbb{E}\big[F_S(A(S)) - F_S(\mathbf{w}_S^*)\big].$$

**Remark 6.** The above bound involves two components. The first component $O(1/(n\sigma))$ depends only on the landscape of the learning problem. The second component involves optimization errors. It shows that optimization is always beneficial to improve generalization for strongly convex problems. Furthermore, it also shows that one can stop the algorithm once we achieve the optimization error bounds $O(1/n)$ since further optimization would not improve essentially the generalization. Theorem 9 also implies bounds of the order $o(1/(n\sigma))$ if $F_S(\mathbf{w}^*)$ is small.

**Theorem 10** (Strongly Convex and Nonsmooth Problems). *Assume $f$ takes a structure as $f(\mathbf{w}; z, z') = \ell(\mathbf{w}; z, z') + r(\mathbf{w})$. Assume for all $z, z'$, the map $\mathbf{w} \mapsto \ell(\mathbf{w}; z, z')$ is $G$-Lipschitz. Assume for all $S \in \mathcal{Z}^n$, $F_S$ is $\sigma$-strongly convex w.r.t. $\|\cdot\|_2$. For any algorithm $A$ we have*

$$\mathbb{E}[F(A(S))] - F(\mathbf{w}^*) \leq \frac{8G^2}{n\sigma} + G\sqrt{\frac{2\mathbb{E}\big[F_S(A(S)) - F_S(\mathbf{w}_S^*)\big]}{\sigma}}.$$

We present the specific applications of the above results to SGD in Corollary G.2 (Section G).

## 5 Pairwise Learning with Nonconvex Loss Functions

In this section, we consider excess risk bounds for pairwise learning in a nonconvex setting. In this case, the excess population risk is not a reasonable measure since we cannot guarantee that the algorithm can find a global minimizer. We therefore use the norm of gradients at $A(S)$ to measure the performance of $A$ [26, 68]. In a general nonconvex setting, SGD requires to choose $\eta_t = O(1/t)$ for a meaningful stability bound [28], for which the optimization errors would decay logarithmically w.r.t the number of iterations [26]. Then, stability analysis fails to trade-off the stability and optimization for a model with good generalization performance in a general nonconvex problem. Therefore, we turn to a different uniform convergence approach in a general nonconvex setting [38]. After that, we study stability of SGD for nonconvex problems under a further PL condition.

### 5.1 Uniform Convergence of Gradients for Pairwise Learning

Our first result for nonconvex pairwise learning is a uniform convergence of gradients. Specifically, we show that the uniform deviation between population gradients and empirical gradients over a space can be bounded by the associated Rademacher chaos complexity. Let $\mathcal{W}_R = \{\mathbf{w} \in \mathcal{W} : \|\mathbf{w}\|_2 \leq R\}$ for $R > 0$. The proofs of results in this subsection are given in Section H.

**Definition 4.** Let $\mathcal{F} := \{f : \mathcal{Z}^4 \mapsto \mathbb{R}\}$ be a function class and $S = \{z_i\}_{i=1}^n \subset \mathcal{Z}$. Let $\{\epsilon_i\}_{i=1}^{\lfloor \frac{n}{2} \rfloor}$ be independent Rademacher variables with $\Pr\{\epsilon_i = 1\} = \Pr\{\epsilon_i = -1\} = 1/2$. The empirical Rademacher chaos complexity for $\mathcal{F}$ w.r.t. $S$ is defined as

$$\mathcal{U}_S(\mathcal{F}) = \frac{1}{\lfloor \frac{n}{2} \rfloor}\mathbb{E}_\epsilon\Big[\sup_{f \in \mathcal{F}} \sum_{1 \leq i < j \leq \lfloor \frac{n}{2} \rfloor} \epsilon_i\epsilon_j f(z_i, z_{i+\lfloor \frac{n}{2} \rfloor}, z_j, z_{j+\lfloor \frac{n}{2} \rfloor})\Big].$$

**Theorem 11** (Uniform Convergence of Gradients). *Let $\delta \in (0, 1)$, $R > 0$ and $S = \{z_1, \ldots, z_n\}$ be drawn independently from $\rho$. If Assumption 3 holds, then with probability at least $1 - \delta$ we have*

$$\sup_{\mathbf{w} \in \mathcal{W}_R} \big\|\nabla F(\mathbf{w}) - \nabla F_S(\mathbf{w})\big\|_2 \leq \frac{2\sqrt{2}(LR + b')\big(2 + \sqrt{\log(1/\delta)}\big)}{\sqrt{n}} + 4\sqrt{\frac{\mathcal{U}_S(\mathcal{F}_R)}{n}},$$

*where*

$$\mathcal{F}_R = \big\{(z_1, z_2, z_3, z_4) \mapsto \langle \nabla f(\mathbf{w}; z_1, z_2), \nabla f(\mathbf{w}; z_3, z_4) \rangle : \mathbf{w} \in \mathcal{W}_R\big\}.$$

We can apply the entropy integral to control the above Rademacher chaos complexity [16, 61], and get the following result. Note $d$ is the dimension of the space $\mathcal{W}$.

**Corollary 12.** *Under Assumptions of Theorem 11, with probability at least $1 - \delta$ we have*

$$\sup_{\mathbf{w} \in \mathcal{W}_R} \left\| \nabla F(\mathbf{w}) - \nabla F_S(\mathbf{w}) \right\|_2 \leq \frac{2\sqrt{2}(LR + b')}{\sqrt{n}} \left( 2 + \sqrt{96e\big(\log 2 + d\log(3e)\big)} + \sqrt{\log(1/\delta)} \right).$$

The above uniform convergence rate involves a square-root dependency on $d$. We show that this dependency can be avoided if we consider a special class of functions with a specific structure

$$f(\mathbf{w}; z, z') = \psi(\langle \mathbf{w}, \phi(x, x') \rangle, \tau(y, y')), \tag{5.1}$$

where $\phi : \mathcal{X} \times \mathcal{X} \mapsto \mathcal{W}$ is a feature map, $\psi : \mathbb{R} \times \mathbb{R} \mapsto \mathbb{R}$ is $L_\psi$-smooth w.r.t. the first argument and $\tau : \mathcal{Y} \times \mathcal{Y} \mapsto \mathbb{R}$. Loss functions of the structure (5.1) have wide applications in robust optimization and generalized linear models [24, 46]. We assume $\kappa = \sup_{x, x' \in \mathcal{X}} \|\phi(x, x')\|_2$.

**Corollary 13.** *Let $\delta \in (0, 1), R > 0$ and $S = \{z_1, \ldots, z_n\}$ be examples drawn independently from $\rho$. Suppose $f : \mathcal{W} \times \mathcal{Z}^2 \mapsto \mathbb{R}$ takes the form (5.1) with $\psi$ being $L_\psi$-smooth w.r.t. the first argument. Then the following inequality holds with probability at least $1 - \delta$*

$$\sup_{\mathbf{w} \in \mathcal{W}_R} \left\| \nabla F(\mathbf{w}) - \nabla F_S(\mathbf{w}) \right\|_2 \leq \frac{4\kappa\big(2L_\psi R\kappa + b'\big)}{\sqrt{n}} + \sqrt{\frac{8\big(L_\psi \kappa^2 R + b'\big)^2 \log(1/\delta)}{n}}.$$

**Remark 7.** Uniform convergence of gradients was studied for pointwise learning based on covering numbers [46, 64] and Rademacher complexities [24]. We extend these results to the pairwise learning setting. A key difference between pointwise learning and pairwise learning is that the empirical gradients for pairwise learning can be no longer written as a summation of i.i.d. terms. Indeed, the $n(n-1)$ terms in $\nabla F_S(\mathbf{w})$ are correlated, which introduces difficulties in applying concentration inequalities. We need to apply decoupling techniques in U-process to handle this correlation.

## 5.2 Smooth Problems

We now study the generalization performance of SGD for pairwise learning based on the uniform convergence of gradients developed in the previous subsection. We first introduce necessary assumptions. Since $\eta_t$ is always small (a typical choice is $\eta_t \asymp 1/\sqrt{T}$), Eq. (5.2) is milder than a bounded gradient assumption. Eq. (5.3) imposes a bounded variance assumption on stochastic gradients, which is a standard assumption for the analysis of SGD [26, 36, 67].

**Assumption 4.** Assume the existence of $G > 0$ and $\sigma_1 > 0$ such that

$$\sqrt{\eta_t} \|\nabla f(\mathbf{w}_t; z, z')\|_2 \leq G, \forall t \in \mathbb{N}, z, z' \in \mathcal{Z}, \tag{5.2}$$

$$\mathbb{E}_{i_t, j_t} \big[ \|\nabla f(\mathbf{w}_t; z_{i_t}, z_{j_t}) - \nabla F_S(\mathbf{w}_t)\|_2^2 \big] \leq \sigma_1^2, \quad \forall t \in \mathbb{N}. \tag{5.3}$$

Theorem 14 gives high-probability bounds on the norm of population gradients. Our basic idea is to use the following error decomposition

$$\|\nabla F(\mathbf{w}_t)\|_2^2 \leq 2\|\nabla F(\mathbf{w}_t) - \nabla F_S(\mathbf{w}_t)\|_2^2 + 2\|\nabla F_S(\mathbf{w}_t)\|_2^2.$$

We refer to the first term on the right-hand side as the generalization error for nonconvex pairwise learning, which can be bounded by the uniform convergence of gradients established in Theorem 11. The second term is the optimization error and can be addressed by techniques in optimization theory. The proof is given in Section I.

**Theorem 14** (Smooth Problems). *Let Assumptions 3 and 4 hold. Let $\{\mathbf{w}_t\}_t$ be the sequence produced by (3.2) with $\eta_t = \eta/\sqrt{T}$ and $\eta \leq \sqrt{T}/(2L)$. Then for any $\delta \in (0, 1)$, we can choose $T \asymp nd^{-1}$ to derive the following inequality with probability at least $1 - \delta$*

$$\frac{1}{T} \sum_{t=1}^{T} \|\nabla F(\mathbf{w}_t)\|_2^2 = O\Big( n^{-\frac{1}{2}} \log^2(1/\delta)\big(d + \log(1/\delta)\big)^{\frac{1}{2}} \Big). \tag{5.4}$$

*Furthermore, if $f$ takes the specific structure (5.1) we can choose $T \asymp n$ to derive the following inequality with probability at least $1 - \delta$*

$$\frac{1}{T} \sum_{t=1}^{T} \|\nabla F(\mathbf{w}_t)\|_2^2 = O\Big( n^{-\frac{1}{2}} \log^{\frac{5}{2}}(1/\delta) \Big). \tag{5.5}$$

**Remark 8.** It is clear that the bound in (5.4) is dimension-dependent, which is due to the use of the uniform convergence approach. Eq. (5.5) further shows that this dependency on the dimension can be avoided for problems of a specific structure. Note the optimization errors of SGD with $T$ iterations for nonconvex problems satisfy $\frac{1}{T}\sum_{t=1}^{T}\|\nabla F_S(\mathbf{w}_t)\|_2^2 = O(T^{-\frac{1}{2}})$ [26]. This is consistent with the error bounds $\frac{1}{T}\sum_{t=1}^{T}\|\nabla F(\mathbf{w}_t)\|_2^2 = O\big(n^{-\frac{1}{2}}\log^{\frac{5}{2}}(1/\delta)\big)$ in (5.5) by noting $T \asymp n$. Therefore, our analysis shows that the extension from optimization to generalization comes for free.

### 5.3 Gradient Dominated Problems

We now study the stability and generalization of SGD for pairwise learning with gradient-dominated objectives (or PL condition). PL condition is widely used in nonconvex learning [24, 33], and was shown to hold true for deep (linear) and shallow neural networks [10]. Intuitively speaking, PL condition means that the suboptimality measured by function values can be bounded by gradients.

**Assumption 5** (Polyak-Lojasiewicz Condition). Denote $\hat{F}_S = \inf_{\mathbf{w}' \in \mathcal{W}} F_S(\mathbf{w}')$. We assume $F_S$ satisfies PL or gradient-dominated condition (in expectation) with parameter $\beta > 0$, i.e.,

$$\mathbb{E}_S\big[F_S(\mathbf{w}) - \hat{F}_S\big] \leq \frac{1}{2\beta}\mathbb{E}_S\big[\|\nabla F_S(\mathbf{w})\|_2^2\big], \quad \forall \mathbf{w} \in \mathcal{W}. \tag{5.6}$$

Under the PL condition, we can get excess population risk bounds based on the stability analysis. The proof of Theorem 15 is given in Section J.

**Theorem 15** (Gradient Dominated Problems). *Let Assumptions 2, 3, 5 hold. Assume $|f(\mathbf{w}; z, z')| \leq B$ for all $\mathbf{w} \in \mathcal{W}, z, z' \in \mathcal{Z}$. Let $\{\mathbf{w}_t\}_t$ be the sequence produced by (3.2) with $\eta_t = \frac{2t+1}{2\beta(t+1)^2}$. Then*

$$\mathbb{E}\big[F(\mathbf{w}_T)\big] - F(\mathbf{w}^*) = O\Big(\frac{T^{\frac{L}{L+\beta}}}{n}\Big) + O\big(1/(T\beta^2)\big). \tag{5.7}$$

*We can choose $T \asymp n^{\frac{1+L/\beta}{1+2L/\beta}}\beta^{-\frac{2+2L/\beta}{1+2L/\beta}}$ to get $\mathbb{E}\big[F(\mathbf{w}_T)\big] - F(\mathbf{w}^*) = O\big(n^{-\frac{1+L/\beta}{1+2L/\beta}}\beta^{-\frac{2L/\beta}{1+2L/\beta}}\big)$.*

**Remark 9.** Note the above bounds depend on the condition number $\mathrm{cond} := L/\beta$. If $\mathrm{cond} \approx 1$, then we get $\mathbb{E}\big[F(\mathbf{w}_T)\big] - F(\mathbf{w}^*) \approx O\big(n^{-\frac{2}{3}}\big)$. As $\mathrm{cond}$ increases, the bound increases to $O(n^{-\frac{1}{2}})$.

## 6 Conclusion

In this paper, we present a systematic study on the generalization performance for pairwise learning. We develop novel high-probability bounds for uniformly stable algorithms, and apply them to develop algorithms with optimal bounds with $O(n)$ gradient computations for nonsmooth problems. We conduct the stability analysis for various convex problems including smooth, nonsmooth and strongly convex objectives, and get optimal excess population risk bounds of the order $O(1/\sqrt{n})$ for convex problems and $O(1/(n\sigma))$ for $\sigma$-strongly convex problems, respectively. We conduct the uniform convergence analysis for general nonconvex problems, which imply the bounds of the order $O(1/\sqrt{n})$ for population gradients. We further study the stability and generalization for nonconvex pairwise learning with gradient dominated objectives. Our discussions can clarify the effect of interpolation on generalization. In Section L we present preliminary experimental results to verify our stability bounds.

It would be interesting to study the stability of SGD in a general nonconvex case for getting dimension-independent bounds. It would also be very interesting to study other stochastic optimization methods for pairwise learning, including variance reduction variants and momentum technique.

### Acknowledgments

We are also grateful to the anonymous reviewers and area chairs for their insightful and constructive comments. Yiming's work is supported by National Science Foundation (NSF) under grants DMS-2110836, IIS-1816227, IIS-2110546, IIS-2103450.

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
