# Appendix for "Generalization Guarantee of SGD for Pairwise Learning"

**Yunwen Lei**[1]    **Mingrui Liu**[2]    **Yiming Ying**[3]

[1]School of Computer Science, University of Birmingham, Birmingham B15 2TT, United Kingdom
[2]Department of Computer Science, George Mason University, Fairfax, VA 22030, USA
[3]Department of Mathematics and Statistics, State University of New York at Albany, USA
y.lei@bham.ac.uk    mingruil@gmu.edu    yying@albany.edu

## A    Table of Notations

We collect in Table A.1 the notations of performance measures used in this paper.

| $\mathcal{X}$ | input space | $\mathcal{Y}$ | output space | $\mathcal{Z}$ | sample space |
|---|---|---|---|---|---|
| $S$ | training dataset | $n$ | sample size | $z_i$ | $i$-th training example |
| $f(\mathbf{w}; z, z')$ | loss function | $F_S$ | training risk | $F$ | population risk |
| $\mathbf{w}_S^*$ | $\arg\min_{\mathbf{w}} F_S(\mathbf{w})$ | $\mathbf{w}^*$ | $\arg\min_{\mathbf{w}} F(\mathbf{w})$ | $A(S)$ | output of algorithm $A$ to $S$ |
| $L$ | smoothness parameter | $G$ | Lipschitz parameter | $\sigma$ | strong convexity parameter |
| $\eta_t$ | step size | $T$ | largest iteration number | $i_t$ | randomly selected index |
| $b$ | $\sup_{z,z' \in \mathcal{Z}} f(0; z, z')$ | $b'$ | $\sup_{z,z' \in \mathcal{Z}} \|\nabla f(0; z, z')\|_2$ | $\rho$ | probability measure |

Table A.1: Table of Notations.

## B    Proof of Theorem 1

In this section, we prove Theorem 1 on the connection between on-average stability and generalization bounds, following the arguments in [15]. To this aim, we require the following lemma on the self-bounding property of smooth loss functions.

**Lemma B.1** ([20]). *Assume for all $z, z'$, the function $\mathbf{w} \mapsto f(\mathbf{w}; z, z')$ is nonnegative and $L$-smooth. Then $\|\nabla f(\mathbf{w}; z, z')\|_2^2 \le 2L f(\mathbf{w}; z, z')$.*

*Proof of Theorem 1.* Part (a) was established in [16]. We only consider Part (b). According to the symmetry between $z_i, z_j$ and $z_i', z_j'$, we know

$$\mathbb{E}[F(A(S)) - F_S(A(S))] = \frac{1}{n(n-1)} \sum_{i,j \in [n]: i \neq j} \mathbb{E}\big[F(A(S_{i,j})) - F_S(A(S))\big]$$

$$= \frac{1}{n(n-1)} \sum_{i,j \in [n]: i \neq j} \mathbb{E}\big[f(A(S_{i,j}); z_i, z_j) - f(A(S); z_i, z_j)\big], \quad \text{(B.1)}$$

35th Conference on Neural Information Processing Systems (NeurIPS 2021).

where we have used $\mathbb{E}_{z_i, z_j}\big[f(A(S_{i,j}); z_i, z_j)\big] = F(A(S_{i,j}))$ since $z_i, z_j$ are independent of $A(S_{i,j})$. By the $L$-smoothness of $f$, we know

$$f(A(S_{i,j}); z_i, z_j) - f(A(S); z_i, z_j) \leq \langle A(S_{i,j}) - A(S), \nabla f(A(S); z_i, z_j) \rangle + \frac{L}{2}\|A(S_{i,j}) - A(S)\|_2^2$$

$$\leq \|A(S_{i,j}) - A(S)\|_2 \|\nabla f(A(S); z_i, z_j)\|_2 + \frac{L}{2}\|A(S_{i,j}) - A(S)\|_2^2$$

$$\leq \frac{\gamma}{2}\|A(S_{i,j}) - A(S)\|_2^2 + \frac{1}{2\gamma}\|\nabla f(A(S); z_i, z_j)\|_2^2 + \frac{L}{2}\|A(S_{i,j}) - A(S)\|_2^2$$

$$\leq \frac{L+\gamma}{2}\|A(S_{i,j}) - A(S)\|_2^2 + \frac{L}{\gamma}f(A(S); z_i, z_j)$$

$$\leq (L+\gamma)\|A(S_{i,j}) - A(S_i)\|_2^2 + (L+\gamma)\|A(S_i) - A(S)\|_2^2 + \frac{L}{\gamma}f(A(S); z_i, z_j),$$

where we have used Lemma B.1 in the last second inequality and the following inequality in the last step
$$\|A(S_{i,j}) - A(S)\|_2^2 \leq 2\|A(S_{i,j}) - A(S_i)\|_2^2 + 2\|A(S_i) - A(S)\|_2^2.$$
Since $\mathbb{E}[\|A(S_{i,j}) - A(S_i)\|_2^2] = \mathbb{E}[\|A(S_j) - A(S)\|_2^2]$, we know

$$\mathbb{E}\big[f(A(S_{i,j}); z_i, z_j) - f(A(S); z_i, z_j)\big] \leq (L+\gamma)\mathbb{E}[\|A(S_i) - A(S)\|_2^2]$$
$$+ (L+\gamma)\mathbb{E}[\|A(S_j) - A(S)\|_2^2] + \frac{L}{\gamma}\mathbb{E}[f(A(S); z_i, z_j)].$$

We can plug the above inequality back into (B.1), and get

$$\mathbb{E}[F(A(S)) - F_S(A(S))] \leq \frac{1}{n(n-1)} \sum_{i,j \in [n]: i \neq j} \left( 2(L+\gamma)\mathbb{E}[\|A(S_i) - A(S)\|_2^2] + \frac{L}{\gamma}\mathbb{E}[f(A(S); z_i, z_j)] \right)$$

$$= \frac{2(L+\gamma)}{n} \sum_{i=1}^{n} \mathbb{E}[\|A(S_i) - A(S)\|_2^2] + \frac{L}{\gamma}\mathbb{E}[F_S(A(S))].$$

The proof is complete. $\qquad\square$

## C  Proof of Theorem 2

In this section, we prove Theorem 2. To this aim, we first introduce some lemmas. The following lemma provides moment bounds for a summation of weakly dependent and mean-zero random functions with bounded increments under a change of any single coordinate [1, 10]. We denote by $S \backslash \{z_i\}$ the set $\{z_1, \ldots, z_{i-1}, z_{i+1}, \ldots, z_n\}$. The $L_p$-norm of a real-valued random variable $Z$ is denoted by $\|Z\|_p := \big(\mathbb{E}[|Z|^p]\big)^{\frac{1}{p}}, p \geq 1$.

**Lemma C.1** ([1]). *Let $S = \{z_1, \ldots, z_n\}$ be a set of independent random variables each taking values in $\mathcal{Z}$ and $M \geq 0$. Let $h_1, \ldots, h_n$ be some functions $h_i : \mathcal{Z}^n \mapsto \mathbb{R}$ such that the following holds for any $i \in [n]$*

*1. $\big|\mathbb{E}_{S \backslash \{z_i\}}[h_i(S)]\big| \leq M$ almost surely (a.s.),*

*2. $\mathbb{E}_{z_i}\big[h_i(S)\big] = 0$ a.s.,*

*3. for any $j \in [n]$ with $j \neq i$, and $z_j'' \in \mathcal{Z}$*

$$\big|h_i(S) - h_i(z_1, \ldots, z_{j-1}, z_j'', z_{j+1}, \ldots, z_n)\big| \leq \beta. \tag{C.1}$$

*Then, for any $p \geq 2$*

$$\Big\|\sum_{i=1}^{n} h_i(S)\Big\|_p \leq 12\sqrt{2}pn\beta\lceil \log_2 n \rceil + 4M\sqrt{pn}.$$

The bounds on moments of random variables can be used to establish concentration inequalities, as shown in the following lemma [1, 10].

**Lemma C.2.** *Let $a, b \in \mathbb{R}_+$ and $\delta \in (0, 1/e)$. Let $Z$ be a random variable with $\|Z\|_p \leq \sqrt{p}a + pb$ for any $p \geq 2$. Then with probability at least $1 - \delta$*

$$|Z| \leq e\Big(a\sqrt{\log(e/\delta)} + b\log(e/\delta)\Big).$$

The following lemma relates $F(A(S) - F_S(A(S)) - \mathbb{E}[F(A(S))]$ to $\mathbb{E}_{S'}\big[f(A(S'); z_i, z_j)\big]$. A notable property is that $A(S')$ is independent of $S$ and therefore can be considered as a fixed point, which simplifies the application of concentration inequalities. Lemma C.3 is motivated by a recent work in pointwise learning [10].

**Lemma C.3.** *Let $A$ be an $\epsilon$-uniformly stable deterministic algorithm. Let $S = \{z_1, \ldots, z_n\}, S' = \{z'_1, \ldots, z'_n\}$ be independent datasets. Then for any $p \geq 2$ there holds*

$$\Big\| F(A(S)) - F_S(A(S)) - \mathbb{E}[F(A(S))] + \frac{1}{n(n-1)} \sum_{i \neq j} \mathbb{E}_{S'}\big[f(A(S'); z_i, z_j)\big] \Big\|_p \leq 4\epsilon + 96\sqrt{2}p\epsilon\lceil\log_2 n/2\rceil.$$

*Proof.* Let $p \geq 2$ be any number. It was shown that [16]

$$\Big| \mathbb{E}_{Z, \tilde{Z}}\big[f(A(S); Z, \tilde{Z})\big] - \frac{1}{n(n-1)} \sum_{i \neq j} f(A(S); z_i, z_j) - \frac{1}{n(n-1)} \sum_{i \neq j} g_{i,j}(S) \Big| \leq 4\epsilon, \quad \text{(C.2)}$$

where we introduce

$$g_{i,j}(S) = \mathbb{E}_{z'_i, z'_j}\Big[\mathbb{E}_{Z, \tilde{Z}}\big[f(A(S_{i,j}); Z, \tilde{Z})\big] - f(A(S_{i,j}); z_i, z_j)\Big], \quad \forall i, j \in [n]$$

and $S_{i,j}$ is defined in Eq. (3.4). For any $i \neq j \in [n]$, define

$$h_{i,j}(S) := g_{i,j}(S) - \mathbb{E}_{S\backslash\{z_i \cup z_j\}}g_{i,j}(S),$$

from which and (C.2) we get the following inequality for any $p \geq 1$

$$\Big\| F(A(S)) - F_S(A(S)) - \frac{1}{n(n-1)} \sum_{i \neq j} \mathbb{E}_{S\backslash\{z_i \cup z_j\}}g_{i,j}(S) \Big\|_p \leq 4\epsilon + \frac{1}{n(n-1)} \Big\| \sum_{i \neq j} h_{i,j}(S) \Big\|_p. \tag{C.3}$$

We have the following representation of U-statistic [3]

$$\frac{1}{n(n-1)} \sum_{i \neq j} h_{i,j}(S) = \frac{1}{n!} \sum_{\pi} \frac{1}{\lfloor\frac{n}{2}\rfloor} \sum_{i=1}^{\lfloor\frac{n}{2}\rfloor} h_{\pi(i),\pi(i+\lfloor\frac{n}{2}\rfloor)}(S),$$

where the sum is taken over all permutations $\pi$ of $\{1, \ldots, n\}$. It then follows from Jensen's inequality that

$$\frac{1}{n(n-1)} \Big\| \sum_{i \neq j} h_{i,j}(S) \Big\|_p \leq \frac{1}{n!} \sum_{\pi} \frac{1}{\lfloor\frac{n}{2}\rfloor} \Big\| \sum_{i=1}^{\lfloor\frac{n}{2}\rfloor} h_{\pi(i),\pi(i+\lfloor\frac{n}{2}\rfloor)}(S) \Big\|_p = \frac{1}{\lfloor\frac{n}{2}\rfloor} \Big\| \sum_{i=1}^{\lfloor\frac{n}{2}\rfloor} h_{i,i+\lfloor\frac{n}{2}\rfloor}(S) \Big\|_p, \tag{C.4}$$

where the last identity is due to the symmetry of permutations (note $\|\cdot\|_p$ involves an expectation). It is clear that

$$\mathbb{E}_{S\backslash\{z_i \cup z_{i+\lfloor\frac{n}{2}\rfloor}\}}h_{i,i+\lfloor\frac{n}{2}\rfloor}(S) = \mathbb{E}_{S\backslash\{z_i \cup z_{i+\lfloor\frac{n}{2}\rfloor}\}}\Big[g_{i,i+\lfloor\frac{n}{2}\rfloor}(S) - \mathbb{E}_{S\backslash\{z_i \cup z_{i+\lfloor\frac{n}{2}\rfloor}\}}g_{i,i+\lfloor\frac{n}{2}\rfloor}(S)\Big] = 0, \tag{C.5}$$

where $\mathbb{E}_{S\backslash\{z_i \cup z_{i+\lfloor\frac{n}{2}\rfloor}\}}$ denotes the expectation w.r.t. $S\backslash\{z_i \cup z_{i+\lfloor\frac{n}{2}\rfloor}\}$. Furthermore, there holds

$$\mathbb{E}_{z_i \cup z_{i+\lfloor\frac{n}{2}\rfloor}}[g_{i,i+\lfloor\frac{n}{2}\rfloor}(S)]$$
$$= \mathbb{E}_{z_i \cup z_{i+\lfloor\frac{n}{2}\rfloor}}\mathbb{E}_{z'_i, z'_{i+\lfloor\frac{n}{2}\rfloor}}\Big[\mathbb{E}_{Z, \tilde{Z}}\big[f(A(S_{i,i+\lfloor\frac{n}{2}\rfloor}); Z, \tilde{Z})\big] - f(A(S_{i,i+\lfloor\frac{n}{2}\rfloor}); z_i, z_{i+\lfloor\frac{n}{2}\rfloor})\Big] = 0. \tag{C.6}$$

For any $k \in [\lfloor\frac{n}{2}\rfloor]$ with $k \neq i$ and $z''_k, z''_{k+\lfloor\frac{n}{2}\rfloor} \in \mathcal{Z}$, it is clear from the uniform stability of $A$ that

$$\Big| \mathbb{E}_{z'_i, z'_{i+\lfloor\frac{n}{2}\rfloor}}\mathbb{E}_{Z, \tilde{Z}}\big[f(A(S_{i,i+\lfloor\frac{n}{2}\rfloor}); Z, \tilde{Z})\big] - \mathbb{E}_{z'_i, z'_{i+\lfloor\frac{n}{2}\rfloor}}\mathbb{E}_{Z, \tilde{Z}}\big[f(A(S^{(k,k+\lfloor\frac{n}{2}\rfloor)}_{i,i+\lfloor\frac{n}{2}\rfloor}); Z, \tilde{Z})\big] \Big| \leq 2\epsilon,$$

where $S_{i,i+\lfloor\frac{n}{2}\rfloor}^{(k,k+\lfloor\frac{n}{2}\rfloor)}$ is the set derived by replacing the $k$-th element of $S_{i,i+\lfloor\frac{n}{2}\rfloor}$ with $z_k''$ and $k+\lfloor\frac{n}{2}\rfloor$-th element with $z_{k+\lfloor\frac{n}{2}\rfloor}''$. In a similar way, one can show

$$\left|\mathbb{E}_{z_i',z_{i+\lfloor\frac{n}{2}\rfloor}'}\left[f(A(S_{i,i+\lfloor\frac{n}{2}\rfloor});z_i,z_{i+\lfloor\frac{n}{2}\rfloor})\right] - \mathbb{E}_{z_i',z_{i+\lfloor\frac{n}{2}\rfloor}'}\left[f(A(S_{i,i+\lfloor\frac{n}{2}\rfloor}^{(k,k+\lfloor\frac{n}{2}\rfloor)});z_i,z_{i+\lfloor\frac{n}{2}\rfloor})\right]\right| \le 2\epsilon.$$

We can combine the above two inequalities together and get

$$\left|g_{i,i+\lfloor\frac{n}{2}\rfloor}(S) - g_{i,i+\lfloor\frac{n}{2}\rfloor}(S^{(k,k+\lfloor\frac{n}{2}\rfloor)})\right| \le 4\epsilon,$$

where $S^{(k,k+\lfloor\frac{n}{2}\rfloor)}$ is the set derived by replacing the $k$-th element of $S$ with $z_k''$ and $k+\lfloor\frac{n}{2}\rfloor$-th element with $z_{k+\lfloor\frac{n}{2}\rfloor}''$ Similarly, one can show

$$\left|\mathbb{E}_{S\setminus\{z_i\cup z_{i+\lfloor\frac{n}{2}\rfloor}\}}\left[g_{i,i+\lfloor\frac{n}{2}\rfloor}(S)\right] - \mathbb{E}_{S\setminus\{z_i\cup z_{i+\lfloor\frac{n}{2}\rfloor}\}}\left[g_{i,i+\lfloor\frac{n}{2}\rfloor}(S^{(k,k+\lfloor\frac{n}{2}\rfloor)})\right]\right| \le 4\epsilon.$$

We can combine the above two inequalities together and get

$$\left|h_{i,i+\lfloor\frac{n}{2}\rfloor}(S) - h_{i,i+\lfloor\frac{n}{2}\rfloor}(S^{(k,k+\lfloor\frac{n}{2}\rfloor)})\right| \le 8\epsilon. \tag{C.7}$$

According to (C.5), (C.6) and (C.7), we know that the conditions of Lemma C.1 hold with $M = 0, n = \lfloor\frac{n}{2}\rfloor, \beta = 8\epsilon, z_i = z_i \cup z_{i+\lfloor\frac{n}{2}\rfloor}$ and $h_i(S) = h_{i,i+\lfloor\frac{n}{2}\rfloor}(S)$. Therefore, one can apply Lemma C.1 to show that

$$\frac{1}{\lfloor\frac{n}{2}\rfloor}\left\|\sum_{i=1}^{\lfloor\frac{n}{2}\rfloor} h_{i,i+\lfloor\frac{n}{2}\rfloor}(S)\right\|_p \le 96\sqrt{2}p\epsilon\lceil\log_2 n/2\rceil.$$

We can plug the above inequality and (C.4) back into (C.3) and get the following inequality for any $p \ge 2$

$$\left\|F(A(S)) - F_S(A(S)) - \frac{1}{n(n-1)}\sum_{i\ne j}\mathbb{E}_{S\setminus\{z_i\cup z_j\}}g_{i,j}(S)\right\|_p \le 4\epsilon + 96\sqrt{2}p\epsilon\lceil\log_2 n/2\rceil. \tag{C.8}$$

Furthermore, the symmetry between $S$ and $S'$ implies (note $\mathbb{E}_{S'}[A(S');z_i,z_j] = \mathbb{E}_{S\setminus\{z_i\cup z_j\}}\mathbb{E}_{z_i',z_j'}[f(A(S_{i,j});z_i,z_j)]$)

$$\mathbb{E}_{S\setminus\{z_i\cup z_j\}}[g_{i,j}(S)] = \mathbb{E}_{S\setminus\{z_i\cup z_j\}}\mathbb{E}_{z_i',z_j'}\left[\mathbb{E}_{Z,\tilde{Z}}\left[f(A(S_{i,j});Z,\tilde{Z})\right] - f(A(S_{i,j});z_i,z_j)\right]$$
$$= \mathbb{E}[F(A(S))] - \mathbb{E}_{S'}\left[f(A(S');z_i,z_j)\right].$$

The stated bound then follows by combining the above two inequalities together. The proof is complete. $\qquad\square$

We require a Bernstein inequality for U-Statistic [3] (inequality A.1 on page 868) to prove Theorem 2.

**Lemma C.4** (Bernstein inequality for U-Statistic). *Let $Z_1, \ldots, Z_n$ be independent variables taking values in $\mathcal{Z}$ and $q : \mathcal{Z} \times \mathcal{Z} \mapsto \mathbb{R}$. Let $B = \sup_{z,\tilde{z}}|q(z,\tilde{z})|$ and $\sigma_0^2$ be the variance of $q(Z,\tilde{Z})$. Then for any $\delta \in (0,1)$ with probability at least $1 - \delta$*

$$\left|\frac{1}{n(n-1)}\sum_{i,j\in[n]:i\ne j}q(Z_i,Z_j) - \mathbb{E}_{Z,\tilde{Z}}[q(Z,\tilde{Z})]\right| \le \frac{2B\log(1/\delta)}{3\lfloor n/2\rfloor} + \sqrt{\frac{2\sigma_0^2\log(1/\delta)}{\lfloor n/2\rfloor}}. \tag{C.9}$$

*Proof of Theorem 2.* Let $S = \{z_1, \ldots, z_n\}, S' = \{z_1', \ldots, z_n'\}$ be independent datasets. We have the following error decomposition

$$F(A(S)) - F_S(A(S)) - F(\mathbf{w}^*) + F_S(\mathbf{w}^*) = \xi+$$
$$\mathbb{E}_S[F(A(S))] - F(\mathbf{w}^*) - \frac{1}{n(n-1)}\sum_{i\ne j}\mathbb{E}_{S'}\left[f(A(S');z_i,z_j)\right] + F_S(\mathbf{w}^*),$$

where

$$\xi = F(A(S) - F_S(A(S)) - \mathbb{E}_S[F(A(S))] + \frac{1}{n(n-1)} \sum_{i \neq j} \mathbb{E}_{S'}\big[f(A(S'); z_i, z_j)\big].$$

Due to the symmetry between $S$ and $S'$ we further get

$$F(A(S)) - F_S(A(S)) - F(\mathbf{w}^*) + F_S(\mathbf{w}^*) = \xi + \mathbb{E}_{S'}[F(A(S'))] - F(\mathbf{w}^*) - \mathbb{E}_{S'}[F_S(A(S'))] + F_S(\mathbf{w}^*)$$

and therefore

$$F(A(S)) - F_S(A(S)) - F(\mathbf{w}^*) + F_S(\mathbf{w}^*) = \xi + \mathbb{E}_{S'}\Big[F(A(S')) - F(\mathbf{w}^*) - F_S(A(S')) + F_S(\mathbf{w}^*)\Big].$$
(C.10)

Note $A(S')$ is independent of $S$ and can be considered as a fixed model if we only consider the randomness induced from $S$. We now apply a concentration inequality to study the behavior of $\mathbb{E}_{S'}\big[F(A(S')) - F(\mathbf{w}^*) - F_S(A(S')) + F_S(\mathbf{w}^*)\big]$. For any $z, z'$, define

$$q(z, z') = \mathbb{E}_{S'}[f(A(S'); z, z')] - f(\mathbf{w}^*; z, z').$$

Then it is clear

$$\mathbb{E}_{S'}\Big[F(A(S')) - F(\mathbf{w}^*) - F_S(A(S')) + F_S(\mathbf{w}^*)\Big] = \mathbb{E}_{Z,Z'}[q(Z, Z')] - \frac{1}{n(n-1)} \sum_{i \neq j} q(z_i, z_j).$$

The variance of $q$ can be bounded by

$$\mathbb{E}_{Z,Z'}[q^2(Z, Z')] \leq \mathbb{E}_{Z,Z'}\mathbb{E}_{S'}\big[\big(f(A(S'); Z, Z') - f(\mathbf{w}^*; Z, Z')\big)^2\big]$$
$$= \mathbb{E}_{Z,Z',S}\big[\big(f(A(S); Z, Z') - f(\mathbf{w}^*; Z, Z')\big)^2\big],$$

where we have used the symmetry between $S$ and $S'$ as well as the Jensen's inequality. We can apply Lemma C.4 with the above $q$ to show the following inequality with probability at least $1 - \delta/2$

$$\mathbb{E}_{S'}\Big[F(A(S')) - F(\mathbf{w}^*) - F_S(A(S')) + F_S(\mathbf{w}^*)\Big] \leq \frac{2B \log(2/\delta)}{3\lfloor n/2 \rfloor} + \sqrt{\frac{2\sigma_0^2 \log(2/\delta)}{\lfloor n/2 \rfloor}}.$$

Furthermore, Lemma C.3 implies that $\|\xi\|_p \leq 2p\epsilon\big(1 + 48\sqrt{2}\lceil \log_2 n/2 \rceil\big)$ for any $p \geq 2$, from which and Lemma C.2 we derive the following inequality with probability at least $1 - \delta/2$

$$\xi \leq 2e\epsilon\big(1 + 48\sqrt{2}\lceil \log_2 n/2 \rceil\big) \log(2e/\delta).$$

We can combine the above two inequalities and Eq. (C.10) together and derive the stated inequality with probability at least $1 - \delta$. The proof is complete. $\qquad\square$

# D  Optimization Errors

The following lemma provides the optimization error bounds of SGD for convex, strongly convex and nonconvex problems. The optimization error analysis of SGD (Algorithm 1) for pairwise learning is the same as that for pointwise learning. The underlying reason is that both algorithms build an unbiased estimator (stochastic gradient) of the true gradient, and perform the update along the negative direction of the stochastic gradient. Part (a) is standard, see, e.g., [17]. Part (b) was given in [15]. Part (c) can be found in [7, 12]. Part (d) was given in [14]. Part (e) can be found in [9].

**Lemma D.1.** *Let $\{\mathbf{w}_t\}_t$ be produced by (3.2) and $\mathbf{w} \in \mathcal{W}$ be independent of SGD.*

*(a) Let $\mathbf{w}_t^{(1)} = \big(\sum_{j=1}^t \eta_j \mathbf{w}_j\big)/\sum_{j=1}^t \eta_j$. If $F_S$ is convex and Assumption 2 holds, then for all $t \in \mathbb{N}$*

$$\mathbb{E}_A[F_S(\mathbf{w}_t^{(1)})] - F_S(\mathbf{w}) \leq \frac{G^2 \sum_{j=1}^t \eta_j^2 + \|\mathbf{w}\|_2^2}{2\sum_{j=1}^t \eta_j}.$$
(D.1)

*(b) Let Assumptions 1, 3 hold. If $\eta_t \leq 1/(2L)$ and is nonincreasing, then for all $t \in \mathbb{N}$*

$$\sum_{j=1}^{t} \eta_j \mathbb{E}_A[F_S(\mathbf{w}_j) - F_S(\mathbf{w})] \leq (1/2 + L\eta_1)\|\mathbf{w}\|_2^2 + 2L\sum_{j=1}^{t} \eta_j^2 F_S(\mathbf{w}) \qquad (D.2)$$

*and*

$$\sum_{j=1}^{t} \eta_j^2 \mathbb{E}_A[F_S(\mathbf{w}_j)] \leq \eta_1\|\mathbf{w}\|_2^2 + 2\sum_{j=1}^{t} \eta_j^2 \mathbb{E}_A[F_S(\mathbf{w})]. \qquad (D.3)$$

*(c) Let $F_S$ be $\sigma$-strongly convex and $\eta_t = 2/(\sigma(t+1))$. Let $\bar{\mathbf{w}}_t' = \left(\sum_{j=1}^{t} j\mathbf{w}_j\right)/\sum_{j=1}^{t} j$. If either Assumption 3 or Assumption 2 holds, then*

$$\mathbb{E}_A[F_S(\bar{\mathbf{w}}_t')] - F_S(\mathbf{w}) = O\left(1/(t\sigma) + \|\mathbf{w}\|_2^2/t^2\right). \qquad (D.4)$$

*If Assumption 3 holds, then with probability at least $1 - \delta$*

$$F_S(\bar{\mathbf{w}}_t') - F_S(\mathbf{w}) = O\left(\log(1/\delta)/(t\sigma)\right). \qquad (D.5)$$

*(d) Let Assumptions 3, 4 hold and $\eta_j \leq 1/(2L)$. For any $\delta \in (0,1)$, the following inequality holds with probability at least $1 - \delta$*

$$\sum_{j=1}^{t} \eta_j\|\nabla F_S(\mathbf{w}_j)\|_2^2 = O\left(\sum_{j=1}^{t} \eta_j^2 + \log(1/\delta)\right). \qquad (D.6)$$

*Furthermore, the following inequality holds with probability at least $1 - \delta$ simultaneously for all $t = 1, \ldots, T$*

$$\|\mathbf{w}_{t+1}\|_2 = O\left(\left(1 + \sum_{k=1}^{T} \eta_k^2\right)^{\frac{1}{2}}\left(1 + \sum_{k=1}^{t} \eta_k\right)^{\frac{1}{2}} \log(1/\delta)\right). \qquad (D.7)$$

*(e) Let Assumptions 3, 5 hold. If $\eta_t = \frac{2t+1}{2\beta(t+1)^2}$, then*

$$\mathbb{E}_A[F_S(\mathbf{w}_t)] - \inf_{\mathbf{w}}[F_S(\mathbf{w})] = O\left(1/(t\beta^2)\right). \qquad (D.8)$$

## E   Proofs on Smooth and Convex Problems

In this section, we present the proof related to stability and generalization for pairwise learning with convex and smooth loss functions. The following lemma shows the gradient map $\mathbf{w} \mapsto \mathbf{w} - \eta\nabla f(\mathbf{w}; z, z')$ is nonexpansive, which is very useful to study the stability bounds.

**Lemma E.1** ([6]). *Assume for all $z \in \mathcal{Z}$, the function $\mathbf{w} \mapsto f(\mathbf{w}; z, z')$ is convex and $L$-smooth. Then for all $\eta \leq 2/L$ and $z, z' \in \mathcal{Z}$ there holds*

$$\|\mathbf{w} - \eta\nabla f(\mathbf{w}; z, z') - \mathbf{w}' + \eta\nabla f(\mathbf{w}'; z, z')\|_2 \leq \|\mathbf{w} - \mathbf{w}'\|_2.$$

Based on Lemma E.1, we can prove Theorem 3 on stability bounds.

*Proof of Theorem 3.* For any $i \in [n]$, define $S_i$ as (3.3). Let $\{\mathbf{w}_t\}, \{\mathbf{w}_t^{(i)}\}$ be produced by SGD (Algorithm 1) w.r.t. $S$ and $S_i$, respectively. For any $S$ and $i \in [n]$, we denote

$$L_{S,i}(\mathbf{w}) = \sum_{j\in[n]:j\neq i} \left(f(\mathbf{w}; z_i, z_j) + f(\mathbf{w}; z_j, z_i)\right), \quad L_{S_i,i}(\mathbf{w}) = \sum_{j\in[n]:j\neq i} \left(f(\mathbf{w}; z_i', z_j) + f(\mathbf{w}; z_j, z_i')\right).$$

$$(E.1)$$

If $i_t \neq i$ and $j_t \neq i$, it follows from (3.2) that

$$\|\mathbf{w}_{t+1} - \mathbf{w}_{t+1}^{(i)}\|_2^2 = \left\|\mathbf{w}_t - \mathbf{w}_t^{(i)} - \eta_t\nabla f(\mathbf{w}_t; z_{i_t}, z_{j_t}) + \eta_t\nabla f(\mathbf{w}_t^{(i)}; z_{i_t}, z_{j_t})\right\|_2^2 \leq \|\mathbf{w}_t - \mathbf{w}_t^{(i)}\|_2^2,$$

where we have used Lemma E.1 in the last inequality. If $i_t = i$, it follows from (3.2) that

$$\|\mathbf{w}_{t+1} - \mathbf{w}_{t+1}^{(i)}\|_2^2 = \left\|\mathbf{w}_t - \mathbf{w}_t^{(i)} - \eta_t \nabla f(\mathbf{w}_t; z_i, z_{j_t}) + \eta_t \nabla f(\mathbf{w}_t^{(i)}; z_i', z_{j_t})\right\|_2^2$$
$$\leq (1+p)\|\mathbf{w}_t - \mathbf{w}_t^{(i)}\|_2^2 + (1+1/p)\eta_t^2 \|\nabla f(\mathbf{w}_t; z_i, z_{j_t}) - \nabla f(\mathbf{w}_t^{(i)}; z_i', z_{j_t})\|_2^2$$
$$\leq (1+p)\|\mathbf{w}_t - \mathbf{w}_t^{(i)}\|_2^2 + 2(1+1/p)\eta_t^2 \Big(\|\nabla f(\mathbf{w}_t; z_i, z_{j_t})\|_2^2 + \|\nabla f(\mathbf{w}_t^{(i)}; z_i', z_{j_t})\|_2^2\Big)$$
$$\leq (1+p)\|\mathbf{w}_t - \mathbf{w}_t^{(i)}\|_2^2 + 4L(1+1/p)\eta_t^2 \Big(f(\mathbf{w}_t; z_i, z_{j_t}) + f(\mathbf{w}_t^{(i)}; z_i', z_{j_t})\Big),$$

where we have used the elementary inequality

$$(a+b)^2 \leq (1+p)a^2 + (1+1/p)b^2, \quad \forall p > 0$$

and the self-bounding property (Lemma B.1). If $j_t = i$, we can similarly show that

$$\|\mathbf{w}_{t+1} - \mathbf{w}_{t+1}^{(i)}\|_2^2 \leq (1+p)\|\mathbf{w}_t - \mathbf{w}_t^{(i)}\|_2^2 + 4L(1+1/p)\eta_t^2 \Big(f(\mathbf{w}_t; z_{i_t}, z_i) + f(\mathbf{w}_t^{(i)}; z_{i_t}, z_i')\Big).$$

Note the event $i_t \neq i$ and $j_t \neq i$ happens with the probability $\frac{(n-1)(n-2)}{n(n-1)}$, and $i_t = i, j_t = j$ for $i \neq j$ happens with probability $1/(n(n-1))$. We can combine the above three cases together and derive

$$\mathbb{E}_{k_t}[\|\mathbf{w}_{t+1} - \mathbf{w}_{t+1}^{(i)}\|_2^2] \leq \frac{(n-1)(n-2)}{n(n-1)}\|\mathbf{w}_t - \mathbf{w}_t^{(i)}\|_2^2$$
$$+ \frac{1}{n(n-1)} \sum_{j \in [n]: j \neq i} \Big((1+p)\|\mathbf{w}_t - \mathbf{w}_t^{(i)}\|_2^2 + 4L(1+1/p)\eta_t^2 \big(f(\mathbf{w}_t; z_i, z_j) + f(\mathbf{w}_t^{(i)}; z_i', z_j)\big)\Big)$$
$$+ \frac{1}{n(n-1)} \sum_{j \in [n]: j \neq i} \Big((1+p)\|\mathbf{w}_t - \mathbf{w}_t^{(i)}\|_2^2 + 4L(1+1/p)\eta_t^2 \big(f(\mathbf{w}_t; z_j, z_i) + f(\mathbf{w}_t^{(i)}; z_j, z_i')\big)\Big)$$
$$= \big(1 + 2p/n\big)\|\mathbf{w}_t - \mathbf{w}_t^{(i)}\|_2^2 + \frac{4L(1+1/p)\eta_t^2}{n(n-1)}\big(L_{S,i}(\mathbf{w}_t) + L_{S_i,i}(\mathbf{w}_t^{(i)})\big),$$

where $\mathbb{E}_{k_t}$ means the conditional expectation w.r.t. $k_t := (i_t, j_t)$. It then follows that

$$\frac{1}{n}\sum_{i=1}^n \mathbb{E}_{k_t}[\|\mathbf{w}_{t+1} - \mathbf{w}_{t+1}^{(i)}\|_2^2] \leq \frac{1}{n}\Big(1 + \frac{2p}{n}\Big)\sum_{i=1}^n \|\mathbf{w}_t - \mathbf{w}_t^{(i)}\|_2^2 + \frac{4L(1+1/p)\eta_t^2}{n^2(n-1)}\sum_{i=1}^n \big(L_{S,i}(\mathbf{w}_t) + L_{S_i,i}(\mathbf{w}_t^{(i)})\big).$$

We can take expectation over both sides and get

$$\frac{1}{n}\sum_{i=1}^n \mathbb{E}[\|\mathbf{w}_{t+1} - \mathbf{w}_{t+1}^{(i)}\|_2^2] \leq \frac{1}{n}\Big(1 + \frac{2p}{n}\Big)\sum_{i=1}^n \mathbb{E}[\|\mathbf{w}_t - \mathbf{w}_t^{(i)}\|_2^2] + \frac{8L(1+1/p)\eta_t^2}{n^2(n-1)}\sum_{i=1}^n \mathbb{E}\big[L_{S,i}(\mathbf{w}_t)\big],$$

where we have used the following identity due to the symmetry between $z_i$ and $z_i'$

$$\mathbb{E}[L_{S_i,i}(\mathbf{w}_t^{(i)})] = \mathbb{E}\big[L_{S,i}(\mathbf{w}_t)\big].$$

According to the definition of $L_{S,i}$ we know

$$\sum_{i=1}^n L_{S,i}(\mathbf{w}) = \sum_{i=1}^n \sum_{j \in [n]: j \neq i} \big(f(\mathbf{w}; z_i, z_j) + f(\mathbf{w}; z_j, z_i)\big) = 2n(n-1)F_S(\mathbf{w}).$$

We can combine the above two equations together and get

$$\frac{1}{n}\sum_{i=1}^n \mathbb{E}[\|\mathbf{w}_{t+1} - \mathbf{w}_{t+1}^{(i)}\|_2^2] \leq \frac{1}{n}\Big(1 + \frac{2p}{n}\Big)\sum_{i=1}^n \mathbb{E}[\|\mathbf{w}_t - \mathbf{w}_t^{(i)}\|_2^2] + \frac{16L(1+1/p)\eta_t^2}{n}\mathbb{E}\big[F_S(\mathbf{w}_t)\big].$$

We can apply the above inequality recursively and get

$$\frac{1}{n}\sum_{i=1}^n \mathbb{E}[\|\mathbf{w}_{t+1} - \mathbf{w}_{t+1}^{(i)}\|_2^2] \leq \frac{16L(1+1/p)}{n}\sum_{j=1}^t \Big(1 + \frac{2p}{n}\Big)^{t-j}\eta_j^2 \mathbb{E}[F_S(\mathbf{w}_j)].$$

We can choose $p = n/(2t)$ in the above inequality and note

$$\left(1 + \frac{2p}{n}\right)^{t-j} \leq (1 + 1/t)^t \leq e.$$

It then follows that

$$\frac{1}{n} \sum_{i=1}^{n} \mathbb{E}[\|\mathbf{w}_{t+1} - \mathbf{w}_{t+1}^{(i)}\|_2^2] \leq \frac{16L(1 + 2t/n)e}{n} \sum_{j=1}^{t} \eta_j^2 \mathbb{E}[F_S(\mathbf{w}_j)].$$

The proof is complete. $\qquad\square$

We now use the above stability bounds to prove generalization bounds in Theorem 4.

*Proof of Theorem 4.* We can plug the on-average argument stability bounds in Theorem 3 into Theorem 1 with $A(S) = \mathbf{w}_t$ and get

$$\mathbb{E}[F(\mathbf{w}_t)] \leq \frac{32L(L+\gamma)(1 + 2t/n)e}{n} \sum_{j=1}^{t-1} \eta_j^2 \mathbb{E}[F_S(\mathbf{w}_j)] + \left(1 + L/\gamma\right)\mathbb{E}[F_S(\mathbf{w}_t)].$$

Multiplying both sides by $\eta_t$ and taking a summation then gives

$$\sum_{t=1}^{T} \eta_t \mathbb{E}[F(\mathbf{w}_t)] \leq \left(1 + L/\gamma\right) \sum_{t=1}^{T} \eta_t \mathbb{E}[F_S(\mathbf{w}_t)] + \frac{32L(L+\gamma)(1 + 2T/n)e}{n} \sum_{t=1}^{T} \eta_t \sum_{j=1}^{t-1} \eta_j^2 \mathbb{E}[F_S(\mathbf{w}_j)].$$

It then follows that

$$\sum_{t=1}^{T} \eta_t \mathbb{E}[F(\mathbf{w}_t) - F_S(\mathbf{w})] \leq \left(1 + L/\gamma\right) \sum_{t=1}^{T} \eta_t \mathbb{E}[F_S(\mathbf{w}_t) - F_S(\mathbf{w})] +$$

$$L/\gamma \sum_{t=1}^{T} \eta_t \mathbb{E}[F_S(\mathbf{w})] + \frac{32L(L+\gamma)(1 + 2T/n)e}{n} \sum_{t=1}^{T} \eta_t \sum_{j=1}^{t-1} \eta_j^2 \mathbb{E}[F_S(\mathbf{w}_j)].$$

According to (D.3) and $\eta_t = \eta$, the above inequality implies further

$$\sum_{t=1}^{T} \eta \mathbb{E}[F(\mathbf{w}_t) - F_S(\mathbf{w})] \leq \left(1 + L/\gamma\right) \sum_{t=1}^{T} \eta \mathbb{E}[F_S(\mathbf{w}_t) - F_S(\mathbf{w})] +$$

$$L/\gamma T\eta \mathbb{E}[F_S(\mathbf{w})] + \frac{32L(L+\gamma)(1 + 2T/n)e}{n} \sum_{t=1}^{T} \eta\left(\eta \mathbb{E}[\|\mathbf{w}\|_2^2] + 2t\eta^2 \mathbb{E}[F_S(\mathbf{w})]\right).$$

We can plug (D.2) into the above inequality and get

$$\sum_{t=1}^{T} \eta \mathbb{E}[F(\mathbf{w}_t) - F_S(\mathbf{w})] \leq \left(1 + L/\gamma\right)\left((1/2 + L\eta)\mathbb{E}[\|\mathbf{w}\|_2^2] + 2L \sum_{t=1}^{T} \eta^2 \mathbb{E}[F_S(\mathbf{w})]\right) +$$

$$L/\gamma T\eta \mathbb{E}[F_S(\mathbf{w})] + \frac{32L(L+\gamma)(1 + 2T/n)e}{n} \sum_{t=1}^{T} \eta\left(\eta \mathbb{E}[\|\mathbf{w}\|_2^2] + 2t\eta^2 \mathbb{E}[F_S(\mathbf{w})]\right).$$

It then follows from the Jensen's inequality that

$$\mathbb{E}[F(\bar{\mathbf{w}}_T)] - \mathbb{E}[F_S(\mathbf{w})] = O\left((T\eta)^{-1}\left(\mathbb{E}[\|\mathbf{w}\|_2^2] + T\eta^2 \mathbb{E}[F_S(\mathbf{w})]\right)\right) + \frac{\mathbb{E}[F_S(\mathbf{w})]}{\gamma} +$$

$$O\left(\frac{\gamma(1 + T/n)}{n}\left(\eta \mathbb{E}[\|\mathbf{w}\|_2^2] + T\eta^2 \mathbb{E}[F_S(\mathbf{w})]\right)\right).$$

The stated bound then follows directly. The proof is complete. $\qquad\square$

Finally, we present the proof of Corollary 5.

*Proof of Corollary 5.* We choose $\mathbf{w} = \mathbf{w}^*$ in Theorem 4 and get

$$\mathbb{E}[F(\bar{\mathbf{w}}_T) - F_S(\mathbf{w}^*)] = O\Big(\Big(\eta + \frac{1}{\gamma} + \frac{\gamma(T + T^2/n)\eta^2}{n}\Big)\mathbb{E}[F_S(\mathbf{w}^*)]\Big) + O\Big(\frac{1}{T\eta} + \frac{\gamma\eta(1 + T/n)}{n}\Big). \tag{E.2}$$

Note $\mathbb{E}[F_S(\mathbf{w}^*)] = F(\mathbf{w}^*)$.

We first prove Part (a). Since $\eta \asymp 1/\sqrt{T}$ and $T \asymp n$, the inequality (E.2) becomes

$$\mathbb{E}[F(\bar{\mathbf{w}}_T)] - F(\mathbf{w}^*) = O\Big(T^{-\frac{1}{2}} + \frac{1}{\gamma} + \frac{\gamma}{T} + \frac{\gamma(1 + T/n)}{n\sqrt{T}}\Big).$$

We can choose $\gamma = \sqrt{n}$ to get that $\mathbb{E}[F(\bar{\mathbf{w}}_T)] - F(\mathbf{w}^*) = O(1/\sqrt{n})$.

We now turn to Part (b). In this case, the inequality (E.2) becomes

$$\mathbb{E}[F(\bar{\mathbf{w}}_T)] - F(\mathbf{w}^*) = O\Big(\frac{1}{n} + \frac{1}{n\gamma} + \frac{\gamma}{n}\Big).$$

We can choose $\gamma = 1$ to get $\mathbb{E}[F(\bar{\mathbf{w}}_T)] - F(\mathbf{w}^*) = O(1/n)$. The proof is complete. $\square$

# F  Proofs on Convex and Nonsmooth Problems

In this section, we present the proof related to stability and generalization for pairwise learning with convex and nonsmooth loss functions. We first prove stability (Theorem 6) and excess risk bounds (Theorem 7) for Algorithm 1. Then we move to excess risk bounds for Algorithm 2 (Theorem 8).

## F.1  Proofs of Theorem 6 and Theorem 7

We need to introduce a concentration inequality [19] which is useful for developing high-probability bounds.

**Lemma F.1** (Chernoff's Bound). *Let $X_1, \ldots, X_t$ be independent random variables taking values in $\{0, 1\}$. Let $X = \sum_{j=1}^{t} X_j$ and $\mu = \mathbb{E}[X]$. Then for any $\tilde{\delta} > 0$ with probability at least $1 - \exp\big(-\mu\tilde{\delta}^2/(2 + \tilde{\delta})\big)$ we have $X \leq (1 + \tilde{\delta})\mu$. Furthermore, for any $\delta \in (0, 1)$ with probability at least $1 - \delta$ we have*

$$X \leq \mu + \log(1/\delta) + \sqrt{2\mu \log(1/\delta)}.$$

*Proof of Theorem 6.* Suppose $S$ and $S'$ differ by the first example. If $i_t \neq 1$ and $j_t \neq 1$, then

$$\begin{aligned}
\|\mathbf{w}_{t+1} - \mathbf{w}'_{t+1}\|_2^2 &= \big\|\mathbf{w}_t - \eta_t \nabla f(\mathbf{w}_t; z_{i_t}, z_{j_t}) - \mathbf{w}'_t + \eta_t \nabla f(\mathbf{w}'_t; z'_{i_t}, z'_{j_t})\big\|_2^2 \\
&= \big\|\mathbf{w}_t - \eta_t \nabla f(\mathbf{w}_t; z_{i_t}, z_{j_t}) - \mathbf{w}'_t + \eta_t \nabla f(\mathbf{w}'_t; z_{i_t}, z_{j_t})\big\|_2^2 \\
&= \|\mathbf{w}_t - \mathbf{w}'_t\|_2^2 - \langle \mathbf{w}_t - \mathbf{w}'_t, \eta_t \nabla f(\mathbf{w}_t; z_{i_t}, z_{j_t}) - \eta_t \nabla f(\mathbf{w}'_t; z_{i_t}, z_{j_t})\rangle + 4\eta_t^2 G^2 \\
&\leq \|\mathbf{w}_t - \mathbf{w}'_t\|_2^2 + 4\eta_t^2 G^2,
\end{aligned}$$

where we have used the fact $\langle \mathbf{w}_t - \mathbf{w}'_t, \nabla f(\mathbf{w}_t; z_{i_t}, z_{j_t}) - \nabla f(\mathbf{w}'_t; z_{i_t}, z_{j_t})\rangle \geq 0$ due to the convexity of $f$. Otherwise,

$$\begin{aligned}
\|\mathbf{w}_{t+1} - \mathbf{w}'_{t+1}\|_2^2 &= \big\|\mathbf{w}_t - \eta_t \nabla f(\mathbf{w}_t; z_{i_t}, z_{j_t}) - \mathbf{w}'_t + \eta_t \nabla f(\mathbf{w}'_t; z'_{i_t}, z'_{j_t})\big\|_2^2 \\
&\leq (1 + p)\|\mathbf{w}_t - \mathbf{w}'_t\|_2^2 + (1 + 1/p)\eta_t^2 \big\|\nabla f(\mathbf{w}_t; z_{i_t}, z_{j_t}) - \nabla f(\mathbf{w}'_t; z'_{i_t}, z'_{j_t})\big\|_2^2 \\
&\leq (1 + p)\|\mathbf{w}_t - \mathbf{w}'_t\|_2^2 + 4(1 + 1/p)\eta_t^2 G^2,
\end{aligned}$$

where we have used Assumption 2. Combining the above two cases, we derive

$$\|\mathbf{w}_{t+1} - \mathbf{w}'_{t+1}\|_2^2 \leq \big(1 + p\mathbb{I}_{[i_t=1 \text{ or } j_t=1]}\big)\|\mathbf{w}_t - \mathbf{w}'_t\|_2^2 + 4G^2\eta_t^2\big(1 + p^{-1}\mathbb{I}_{[i_t=1 \text{ or } j_t=1]}\big) \tag{F.1}$$

$$= \big(1 + p\big)^{\mathbb{I}_{[i_t=1 \text{ or } j_t=1]}}\|\mathbf{w}_t - \mathbf{w}'_t\|_2^2 + 4G^2\eta_t^2\big(1 + p^{-1}\mathbb{I}_{[i_t=1 \text{ or } j_t=1]}\big), \tag{F.2}$$

where $\mathbb{I}_{[\cdot]}$ denotes the indicator function. Taking expectations over both sides of (F.1), we get

$$\mathbb{E}\big[\|\mathbf{w}_{t+1} - \mathbf{w}'_{t+1}\|_2^2\big] \leq \big(1 + 2p/n\big)\mathbb{E}[\|\mathbf{w}_t - \mathbf{w}'_t\|_2^2] + 4G^2\eta_t^2\big(1 + 2/(pn)\big),$$

where we have used $\mathbb{E}[\mathbb{I}_{[i_t=1 \text{ or } j_t=1]}] \leq 2/n$. We apply the above inequality recursively and get

$$\mathbb{E}\big[\|\mathbf{w}_{t+1} - \mathbf{w}'_{t+1}\|_2^2\big] \leq 4G^2\big(1 + 2/(pn)\big)\sum_{j=1}^{t} \eta_j^2\big(1 + 2p/n\big)^{t-j} \leq 4G^2\big(1 + 2/(pn)\big)\eta^2 t\big(1 + 2p/n\big)^t.$$

We can choose $p = n/(2t)$ and use the standard inequality $(1 + 1/t)^t \leq e$ to get

$$\mathbb{E}\big[\|\mathbf{w}_{t+1} - \mathbf{w}'_{t+1}\|_2^2\big] \leq 4G^2 et\big(1 + 4t/n^2\big)\eta^2.$$

This proves the stability bound in expectation. We now turn to high-probability bounds. It follows from (F.2) that

$$\|\mathbf{w}_{t+1} - \mathbf{w}'_{t+1}\|_2^2 \leq 4G^2\sum_{k=1}^{t} \eta_k^2\big(1 + p^{-1}\mathbb{I}_{[i_k=1 \text{ or } j_k=1]}\big)\prod_{k'=k+1}^{t}\big(1 + p\big)^{\mathbb{I}_{[i_{k'}=1 \text{ or } j_{k'}=1]}}$$

$$\leq 4G^2\eta^2\prod_{k=1}^{t}\big(1 + p\big)^{\mathbb{I}_{[i_k=1 \text{ or } j_k=1]}}\sum_{k=1}^{t}\big(1 + p^{-1}\mathbb{I}_{[i_k=1 \text{ or } j_k=1]}\big)$$

$$= 4G^2\eta^2\big(1 + p\big)^{\sum_{k=1}^{t}\mathbb{I}_{[i_k=1 \text{ or } j_k=1]}}\Big(t + p^{-1}\sum_{k=1}^{t}\mathbb{I}_{[i_k=1 \text{ or } j_k=1]}\Big).$$

We can apply Lemma F.1 with $X_k = \mathbb{I}_{[i_k=1 \text{ or } j_k=1]}, \mu \leq 2t/n$ to get the following inequality with probability at least $1 - \delta$

$$\sum_{k=1}^{t}\mathbb{I}_{[i_k=1 \text{ or } j_k=1]} \leq 2t/n + \log(1/\delta) + \sqrt{4tn^{-1}\log(1/\delta)}.$$

Therefore, with probability at least $1 - \delta$ there holds

$$\|\mathbf{w}_{t+1} - \mathbf{w}'_{t+1}\|_2^2 \leq$$
$$4G^2\eta^2\big(1 + p\big)^{2t/n + \log(1/\delta) + \sqrt{4tn^{-1}\log(1/\delta)}}\big(t + p^{-1}\big(2t/n + \log(1/\delta) + \sqrt{4tn^{-1}\log(1/\delta)}\big)\big).$$

We can choose

$$p = \frac{1}{2t/n + \log(1/\delta) + \sqrt{4tn^{-1}\log(1/\delta)}}$$

in the above inequality and derive the following inequality with probability at least $1 - \delta$ $((1 + 1/x)^x \leq e)$

$$\|\mathbf{w}_{t+1} - \mathbf{w}'_{t+1}\|_2^2 \leq 4G^2\eta^2 e\big(t + \big(2t/n + \log(1/\delta) + \sqrt{4tn^{-1}\log(1/\delta)}\big)^2\big).$$

The proof is complete. $\qquad\qquad\square$

We can use the above stability bounds to develop excess risk bounds in Theorem 7 for SGD with nonsmooth problems.

*Proof of Theorem 7.* Let $\{\mathbf{w}_t\}, \{\mathbf{w}'_t\}$ be defined in Theorem 6. According to (4.2) and Jensen's inequality, we know

$$\mathbb{E}[\|\mathbf{w}_{t+1} - \mathbf{w}'_{t+1}\|_2] \leq 2G\sqrt{2et}\big(1 + 2\sqrt{t}/n\big)\eta.$$

It then follows that SGD with $t$-iterations for nonsmooth problems is on-average loss $\epsilon$-stable with

$$\epsilon \leq 4G^2\sqrt{2et}\big(1 + 2\sqrt{t}/n\big)\eta.$$

This together with the relationship between on-average stability and generalization shows

$$\mathbb{E}[F(\mathbf{w}_t) - F_S(\mathbf{w}_t)] \leq 4G^2\sqrt{2et}\big(1 + 2\sqrt{t}/n\big)\eta.$$

We can take an average of the above inequalities to get

$$\frac{1}{n}\sum_{t=1}^{T}\mathbb{E}[F(\mathbf{w}_t) - F_S(\mathbf{w}_t)] \leq 4G^2\sqrt{2eT}\big(1 + 2\sqrt{T}/n\big)\eta.$$

It then follows from (D.1) that

$$\mathbb{E}[F(\bar{\mathbf{w}}_T)] - F(\mathbf{w}^*) = \mathbb{E}[F(\bar{\mathbf{w}}_T) - F_S(\bar{\mathbf{w}}_T)] + \mathbb{E}[F_S(\bar{\mathbf{w}}_T) - F_S(\mathbf{w}^*)]$$

$$\leq 4G^2\sqrt{2eT}\left(1 + 2\sqrt{T}/n\right)\eta + \frac{G^2 T\eta^2 + \|\mathbf{w}^*\|_2^2}{2T\eta},$$

where we have used the Jensen's inequality and (D.1). The stated bound then follows from the choice $T \asymp n^2$ and $\eta = T^{-\frac{3}{4}}$. The proof is complete. $\qquad\square$

## F.2    Proof of Theorem 8

We now turn to Theorem 8 on excess risk bounds of Algorithm 2 based on the iterative localization technique [5, 11]. We need to introduce some definitions. For any $i$, let

$$\hat{\mathbf{w}}_i = \arg\min_{\mathbf{w}} \widetilde{F}_{S_i}(\mathbf{w}). \tag{F.3}$$

Note $\mathbf{w}_i$ is derived by applying SGD with $\eta_t = \gamma_i n_i/(t+1)$ to minimize $\widetilde{F}_{S_i}(\mathbf{w})$, with the iterates weighted according to Part (c) of Lemma D.1. We need the following lemmas.

**Lemma F.2.** *Let Assumptions 1, 2 hold. For any $\delta \in (0,1)$, the following inequality holds with probability at least $1 - \delta/(2k)$: $\|\hat{\mathbf{w}}_i - \mathbf{w}_i\|_2 = O\left(\sqrt{n_i}\gamma_i \log^{\frac{1}{2}}(2k/\delta)\right)$.*

*Proof.* It is clear that $\widetilde{F}_{S_i}$ is $\lambda_i := 2/(\gamma_i n_i)$-strongly convex. According to (D.5), the following inequality holds with probability at least $1 - \delta/(2k)$

$$\widetilde{F}_{S_i}(\mathbf{w}_i) - \widetilde{F}_{S_i}(\hat{\mathbf{w}}_i) = O(\log(2k/\delta)/(T_i\lambda_i)) = O(\log(2k/\delta)/(n_i\lambda_i)).$$

It then follows from the definition of $\hat{\mathbf{w}}_i$ and the strong convexity that

$$\frac{\lambda_i}{2}\|\hat{\mathbf{w}}_i - \mathbf{w}_i\|_2^2 \leq \widetilde{F}_{S_i}(\mathbf{w}_i) - \widetilde{F}_{S_i}(\hat{\mathbf{w}}_i) = O\left(\log(2k/\delta)/(n_i\lambda_i)\right) \tag{F.4}$$

and therefore

$$\|\hat{\mathbf{w}}_i - \mathbf{w}_i\|_2^2 = O\left(\log(2k/\delta)/(n_i\lambda_i^2)\right) = O\left(n_i\gamma_i^2 \log(2k/\delta)\right).$$

The proof is complete. $\qquad\square$

The following lemma establishes the uniform stability of pairwise learning with strongly convex objectives.

**Lemma F.3** ([16]). *Suppose $f : \mathcal{W} \times \mathcal{Z} \times \mathcal{Z} \mapsto \mathbb{R}$ takes a structure $f = \ell + r$, where $\ell : \mathcal{W} \times \mathcal{Z} \times \mathcal{Z} \mapsto \mathbb{R}$ and $r : \mathcal{W} \mapsto \mathbb{R}$. Assume for all $z, z'$, we have $\|\nabla\ell(\mathbf{w}; z, z')\|_2 \leq G$. Suppose $F_S$ is $\sigma$-strongly convex and define $A$ as $A(S) = \arg\min_{\mathbf{w}\in\mathcal{W}} F_S(\mathbf{w})$. Then $A$ is $\frac{8G^2}{n\sigma}$-uniformly stable.*

The following lemma establishes the excess risk bounds for the empirical risk minimizer defined in (F.3).

**Lemma F.4.** *Let Assumptions 1, 2 hold. Let $\hat{\mathbf{w}}_i$ be defined in (F.3). With probability at least $1 - \delta/(2k)$ the following inequality holds for any $\mathbf{w} \in \mathcal{W}$*

$$F(\hat{\mathbf{w}}_i) - F(\mathbf{w}) = O\left(\gamma_i \log n_i \log(k/\delta) + n_i^{-1}\log(k/\delta)\right) + \frac{1}{\gamma_i n_i}\|\mathbf{w} - \mathbf{w}_{i-1}\|_2^2.$$

*Proof.* For any $i$, define

$$\widetilde{F}_i(\mathbf{w}) = \mathbb{E}_{Z,Z'}\left[f(\mathbf{w}; Z, Z')\right] + \frac{1}{\gamma_i n_i}\|\mathbf{w} - \mathbf{w}_{i-1}\|_2^2$$

and $\mathbf{w}_i^* = \arg\min_{\mathbf{w}} \widetilde{F}_i(\mathbf{w})$. Denote by $A_i'$ the deterministic algorithm outputting the minimizer of $\widetilde{F}_{S_i}$. Since $\widetilde{F}_{S_i}$ is $\lambda_i = 2/(\gamma_i n_i)$-strongly convex and $f$ is Lipschitz continuous, it follows

from Lemma F.3 that $A_i'$ is $4G^2\gamma_i$-uniformly stable. Furthermore, we have the following bound on variances

$$\mathbb{E}_{Z,Z',S_i}\big[\big(f(A_i'(S_i);Z,Z') - f(\mathbf{w}_i^*;Z,Z')\big)^2\big] \leq G^2\mathbb{E}_{S_i}\big[\|A_i'(S_i) - \mathbf{w}_i^*\|_2^2\big]$$

$$\leq \frac{2G^2}{\lambda_i}\mathbb{E}_{S_i}\big[\widetilde{F}_i(A_i'(S_i)) - \widetilde{F}_i(\mathbf{w}_i^*)\big] = G^2\gamma_i n_i\mathbb{E}_{S_i}\big[\widetilde{F}_i(A_i'(S_i)) - \widetilde{F}_i(\mathbf{w}_i^*)\big].$$

It follows from the definition of $\hat{\mathbf{w}}_i$ and Theorem 2 ($A = A_i'$, $F = \widetilde{F}_i$) that with probability at least $1 - \delta/(2k)$

$$\widetilde{F}_i(\hat{\mathbf{w}}_i) - \widetilde{F}_i(\mathbf{w}_i^*) \leq \widetilde{F}_i(\hat{\mathbf{w}}_i) - \widetilde{F}_{S_i}(\hat{\mathbf{w}}_i) - \widetilde{F}_i(\mathbf{w}_i^*) + \widetilde{F}_{S_i}(\mathbf{w}_i^*) =$$

$$O\Big(\gamma_i \log n_i \log(k/\delta) + n_i^{-1}\log(k/\delta) + n_i^{-\frac{1}{2}}\big(\gamma_i n_i \log(k/\delta)\mathbb{E}_{S_i}\big[\widetilde{F}_i(A_i'(S_i)) - \widetilde{F}_i(\mathbf{w}_i^*)\big]\big)^{\frac{1}{2}}\Big). \tag{F.5}$$

On the other hand, the uniform stability of $A_i'$ and Part (a) of Theorem 1 implies that

$$\mathbb{E}_{S_i}\big[\widetilde{F}_i(\hat{\mathbf{w}}_i) - \widetilde{F}_{S_i}(\hat{\mathbf{w}}_i)\big] = O(\gamma_i).$$

It then follows that (note $\mathbb{E}_{S_i}[\widetilde{F}_{S_i}(\mathbf{w}_i^*)] = \widetilde{F}_i(\mathbf{w}_i^*)$ since $\mathbf{w}_i^*$ is independent of $S_i$)

$$\mathbb{E}_{S_i}\big[\widetilde{F}_i(\hat{\mathbf{w}}_i) - \widetilde{F}_i(\mathbf{w}_i^*)\big] = \mathbb{E}_{S_i}\big[\widetilde{F}_i(\hat{\mathbf{w}}_i) - \widetilde{F}_{S_i}(\mathbf{w}_i^*)\big]$$

$$\leq \mathbb{E}_{S_i}\big[\widetilde{F}_i(\hat{\mathbf{w}}_i) - \widetilde{F}_{S_i}(\hat{\mathbf{w}}_i)\big] = O(\gamma_i).$$

We can plug the above inequality back into (F.5) and get the following inequality with probability at least $1 - \delta/(2k)$

$$\widetilde{F}_i(\hat{\mathbf{w}}_i) - \widetilde{F}_i(\mathbf{w}) \leq \widetilde{F}_i(\hat{\mathbf{w}}_i) - \widetilde{F}_i(\mathbf{w}_i^*) = O\Big(\gamma_i \log n_i \log(k/\delta) + n_i^{-1}\log(k/\delta)\Big).$$

Then the following inequality holds with probability at least $1 - \delta/(2k)$

$$F(\hat{\mathbf{w}}_i) - F(\mathbf{w}) = \widetilde{F}_i(\hat{\mathbf{w}}_i) - \widetilde{F}_i(\mathbf{w}) - \frac{1}{\gamma_i n_i}\|\hat{\mathbf{w}}_i - \mathbf{w}_{i-1}\|_2^2 + \frac{1}{\gamma_i n_i}\|\mathbf{w} - \mathbf{w}_{i-1}\|_2^2$$

$$= O\Big(\gamma_i \log n_i \log(k/\delta) + n_i^{-1}\log(k/\delta)\Big) - \frac{1}{\gamma_i n_i}\|\hat{\mathbf{w}}_i - \mathbf{w}_{i-1}\|_2^2 + \frac{1}{\gamma_i n_i}\|\mathbf{w} - \mathbf{w}_{i-1}\|_2^2.$$

The stated bound then follows directly. The proof is complete. $\qquad\square$

Based on the above lemmas, we are now ready to prove Theorem 8.

*Proof of Theorem 8.* Let $\hat{\mathbf{w}}_i$ be defined by (F.3). Let $\hat{\mathbf{w}}_0 = \mathbf{w}^*$. We have the following error decomposition

$$F(\mathbf{w}_k) - F(\mathbf{w}^*) = \sum_{i=1}^{k}\big(F(\hat{\mathbf{w}}_i) - F(\hat{\mathbf{w}}_{i-1})\big) + F(\mathbf{w}_k) - F(\hat{\mathbf{w}}_k). \tag{F.6}$$

According to Lemma F.2, we know the following inequality with probability at least $1 - \delta/(2k)$

$$F(\mathbf{w}_k) - F(\hat{\mathbf{w}}_k) \leq G\|\mathbf{w}_k - \hat{\mathbf{w}}_k\|_2 = O(\sqrt{n_k}\gamma_k \log^{\frac{1}{2}}(2k/\delta)). \tag{F.7}$$

Furthermore, we can apply Lemma F.4 with $\mathbf{w} = \hat{\mathbf{w}}_{i-1}$ for different $i$ to get the following inequality with probability $1 - \delta$

$$\sum_{i=1}^{k}\big(F(\hat{\mathbf{w}}_i) - F(\hat{\mathbf{w}}_{i-1})\big) = \sum_{i=1}^{k}\Big(O(\gamma_i \log n_i \log(k/\delta) + n_i^{-1}\log(k/\delta)) + \frac{\|\hat{\mathbf{w}}_{i-1} - \mathbf{w}_{i-1}\|_2^2}{\gamma_i n_i}\Big)$$

$$= O(\gamma_1 \log n_1 + n_1^{-1})\log(k/\delta) + \frac{\|\hat{\mathbf{w}}_0 - \mathbf{w}_0\|_2^2}{\gamma_1 n_1} + \sum_{i=2}^{k}\Big(O(\gamma_i \log n_i + n_i^{-1})\log(k/\delta) + \frac{\|\hat{\mathbf{w}}_{i-1} - \mathbf{w}_{i-1}\|_2^2}{\gamma_i n_i}\Big)$$

$$= O(\gamma_1 \log n_1 + n_1^{-1})\log(k/\delta) + \frac{\|\mathbf{w}^*\|_2^2}{\gamma_1 n_1} + \sum_{i=2}^{k}O\Big(\gamma_i \log n_i + n_i^{-1} + \frac{n_{i-1}\gamma_{i-1}^2}{\gamma_i n_i}\Big)\log(k/\delta),$$

where we have used Lemma F.2 in the last step. We can combine the above three inequalities together and get the following inequality with probability at least $1 - \delta$

$$F(\mathbf{w}_k) - F(\mathbf{w}^*) = O(\sqrt{n_k}\gamma_k \log^{1/2}(k/\delta))$$

$$+ O(\gamma_1 \log n_1 + n_1^{-1}) \log(k/\delta) + \frac{\|\mathbf{w}^*\|_2^2}{\gamma_1 n_1} + \sum_{i=2}^{k} O\Big(\gamma_i \log n_i + n_i^{-1} + \frac{n_{i-1}\gamma_{i-1}^2}{\gamma_i n_i}\Big) \log(k/\delta)$$

$$= O\Big(\sqrt{n}\gamma 2^{-k-k/2} + \gamma \log n + \frac{1}{\gamma n}\|\mathbf{w}^*\|_2^2 + \sum_{i=2}^{k} \big(2^{-i}\gamma \log n + 2^i n^{-1} + \frac{2^{1-i}n 2^{2(1-i)}\gamma^2}{2^{-i}\gamma 2^{-i}n}\big)\Big) \log(k/\delta)$$

$$= O\Big(\sqrt{n}\gamma n^{-\frac{3}{4}} + \gamma \log n + \frac{1}{\gamma n}\|\mathbf{w}^*\|_2^2 + n^{-\frac{1}{2}} + \gamma\Big) \log(\log n/\delta),$$

where we have used $2^k \asymp \sqrt{n}$ and

$$k = \frac{1}{2}\lceil \log_2 n \rceil, \quad \gamma_i = \gamma/2^i, \quad n_i = \lceil n/2^i \rceil.$$

We can take $\gamma \asymp n^{-\frac{1}{2}}\|\mathbf{w}^*\|_2$ to get $F(\mathbf{w}_k) - F(\mathbf{w}^*) = O(\log(\log n/\delta) \log n \|\mathbf{w}^*\|/\sqrt{n})$ with probability at least $1 - \delta$.

Furthermore, it is clear that the total number of gradient computations is of the order of

$$\sum_{i=1}^{k} T_i \asymp \sum_{i=1}^{k} n/2^i \asymp n.$$

The proof is complete. $\qquad\square$

## G  Proofs on Strongly Convex Problems

In this section, we present the proofs related to excess risk bounds for pairwise learning with strongly convex objectives (Theorem 9 and Theorem 10). We first prove generalization bounds for smooth problems. To this aim, we introduce a lemma.

**Lemma G.1** ([16])**.** *Assume for all $S \in \mathcal{Z}^n$, $F_S$ is $\sigma$-strongly convex w.r.t. $\|\cdot\|$. Let $A(S) = \arg\min_{\mathbf{w}\in\mathcal{W}} F_S(\mathbf{w})$ and Assumption 3 hold. If $\sigma n \geq 8L$, then*

$$\mathbb{E}\big[F(A(S))\big] - F_S(A(S)) \leq \Big(\frac{1024L^2}{n^2\sigma^2} + \frac{64L}{n\sigma}\Big)\mathbb{E}\big[F_S(A(S))\big]. \tag{G.1}$$

*Proof of Theorem 9.* According to (G.1), we know the following generalization bound for ERM applied to strongly convex and smooth problems

$$\mathbb{E}[F(\mathbf{w}_S^*) - F_S(\mathbf{w}_S^*)] \leq 64L\Big(\frac{16L}{n^2\sigma^2} + \frac{1}{n\sigma}\Big)\mathbb{E}\big[F_S(\mathbf{w}_S^*)\big]. \tag{G.2}$$

The $L$-smoothness of $f$ implies the $L$-smoothness of $F$, which implies

$$F(A(S)) - F(\mathbf{w}_S^*) \leq \langle A(S) - \mathbf{w}_S^*, \nabla F(\mathbf{w}_S^*)\rangle + \frac{L}{2}\|A(S) - \mathbf{w}_S^*\|_2^2$$

$$\leq \|A(S) - \mathbf{w}_S^*\|_2\|\nabla F(\mathbf{w}_S^*)\|_2 + \frac{L}{2}\|A(S) - \mathbf{w}_S^*\|_2^2$$

$$\leq \frac{1}{2L}\|\nabla F(\mathbf{w}_S^*)\|_2^2 + L\|A(S) - \mathbf{w}_S^*\|_2^2,$$

where we have used the Cauchy-Schwartz inequality in the last step. According to Lemma B.1 and the inequality $F_S(\mathbf{w}_S^*) \leq F_S(\mathbf{w}^*)$, we know

$$\mathbb{E}\big[\|\nabla F(\mathbf{w}_S^*)\|_2^2\big] = \mathbb{E}\big[\|\nabla F(\mathbf{w}_S^*) - \nabla F(\mathbf{w}^*)\|_2^2\big] \leq 2L\mathbb{E}[F(\mathbf{w}_S^*) - F(\mathbf{w}^*)] = 2L\mathbb{E}[F(\mathbf{w}_S^*) - F_S(\mathbf{w}^*)]$$

$$\leq 2L\mathbb{E}[F(\mathbf{w}_S^*) - F_S(\mathbf{w}_S^*)] \leq 128L^2\Big(\frac{16L}{n^2\sigma^2} + \frac{1}{n\sigma}\Big)\mathbb{E}\big[F_S(\mathbf{w}_S^*)\big],$$

where we have used (G.2). Furthermore, the $\sigma$-strong convexity of $F_S$ implies

$$\|A(S) - \mathbf{w}_S^*\|_2^2 \leq \frac{2}{\sigma}\big(F_S(A(S)) - F_S(\mathbf{w}_S^*)\big).$$

We can combine the above three inequalities together and derive

$$\mathbb{E}[F(A(S)) - F(\mathbf{w}_S^*)] \leq 64L\Big(\frac{16L}{n^2\sigma^2} + \frac{1}{n\sigma}\Big)\mathbb{E}\big[F_S(\mathbf{w}_S^*)\big] + \frac{2L}{\sigma}\mathbb{E}\big[F_S(A(S)) - F_S(\mathbf{w}_S^*)\big].$$

We can combine the above inequality and (G.2) together and get

$$\mathbb{E}[F(A(S)) - F_S(\mathbf{w}_S^*)] = \mathbb{E}[F(A(S)) - F(\mathbf{w}_S^*)] + \mathbb{E}[F(\mathbf{w}_S^*) - F_S(\mathbf{w}_S^*)] \tag{G.3}$$

$$\leq 128L\Big(\frac{16L}{n^2\sigma^2} + \frac{1}{n\sigma}\Big)\mathbb{E}\big[F_S(\mathbf{w}_S^*)\big] + \frac{2L}{\sigma}\mathbb{E}\big[F_S(A(S)) - F_S(\mathbf{w}_S^*)\big].$$

The stated bound then follows since

$$\mathbb{E}[F_S(\mathbf{w}_S^*)] \leq \mathbb{E}[F_S(\mathbf{w}^*)] = F(\mathbf{w}^*). \tag{G.4}$$

The proof is complete. $\qquad\square$

We now turn to Theorem 10 on nonsmooth problems.

*Proof of Theorem 10.* According to Lemma F.3 and Part (a) of Theorem 1, we know the following generalization bound for ERM applied to strongly convex and Lipschitz continuous problems

$$\mathbb{E}[F(\mathbf{w}_S^*) - F_S(\mathbf{w}_S^*)] \leq \frac{8G^2}{n\sigma}.$$

The Lipschitz continuity of $f$ implies the Lipschitz continuity of $F$. Therefore, it follows from the strong convexity of $F_S$ that

$$F(A(S)) - F(\mathbf{w}_S^*) \leq G\|A(S) - \mathbf{w}_S^*\|_2 \leq G\sqrt{\frac{2\big(F_S(A(S)) - F_S(\mathbf{w}_S^*)\big)}{\sigma}}.$$

We can combine the above two inequalities together and use (G.3) to derive

$$\mathbb{E}[F(A(S)) - F_S(\mathbf{w}_S^*)] \leq \frac{8G^2}{n\sigma} + G\sqrt{\frac{2\mathbb{E}\big[F_S(A(S) - F_S(\mathbf{w}_S^*))\big]}{\sigma}}.$$

The stated bound then follows from (G.4). The proof is complete. $\qquad\square$

We now consider the application to the specific SGD. It shows how we should early-stop the algorithm to get the optimal bound $O(1/(n\sigma))$. Part (a) and Part (b) are for smooth and nonsmooth cases, respectively.

**Corollary G.2** (SGD). *Let $\{\mathbf{w}_t\}$ be the sequence produced by SGD with $\eta_t = 2/(\sigma(t+1))$. Let $F_S$ be $\sigma$-strongly convex and $\bar{\mathbf{w}}_t' = \big(\sum_{j=1}^t j\mathbf{w}_j\big)/\sum_{j=1}^t j$.*

*(a) If Assumption 3 holds and $\sigma n \geq 8L$, then*

$$\mathbb{E}[F(\bar{\mathbf{w}}_T') - F(\mathbf{w}^*)] = O\Big(\frac{\mathbb{E}\big[F_S(\mathbf{w}_S^*)\big]}{n\sigma} + 1/(T\sigma^2) + \mathbb{E}[\|\mathbf{w}_S^*\|_2^2]/(T^2\sigma)\Big). \tag{G.5}$$

*In particular, one can choose $T \asymp n/\sigma$ to get the excess population risk bound $O(1/(n\sigma))$.*

*(b) If Assumption 2 holds, then*

$$\mathbb{E}[F(A(S)) - F_S(\mathbf{w}_S^*)] = \frac{8G^2}{n\sigma} + O\Big(\sqrt{\frac{1/(T\sigma) + \mathbb{E}[\|\mathbf{w}_S^*\|_2^2]/T^2}{\sigma}}\Big). \tag{G.6}$$

*In particular, one can choose $T \asymp n^2$ to get the excess population risk bound $O(1/(n\sigma))$.*

*Proof.* According to Eq. (D.4), we know

$$\mathbb{E}_A[F_S(\bar{\mathbf{w}}'_T)] - F_S(\mathbf{w}^*_S) = O\big(1/(T\sigma) + \|\mathbf{w}^*_S\|^2_2/T^2\big). \tag{G.7}$$

We first prove Part (a). We can plug the above optimization error bounds into Theorem 9 and derive

$$\mathbb{E}[F(\bar{\mathbf{w}}'_T) - F(\mathbf{w}^*)] = 128L\Big(\frac{16L}{n^2\sigma^2} + \frac{1}{n\sigma}\Big)\mathbb{E}\big[F_S(\mathbf{w}^*_S)\big] + O\Big(1/(T\sigma^2) + \mathbb{E}[\|\mathbf{w}^*_S\|^2_2]/(T^2\sigma)\Big).$$

This gives (G.5).

We now consider Part (b). We plug (G.7) into Theorem 10 and derive

$$\mathbb{E}[F(A(S)) - F_S(\mathbf{w}^*_S)] \le \frac{8G^2}{n\sigma} + O\Big(\sqrt{\frac{1/(T\sigma) + \mathbb{E}[\|\mathbf{w}^*_S\|^2_2]/T^2}{\sigma}}\Big).$$

This gives (G.6). The proof is complete. □

# H   Proofs on Uniform Convergence of Gradients for Pairwise Learning

In this section, we present the proofs on the uniform convergence of gradients (Theorem 11, Corollary 12 and Corollary 13).

## H.1   Proof of Theorem 11

To prove Theorem 11, we first introduce a useful lemma called the McDiarmid's inequality [19] for handling the concentration of functions with bounded increments.

**Lemma H.1.** *Let $c_1, \ldots, c_n \in \mathbb{R}_+$. Let $Z_1, \ldots, Z_n$ be independent random variables taking values in a set $\mathcal{Z}$, and assume that $g : \mathcal{Z}^n \mapsto \mathbb{R}$ satisfies*

$$\sup_{z_1,\ldots,z_n,\bar{z}_i \in \mathcal{Z}} |g(z_1, \cdots, z_n) - g(\cdots, z_{i-1}, \bar{z}_i, z_{i+1}, \cdots)| \le c_i \tag{H.1}$$

*for $i = 1, \ldots, n$. Then, for any $0 < \delta < 1$, with probability at least $1 - \delta$ we have*

$$g(Z_1, \ldots, Z_n) \le \mathbb{E}\big[g(Z_1, \ldots, Z_n)\big] + \Big(\frac{1}{2}\sum_{i=1}^{n} c_i^2 \log(1/\delta)\Big)^{\frac{1}{2}}.$$

The following lemma gives a high-probability bound on the uniform deviation between population gradients and empirical gradients.

**Lemma H.2.** *Let $\delta \in (0, 1)$ and $S = \{z_1, \ldots, z_n\}$ be examples drawn independently from $\rho$. Suppose Assumption 3 holds. Then with probability at least $1 - \delta$ we have*

$$\sup_{\mathbf{w} \in \mathcal{W}_R} \big\|\nabla F(\mathbf{w}) - \nabla F_S(\mathbf{w})\big\|_2 \le \frac{2}{\lfloor\frac{n}{2}\rfloor}\mathbb{E}_S\mathbb{E}_\epsilon \sup_{\mathbf{w} \in \mathcal{W}_R} \Big\|\sum_{i=1}^{\lfloor\frac{n}{2}\rfloor} \epsilon_i \nabla f(\mathbf{w}; z_i, z_{i+\lfloor\frac{n}{2}\rfloor})\Big\|_2$$

$$+ \sqrt{\frac{8(LR + b')^2 \log(1/\delta)}{n}},$$

*where $\epsilon_i$ are independent Rademacher variables.*

*Proof.* By the $L$-Lipschitz continuity of $\nabla f$, the following inequality holds for all $\mathbf{w} \in \mathcal{W}_R$

$$\|\nabla f(\mathbf{w}; z_i, z_j)\|_2 \le \|\nabla f(0; z_i, z_j)\|_2 + L\|\mathbf{w}\|_2 \le LR + b'. \tag{H.2}$$

Let $S' = \{z'_1, \ldots, z'_n\}$ be independent examples drawn independently from $\rho$ and $S_i = \{z_1, \ldots, z_{i-1}, z'_i, z_{i+1}, \ldots, z_n\}$. Then, we have

$$\Big| \sup_{\mathbf{w} \in \mathcal{W}_R} \big\|\nabla F(\mathbf{w}) - \nabla F_S(\mathbf{w})\big\|_2 - \sup_{\mathbf{w} \in \mathcal{W}_R} \big\|\nabla F(\mathbf{w}) - \nabla F_{S_i}(\mathbf{w})\big\|_2 \Big|$$

$$\le \sup_{\mathbf{w} \in \mathcal{W}_R} \Big| \big\|\nabla F(\mathbf{w}) - \nabla F_S(\mathbf{w})\big\|_2 - \big\|\nabla F(\mathbf{w}) - \nabla F_{S_i}(\mathbf{w})\big\|_2 \Big| \le \sup_{\mathbf{w} \in \mathcal{W}_R} \big\|\nabla F_S(\mathbf{w}) - \nabla F_{S_i}(\mathbf{w})\big\|_2$$

$$= \frac{1}{n(n-1)} \sup_{\mathbf{w} \in \mathcal{W}_R} \Big\| \sum_{j \in [n]: j \neq i} \big(\nabla f(\mathbf{w}; z_i, z_j) + f(\mathbf{w}; z_j, z_i) - \nabla f(\mathbf{w}; z'_i, z_j) - \nabla f(\mathbf{w}; z_j, z'_i)\big)\Big\|$$

$$\le \frac{4(n-1)}{n(n-1)}(LR + b') = \frac{4(LR + b')}{n},$$

where we have used (H.2) for all $\mathbf{w} \in \mathcal{W}_R$. Therefore, (H.1) holds with

$$g(z_1, \ldots, z_n) := \sup_{\mathbf{w} \in \mathcal{W}_R} \left[ \left\| \nabla F(\mathbf{w}) - \nabla F_S(\mathbf{w}) \right\|_2 \right]$$

and $c_i = 4(LR + b')/n$. We can apply Lemma H.1 to derive the following inequality with probability $1 - \delta$

$$\sup_{\mathbf{w} \in \mathcal{W}_R} \left\| \nabla F(\mathbf{w}) - \nabla F_S(\mathbf{w}) \right\|_2 \leq \mathbb{E}_S \left[ \sup_{\mathbf{w} \in \mathcal{W}_R} \left\| \nabla F(\mathbf{w}) - \nabla F_S(\mathbf{w}) \right\|_2 \right] + \sqrt{\frac{8 \left( LR + b' \right)^2 \log(1/\delta)}{n}}. \tag{H.3}$$

For any $\mathbf{w} \in \mathcal{W}_R$, define $q_{\mathbf{w}} : \mathcal{Z} \times \mathcal{Z} \mapsto \mathbb{R}$ as

$$q_{\mathbf{w}}(z, z') = \mathbb{E}_{Z,Z'} \left[ \nabla f(\mathbf{w}; Z, Z') \right] - \nabla f(\mathbf{w}; z, z').$$

Then it is clear that

$$\nabla F(\mathbf{w}) - \nabla F_S(\mathbf{w}) = \frac{1}{n(n-1)} \sum_{i,j \in [n]: i \neq j} q_{\mathbf{w}}(z_i, z_j).$$

Analogous to Eq. (C.4), we have

$$\mathbb{E}_S \left[ \sup_{\mathbf{w} \in \mathcal{W}_R} \left\| \nabla F(\mathbf{w}) - \nabla F_S(\mathbf{w}) \right\|_2 \right] \leq \mathbb{E}_S \left[ \sup_{\mathbf{w} \in \mathcal{W}_R} \left\| \frac{1}{\lfloor \frac{n}{2} \rfloor} \sum_{i=1}^{\lfloor \frac{n}{2} \rfloor} q_{\mathbf{w}} \left( z_i, z_{i+\lfloor \frac{n}{2} \rfloor} \right) \right\|_2 \right].$$

By the standard symmetrization trick, we get

$$\mathbb{E}_S \left[ \sup_{\mathbf{w} \in \mathcal{W}_R} \left\| \frac{1}{\lfloor \frac{n}{2} \rfloor} \sum_{i=1}^{\lfloor \frac{n}{2} \rfloor} \left( \mathbb{E}_{Z,Z'} \left[ \nabla f(\mathbf{w}; Z, Z') \right] - \nabla f(\mathbf{w}; z_i, z_{i+\lfloor \frac{n}{2} \rfloor}) \right) \right\|_2 \right]$$

$$= \mathbb{E}_S \left[ \sup_{\mathbf{w} \in \mathcal{W}_R} \left\| \frac{1}{\lfloor \frac{n}{2} \rfloor} \sum_{i=1}^{\lfloor \frac{n}{2} \rfloor} \mathbb{E}_{S'} \left[ \nabla f(\mathbf{w}; z_i', z_{i+\lfloor \frac{n}{2} \rfloor}') - \nabla f(\mathbf{w}; z_i, z_{i+\lfloor \frac{n}{2} \rfloor}) \right] \right\|_2 \right]$$

$$\leq \mathbb{E}_{S,S'} \left[ \sup_{\mathbf{w} \in \mathcal{W}_R} \left\| \frac{1}{\lfloor \frac{n}{2} \rfloor} \sum_{i=1}^{\lfloor \frac{n}{2} \rfloor} \left( \nabla f(\mathbf{w}; z_i', z_{i+\lfloor \frac{n}{2} \rfloor}') - \nabla f(\mathbf{w}; z_i, z_{i+\lfloor \frac{n}{2} \rfloor}) \right) \right\|_2 \right]$$

$$= \frac{1}{\lfloor \frac{n}{2} \rfloor} \mathbb{E}_{S,S',\boldsymbol{\epsilon}} \left[ \sup_{\mathbf{w} \in \mathcal{W}_R} \left\| \sum_{i=1}^{\lfloor \frac{n}{2} \rfloor} \epsilon_i \left( \nabla f(\mathbf{w}; z_i', z_{i+\lfloor \frac{n}{2} \rfloor}') - \nabla f(\mathbf{w}; z_i, z_{i+\lfloor \frac{n}{2} \rfloor}) \right) \right\|_2 \right]$$

$$\leq \frac{2}{\lfloor \frac{n}{2} \rfloor} \mathbb{E}_S \mathbb{E}_{\boldsymbol{\epsilon}} \sup_{\mathbf{w} \in \mathcal{W}_R} \left\| \sum_{i=1}^{\lfloor \frac{n}{2} \rfloor} \epsilon_i \nabla f(\mathbf{w}; z_i, z_{i+\lfloor \frac{n}{2} \rfloor}) \right\|_2.$$

We can plug the above two inequalities back into (H.3) to derive the stated inequality with probability $1 - \delta$. The proof is complete. $\qquad \square$

We now use Lemma H.2 to prove Theorem 11.

*Proof of Theorem 11.* According to Jensen's inequality, we know

$$\left( \mathbb{E}_{\boldsymbol{\epsilon}} \sup_{\mathbf{w} \in \mathcal{W}_R} \left[ \left\| \sum_{i=1}^{\lfloor \frac{n}{2} \rfloor} \epsilon_i \nabla f(\mathbf{w}; z_i, z_{i+\lfloor \frac{n}{2} \rfloor}) \right\|_2 \right] \right)^2 \leq \mathbb{E}_{\boldsymbol{\epsilon}} \left[ \sup_{\mathbf{w} \in \mathcal{W}_R} \left\| \sum_{i=1}^{\lfloor \frac{n}{2} \rfloor} \epsilon_i \nabla f(\mathbf{w}; z_i, z_{i+\lfloor \frac{n}{2} \rfloor}) \right\|_2^2 \right]$$

$$= \mathbb{E}_{\boldsymbol{\epsilon}} \left[ \sup_{\mathbf{w} \in \mathcal{W}_R} \left\langle \sum_{i=1}^{\lfloor \frac{n}{2} \rfloor} \epsilon_i \nabla f(\mathbf{w}; z_i, z_{i+\lfloor \frac{n}{2} \rfloor}), \sum_{i=1}^{n} \epsilon_i \nabla f(\mathbf{w}; z_i, z_{i+\lfloor \frac{n}{2} \rfloor}) \right\rangle \right]$$

$$\leq \sup_{\mathbf{w} \in \mathcal{W}_R} \sum_{i=1}^{\lfloor \frac{n}{2} \rfloor} \left\langle \nabla f(\mathbf{w}; z_i, z_{i+\lfloor \frac{n}{2} \rfloor}), \nabla f(\mathbf{w}; z_i, z_{i+\lfloor \frac{n}{2} \rfloor}) \right\rangle$$

$$+ 2\mathbb{E}_{\boldsymbol{\epsilon}} \left[ \sup_{\mathbf{w} \in \mathcal{W}_R} \sum_{1 \leq i < j \leq \lfloor \frac{n}{2} \rfloor} \epsilon_i \epsilon_j \left\langle \nabla f(\mathbf{w}; z_i, z_{i+\lfloor \frac{n}{2} \rfloor}), \nabla f(\mathbf{w}; z_j, z_{j+\lfloor \frac{n}{2} \rfloor}) \right\rangle \right]$$

$$\leq \lfloor \frac{n}{2} \rfloor (LR + b')^2 + n \mathcal{U}_S(\mathcal{F}_R), \tag{H.4}$$

where we have used (H.2) and the definition of Rademacher chaos complexities. It then follows that

$$\mathbb{E}_\epsilon \sup_{\mathbf{w}\in\mathcal{W}_R}\left[\Big\|\sum_{i=1}^n \epsilon_i \nabla f(\mathbf{w};z_i,z_{i+\lfloor\frac{n}{2}\rfloor})\Big\|_2\right] \leq \sqrt{\lfloor\tfrac{n}{2}\rfloor}(LR+b') + \sqrt{n\mathcal{U}_S(\mathcal{F}_R)}.$$

We can plug the above bound into Lemma H.2 to derive the stated bound with high probability.  $\square$

### H.2 Proof of Corollary 12

To prove Corollary 12, it suffices to estimate the involved Rademacher chaos complexity [13, 21]. We handle this term by applying the entropy integral (Lemma H.3) in terms of covering numbers.

**Definition 1** (Covering number). Let $(\mathcal{G}, d)$ be a metric space and set $\mathcal{F} \subseteq \mathcal{G}$. For any $\epsilon > 0$, a set $\mathcal{F}^\triangle \subset \mathcal{F}$ is called an $\epsilon$-cover of $\mathcal{F}$ if for every $f \in \mathcal{F}$ we can find an element $g \in \mathcal{F}^\triangle$ satisfying $d(f,g) \leq \epsilon$. The covering number $\mathcal{N}(\epsilon, \mathcal{F}, d)$ is the cardinality of the minimal $\epsilon$-cover of $\mathcal{F}$:

$$\mathcal{N}(\epsilon, \mathcal{F}, d) := \min\left\{|\mathcal{F}^\triangle| : \mathcal{F}^\triangle \text{ is an } \epsilon\text{-cover of } \mathcal{F}\right\}.$$

**Lemma H.3** ([21]). *Let $\mathcal{F} : \widetilde{\mathcal{Z}} \times \widetilde{\mathcal{Z}} \mapsto \mathbb{R}$ be a function class with $\sup_{f\in\mathcal{F}} d_S(f,0) \leq D$ and $S = \{\tilde{z}_1,\ldots,\tilde{z}_n\} \subset \widetilde{\mathcal{Z}}$, where $d_S$ is a pseudometric on $\mathcal{F}$ defined as follows*

$$d_S(f,g) := \left(\frac{1}{n^2}\sum_{1\leq i<j\leq n}|f(\tilde{z}_i,\tilde{z}_j) - g(\tilde{z}_i,\tilde{z}_j)|^2\right)^{1/2}. \tag{H.5}$$

*Then*

$$\frac{1}{n}\mathbb{E}_\epsilon\left[\sup_{f\in\mathcal{F}}\sum_{1\leq i<j\leq n}\epsilon_i\epsilon_j f(\tilde{z}_i,\tilde{z}_j)\right] \leq 24e\int_0^D \log\big(\mathcal{N}(r,\mathcal{F},d_S)+1\big)\mathrm{d}r.$$

*Proof of Corollary 12.* For any $i \in [\lfloor\frac{n}{2}\rfloor]$, we define $\tilde{z}_i = (z_i, z_{i+\lfloor\frac{n}{2}\rfloor})$ and $\tilde{f}(\mathbf{w};\tilde{z}_i) = f(\mathbf{w};z_i,z_{i+\lfloor\frac{n}{2}\rfloor})$. Then the Rademacher chaos complexity $\mathcal{U}_n(\mathcal{F}_R)$ can be written as

$$\mathcal{U}_{\widetilde{S}}(\mathcal{F}_R) = \frac{1}{\lfloor\frac{n}{2}\rfloor}\mathbb{E}_\epsilon\left[\sup_{\mathbf{w}\in\mathcal{W}_R}\sum_{1\leq i<j\leq\lfloor\frac{n}{2}\rfloor}\epsilon_i\epsilon_j\langle\nabla\tilde{f}(\mathbf{w};\tilde{z}_i),\nabla\tilde{f}(\mathbf{w};\tilde{z}_j)\rangle\right], \tag{H.6}$$

where $\widetilde{S} = \{\tilde{z}_1,\ldots,\tilde{z}_{\lfloor\frac{n}{2}\rfloor}\}$. We define a metric $d_{\widetilde{S}}$ over $\mathcal{F}_R$ by

$$d_{\widetilde{S}}(\mathbf{w},\mathbf{w}') = \left(\frac{1}{\lfloor\frac{n}{2}\rfloor^2}\sum_{1\leq i<j\leq\lfloor\frac{n}{2}\rfloor}\big|\langle\nabla\tilde{f}(\mathbf{w};\tilde{z}_i),\nabla\tilde{f}(\mathbf{w};\tilde{z}_j)\rangle - \langle\nabla\tilde{f}(\mathbf{w}';\tilde{z}_i),\nabla\tilde{f}(\mathbf{w}';\tilde{z}_j)\rangle\big|^2\right)^{1/2}.$$

For any $\mathbf{w}$ and $\mathbf{w}'$ in $\mathcal{W}_R$, there holds

$$\lfloor\tfrac{n}{2}\rfloor^2 d_{\widetilde{S}}^2(\mathbf{w},\mathbf{w}') = \sum_{1\leq i<j\leq\lfloor\frac{n}{2}\rfloor}\big|\langle\nabla\tilde{f}(\mathbf{w};\tilde{z}_i),\nabla\tilde{f}(\mathbf{w};\tilde{z}_j)\rangle - \langle\nabla\tilde{f}(\mathbf{w}';\tilde{z}_i),\nabla\tilde{f}(\mathbf{w}';\tilde{z}_j)\rangle\big|^2$$

$$\leq 2\sum_{1\leq i<j\leq\lfloor\frac{n}{2}\rfloor}\langle\nabla\tilde{f}(\mathbf{w};\tilde{z}_i)-\nabla\tilde{f}(\mathbf{w}';\tilde{z}_i),\nabla\tilde{f}(\mathbf{w};\tilde{z}_j)\rangle^2 + 2\sum_{1\leq i<j\leq\lfloor\frac{n}{2}\rfloor}\langle\nabla\tilde{f}(\mathbf{w}';\tilde{z}_i),\nabla\tilde{f}(\mathbf{w};\tilde{z}_j)-\nabla\tilde{f}(\mathbf{w}';\tilde{z}_j)\rangle^2$$

$$\leq 2\sum_{1\leq i<j\leq\lfloor\frac{n}{2}\rfloor}\big\|\nabla\tilde{f}(\mathbf{w};\tilde{z}_i)-\nabla\tilde{f}(\mathbf{w}';\tilde{z}_i)\big\|_2^2\big\|\nabla\tilde{f}(\mathbf{w};\tilde{z}_j)\big\|_2^2 + 2\sum_{1\leq i<j\leq\lfloor\frac{n}{2}\rfloor}\big\|\nabla\tilde{f}(\mathbf{w}';\tilde{z}_i)\big\|_2^2\big\|\nabla\tilde{f}(\mathbf{w};\tilde{z}_j)-\nabla\tilde{f}(\mathbf{w}';\tilde{z}_j)\big\|_2^2$$

$$\leq 2L^2\sum_{1\leq i<j\leq\lfloor\frac{n}{2}\rfloor}\Big[\big\|\nabla\tilde{f}(\mathbf{w};\tilde{z}_j)\big\|_2^2 + \big\|\nabla\tilde{f}(\mathbf{w}';\tilde{z}_i)\big\|_2^2\Big]\|\mathbf{w}-\mathbf{w}'\|_2^2$$

$$\leq L^2(LR+b')^2 n(n-1)\|\mathbf{w}-\mathbf{w}'\|_2^2, \tag{H.7}$$

where we have used the elementary inequality $(a_1+a_2)^2 \leq 2(a_1^2+a_2^2)$ and the decomposition

$$\langle\nabla\tilde{f}(\mathbf{w};\tilde{z}_i),\nabla\tilde{f}(\mathbf{w};\tilde{z}_j)\rangle - \langle\nabla\tilde{f}(\mathbf{w}';\tilde{z}_i),\nabla\tilde{f}(\mathbf{w}';\tilde{z}_j)\rangle =$$
$$\langle\nabla\tilde{f}(\mathbf{w};\tilde{z}_i)-\nabla\tilde{f}(\mathbf{w}';\tilde{z}_i),\nabla\tilde{f}(\mathbf{w};\tilde{z}_j)\rangle + \langle\nabla\tilde{f}(\mathbf{w}';\tilde{z}_i),\nabla\tilde{f}(\mathbf{w};\tilde{z}_j)-\nabla\tilde{f}(\mathbf{w}';\tilde{z}_j)\rangle$$

in the first inequality, the $L$-smoothness of $f$ in the third inequality and (H.2) in the last inequality. It then follows that

$$\log \mathcal{N}(r, \mathcal{F}_R, d_{\widetilde{S}}) \leq \log \mathcal{N}\Big(r/\big(2L(LR+b')\big), \mathcal{W}_R, d_2\Big)$$
$$\leq d \log \Big(6LR(LR+b')r^{-1}\Big),$$

where we have used the classical result $\log \mathcal{N}(r, \mathcal{W}_R, d_2) \leq d \log(3R/r)$ [18] and $d_2$ is the metric over $\mathcal{W}_R$ defined by $d_2(\mathbf{w}, \tilde{\mathbf{w}}) = \|\mathbf{w} - \tilde{\mathbf{w}}\|_2$. Furthermore, (H.7) also implies $d_{\widetilde{S}}(\mathbf{w}, 0) \leq 2LR(LR+b')$ for $\mathbf{w} \in \mathcal{W}_R$. We can now apply Lemma H.3 to show that

$$\mathcal{U}_{\widetilde{S}}(\mathcal{F}_R) \leq 24e \int_0^{2(LR+b')LR} \log\big(1 + \mathcal{N}(r, \mathcal{F}_R, d_{\widetilde{S}})\big) dr$$
$$\leq 24e \int_0^{2(LR+b')LR} \Big( \log 2 + d \log \big(6LR(LR+b')r^{-1}\big)\Big) dr$$
$$\leq 48e(LR+b')LR\big(\log 2 + d\log(3e)\big),$$

where we have used

$$\int_0^{2(LR+b')LR} \log\Big(6LR(LR+b')r^{-1}\Big) dr = 2LR(LR+b')\int_0^1 \log(3/r)dr = 2LR(LR+b')\log(3e).$$

The stated bound then follows by plugging the above bound on Rademacher chaos complexities into Theorem 11. The proof is complete. $\qquad\square$

## H.3 Proof of Corollary 13

Our scheme to prove Corollary 13 is to directly control the term $\Big\| \sum_{i=1}^{\lfloor \frac{n}{2} \rfloor} \epsilon_i \nabla f(\mathbf{w}; z_i, z_{i+\lfloor \frac{n}{2} \rfloor}) \Big\|_2$ in Lemma H.2. In more details, we show this term is related to two Gaussian processes which are more easy to handle. Our analysis requires the following classical lemma on comparison between two Gaussian processes (Slepian's lemma).

**Lemma H.4.** *Let* $\{\mathfrak{X}_\theta : \theta \in \Theta\}$ *and* $\{\mathfrak{Y}_\theta : \theta \in \Theta\}$ *be two mean-zero separable Gaussian processes indexed by the same set* $\Theta$ *and suppose that*

$$\mathbb{E}[(\mathfrak{X}_\theta - \mathfrak{X}_{\bar{\theta}})^2] \leq \mathbb{E}[(\mathfrak{Y}_\theta - \mathfrak{Y}_{\bar{\theta}})^2], \quad \forall \theta, \bar{\theta} \in \Theta. \tag{H.8}$$

*Then* $\mathbb{E}[\sup_{\theta \in \Theta} \mathfrak{X}_\theta] \leq \mathbb{E}[\sup_{\theta \in \Theta} \mathfrak{Y}_\theta]$.

**Lemma H.5.** *Suppose* $f : \mathcal{W} \times \mathcal{Z}^2 \mapsto \mathbb{R}$ *takes the form* (5.1). *Then*

$$\mathbb{E}_\epsilon \sup_{\mathbf{w} \in \mathcal{W}_R} \Big\| \sum_{i=1}^{\lfloor \frac{n}{2} \rfloor} \epsilon_i \nabla f(\mathbf{w}; z_i, z_j) \Big\|_2 \leq \sqrt{2}\big(2L_\psi R\kappa + b'\big)\Big( \sum_{i=1}^{\lfloor \frac{n}{2} \rfloor} \|\phi(x_i, x_{i+\lfloor \frac{n}{2} \rfloor})\|_2^2 \Big)^{\frac{1}{2}}.$$

*Proof.* By the structure of $f$, we know

$$\mathbb{E}_\epsilon \sup_{\mathbf{w} \in \mathcal{W}_R} \Big\| \sum_{i=1}^{\lfloor \frac{n}{2} \rfloor} \epsilon_i \nabla f(\mathbf{w}; z_i, z_{i+\lfloor \frac{n}{2} \rfloor}) \Big\|_2$$
$$= \mathbb{E}_\epsilon \sup_{\mathbf{w} \in \mathcal{W}_R} \Big\| \sum_{i=1}^{\lfloor \frac{n}{2} \rfloor} \epsilon_i \psi'\big(\langle \mathbf{w}, \phi(x_i, x_{i+\lfloor \frac{n}{2} \rfloor})\rangle, \tau(y_i, y_{i+\lfloor \frac{n}{2} \rfloor})\big)\phi(x_i, x_{i+\lfloor \frac{n}{2} \rfloor}) \Big\|_2$$
$$= \mathbb{E}_\epsilon \sup_{\mathbf{w} \in \mathcal{W}_R, \mathbf{v} \in \mathcal{W}_1} \Big\langle \sum_{i=1}^{\lfloor \frac{n}{2} \rfloor} \epsilon_i \psi'\big(\langle \mathbf{w}, \phi(x_i, x_{i+\lfloor \frac{n}{2} \rfloor})\rangle, \tau(y_i, y_{i+\lfloor \frac{n}{2} \rfloor})\big)\phi(x_i, x_{i+\lfloor \frac{n}{2} \rfloor}), \mathbf{v} \Big\rangle$$
$$\leq \mathbb{E}_g \sup_{\mathbf{w} \in \mathcal{W}_R, \mathbf{v} \in \mathcal{W}_1} \sum_{i=1}^{\lfloor \frac{n}{2} \rfloor} g_i \psi'\big(\langle \mathbf{w}, \phi(x_i, x_{i+\lfloor \frac{n}{2} \rfloor})\rangle, \tau(y_i, y_{i+\lfloor \frac{n}{2} \rfloor})\big)\langle\phi(x_i, x_{i+\lfloor \frac{n}{2} \rfloor}), \mathbf{v}\rangle, \tag{H.9}$$

where $\psi'$ denotes the derivative of $\psi$ w.r.t. the first argument, $g_1, \ldots, g_n$ are independent $N(0,1)$ random variables. Note the last step follows from the following inequality on Rademacher and Gaussian complexities

$$\mathbb{E}_\epsilon \sup_f \sum_{i=1}^{\lfloor \frac{n}{2} \rfloor} \epsilon_i f(z_i) \le \mathbb{E}_g \sup_f \sum_{i=1}^{\lfloor \frac{n}{2} \rfloor} g_i f(z_i).$$

Define two mean-zero separable Gaussian processes indexed by $\mathcal{W}_R \times \mathcal{W}_1$

$$\mathfrak{X}_{\mathbf{w},\mathbf{v}} = \sum_{i=1}^{\lfloor \frac{n}{2} \rfloor} g_i \psi'\big(\langle \mathbf{w}, \phi(x_i, x_{i+\lfloor \frac{n}{2} \rfloor})\rangle, \tau(y_i, y_{i+\lfloor \frac{n}{2} \rfloor})\big)\langle \phi(x_i, x_{i+\lfloor \frac{n}{2} \rfloor}), \mathbf{v}\rangle$$

$$\mathfrak{Y}_{\mathbf{w},\mathbf{v}} = \sqrt{2}\kappa \sum_{i=1}^{\lfloor \frac{n}{2} \rfloor} g_i \psi'\big(\langle \mathbf{w}, \phi(x_i, x_{i+\lfloor \frac{n}{2} \rfloor})\rangle, \tau(y_i, y_{i+\lfloor \frac{n}{2} \rfloor})\big) + \sqrt{2}\big(b' + L_\psi R\kappa\big) \sum_{i=1}^{\lfloor \frac{n}{2} \rfloor} \tilde{g}_i \langle \phi(x_i, x_{i+\lfloor \frac{n}{2} \rfloor}), \mathbf{v}\rangle,$$

where $\tilde{g}_1, \ldots, \tilde{g}_n$ are independent $N(0,1)$ random variables. For any $\mathbf{w}, \mathbf{w}' \in \mathcal{W}_R$ and $\mathbf{v}, \mathbf{v}' \in \mathcal{W}_1$, it follows from the independence among $g_i$ and $\mathbb{E}g_i^2 = 1, \forall i = 1, \ldots, n$ that

$$\mathbb{E}_g\big[(\mathfrak{X}_{\mathbf{w},\mathbf{v}} - \mathfrak{X}_{\mathbf{w}',\mathbf{v}'})^2\big] = \sum_{i=1}^{\lfloor \frac{n}{2} \rfloor} \Big(\psi'\big(\langle \mathbf{w}, \phi(x_i, x_{i+\lfloor \frac{n}{2} \rfloor})\rangle, \tau(y_i, y_{i+\lfloor \frac{n}{2} \rfloor})\big)\langle \phi(x_i, x_{i+\lfloor \frac{n}{2} \rfloor}), \mathbf{v}\rangle$$

$$- \psi'\big(\langle \mathbf{w}', \phi(x_i, x_{i+\lfloor \frac{n}{2} \rfloor})\rangle, \tau(y_i, y_{i+\lfloor \frac{n}{2} \rfloor})\big)\langle \phi(x_i, x_{i+\lfloor \frac{n}{2} \rfloor}), \mathbf{v}'\rangle\Big)^2$$

$$\le 2 \sum_{i=1}^{\lfloor \frac{n}{2} \rfloor} \Big(\psi'\big(\langle \mathbf{w}, \phi(x_i, x_{i+\lfloor \frac{n}{2} \rfloor})\rangle, \tau(y_i, y_{i+\lfloor \frac{n}{2} \rfloor})\big) - \psi'\big(\langle \mathbf{w}', \phi(x_i, x_{i+\lfloor \frac{n}{2} \rfloor})\rangle, \tau(y_i, y_{i+\lfloor \frac{n}{2} \rfloor})\big)\Big)^2 \big(\langle \phi(x_i, x_{i+\lfloor \frac{n}{2} \rfloor}), \mathbf{v}\rangle\big)^2$$

$$+ 2 \sum_{i=1}^{\lfloor \frac{n}{2} \rfloor} \Big(\psi'\big(\langle \mathbf{w}', \phi(x_i, x_{i+\lfloor \frac{n}{2} \rfloor})\rangle, \tau(y_i, y_{i+\lfloor \frac{n}{2} \rfloor})\big)\Big)^2 \big(\langle \phi(x_i, x_{i+\lfloor \frac{n}{2} \rfloor}), \mathbf{v}\rangle - \langle \phi(x_i, x_{i+\lfloor \frac{n}{2} \rfloor}), \mathbf{v}'\rangle\big)^2$$

$$\le 2\kappa^2 \sum_{i=1}^{\lfloor \frac{n}{2} \rfloor} \Big(\psi'\big(\langle \mathbf{w}, \phi(x_i, x_{i+\lfloor \frac{n}{2} \rfloor})\rangle, \tau(y_i, y_{i+\lfloor \frac{n}{2} \rfloor})\big) - \psi'\big(\langle \mathbf{w}', \phi(x_i, x_{i+\lfloor \frac{n}{2} \rfloor})\rangle, \tau(y_i, y_{i+\lfloor \frac{n}{2} \rfloor})\big)\Big)^2$$

$$+ 2\big(b' + L_\psi R\kappa\big)^2 \sum_{i=1}^{\lfloor \frac{n}{2} \rfloor} \big(\langle \phi(x_i, x_{i+\lfloor \frac{n}{2} \rfloor}), \mathbf{v}\rangle - \langle \phi(x_i, x_{i+\lfloor \frac{n}{2} \rfloor}), \mathbf{v}'\rangle\big)^2$$

$$= \mathbb{E}_g\big[(\mathfrak{Y}_{\mathbf{w},\mathbf{v}} - \mathfrak{Y}_{\mathbf{w}',\mathbf{v}'})^2\big],$$

where we have used $(a+b)^2 \le 2a^2 + 2b^2$, the decomposition

$$\psi'\big(\langle \mathbf{w}, \phi(x_i, x_{i+\lfloor \frac{n}{2} \rfloor})\rangle, \tau(y_i, y_{i+\lfloor \frac{n}{2} \rfloor})\big)\langle \phi(x_i, x_{i+\lfloor \frac{n}{2} \rfloor}), \mathbf{v}\rangle - \psi'\big(\langle \mathbf{w}', \phi(x_i, x_{i+\lfloor \frac{n}{2} \rfloor})\rangle, \tau(y_i, y_{i+\lfloor \frac{n}{2} \rfloor})\big)\langle \phi(x_i, x_{i+\lfloor \frac{n}{2} \rfloor}), \mathbf{v}'\rangle$$

$$= \Big(\psi'\big(\langle \mathbf{w}, \phi(x_i, x_{i+\lfloor \frac{n}{2} \rfloor})\rangle, \tau(y_i, y_{i+\lfloor \frac{n}{2} \rfloor})\big) - \psi'\big(\langle \mathbf{w}', \phi(x_i, x_{i+\lfloor \frac{n}{2} \rfloor})\rangle, \tau(y_i, y_{i+\lfloor \frac{n}{2} \rfloor})\big)\Big)\langle \phi(x_i, x_{i+\lfloor \frac{n}{2} \rfloor}), \mathbf{v}\rangle$$

$$+ \psi'\big(\langle \mathbf{w}', \phi(x_i, x_{i+\lfloor \frac{n}{2} \rfloor})\rangle, \tau(y_i, y_{i+\lfloor \frac{n}{2} \rfloor})\big)\big(\langle \phi(x_i, x_{i+\lfloor \frac{n}{2} \rfloor}), \mathbf{v}\rangle - \langle \phi(x_i, x_{i+\lfloor \frac{n}{2} \rfloor}), \mathbf{v}'\rangle\big)$$

and the following inequality due to the $L_\psi$-smoothness of $\phi$

$$\big|\psi'\big(\langle \mathbf{w}', \phi(x_i, x_{i+\lfloor \frac{n}{2} \rfloor})\rangle, \tau(y_i, y_{i+\lfloor \frac{n}{2} \rfloor})\big)\big| \le b' + L_\psi|\langle \mathbf{w}', \phi(x_i, x_{i+\lfloor \frac{n}{2} \rfloor})\rangle - 0| \le b' + L_\psi R\kappa.$$

Therefore, we can apply Lemma H.4 to show

$$\mathbb{E}_g \sup_{\mathbf{w} \in \mathcal{W}_R, \mathbf{v} \in \mathcal{W}_1} \mathfrak{X}_{\mathbf{w},\mathbf{v}} \le \mathbb{E}_g \sup_{\mathbf{w} \in \mathcal{W}_R, \mathbf{v} \in \mathcal{W}_1} \mathfrak{Y}_{\mathbf{w},\mathbf{v}}$$

$$\le \sqrt{2}\kappa \mathbb{E}_g \sup_{\mathbf{w} \in \mathcal{W}_R} \sum_{i=1}^{\lfloor \frac{n}{2} \rfloor} g_i \psi'\big(\langle \mathbf{w}, \phi(x_i, x_{i+\lfloor \frac{n}{2} \rfloor})\rangle, \tau(y_i, y_{i+\lfloor \frac{n}{2} \rfloor})\big) + \sqrt{2}\big(b' + L_\psi R\kappa\big)\mathbb{E}_g \sup_{\mathbf{v} \in \mathcal{W}_1} \sum_{i=1}^{\lfloor \frac{n}{2} \rfloor} g_i \langle \phi(x_i, x_{i+\lfloor \frac{n}{2} \rfloor}), \mathbf{v}\rangle$$

$$\le \sqrt{2}L_\psi \kappa \mathbb{E}_g \sup_{\mathbf{w} \in \mathcal{W}_R} \sum_{i=1}^{\lfloor \frac{n}{2} \rfloor} g_i \langle \mathbf{w}, \phi(x_i, x_{i+\lfloor \frac{n}{2} \rfloor})\rangle + \sqrt{2}\big(b' + L_\psi R\kappa\big)\mathbb{E}_g \sup_{\mathbf{v} \in \mathcal{W}_1} \sum_{i=1}^{\lfloor \frac{n}{2} \rfloor} g_i \langle \phi(x_i, x_{i+\lfloor \frac{n}{2} \rfloor}), \mathbf{v}\rangle,$$

where we have used the $L_\psi$-Lipschitz continuity of $\psi'$ and the contraction lemma of Gaussian complexities in the last step. Furthermore, it follows from the Jensen's inequality that

$$
\mathbb{E}_g \sup_{\mathbf{w}\in\mathcal{W}_R} \sum_{i=1}^{\lfloor\frac{n}{2}\rfloor} g_i\langle \mathbf{w}, \phi(x_i, x_{i+\lfloor\frac{n}{2}\rfloor})\rangle = \mathbb{E}_g \sup_{\mathbf{w}\in\mathcal{W}_R} \Big\langle \mathbf{w}, \sum_{i=1}^{\lfloor\frac{n}{2}\rfloor} g_i\phi(x_i, x_{i+\lfloor\frac{n}{2}\rfloor})\Big\rangle
$$

$$
\leq R\mathbb{E}_g\Big\|\sum_{i=1}^{\lfloor\frac{n}{2}\rfloor} g_i\phi(x_i, x_{i+\lfloor\frac{n}{2}\rfloor})\Big\|_2 \leq R\sqrt{\mathbb{E}_g\Big[\Big\langle \sum_{i=1}^{\lfloor\frac{n}{2}\rfloor} g_i\phi(x_i, x_{i+\lfloor\frac{n}{2}\rfloor}), \sum_{i=1}^{\lfloor\frac{n}{2}\rfloor} g_i\phi(x_i, x_{i+\lfloor\frac{n}{2}\rfloor})\Big\rangle\Big]}
$$

$$
= R\Big(\sum_{i=1}^{\lfloor\frac{n}{2}\rfloor} \|\phi(x_i, x_{i+\lfloor\frac{n}{2}\rfloor})\|_2^2\Big)^{\frac{1}{2}}.
$$

In a similar way, we can show

$$
\mathbb{E}_g \sup_{\mathbf{v}\in\mathcal{W}_1} \sum_{i=1}^{\lfloor\frac{n}{2}\rfloor} g_i\langle \phi(x_i, x_{i+\lfloor\frac{n}{2}\rfloor}), \mathbf{v}\rangle \leq \Big(\sum_{i=1}^{\lfloor\frac{n}{2}\rfloor} \|\phi(x_i, x_{i+\lfloor\frac{n}{2}\rfloor})\|_2^2\Big)^{\frac{1}{2}}.
$$

Therefore,

$$
\mathbb{E}_g \sup_{\mathbf{w}\in\mathcal{W}_R, \mathbf{v}\in\mathcal{W}_1} \mathfrak{X}_{\mathbf{w},\mathbf{v}} \leq \big(2L_\psi R\kappa + b'\big)\sqrt{2}\Big(\sum_{i=1}^{\lfloor\frac{n}{2}\rfloor} \|\phi(x_i, x_{i+\lfloor\frac{n}{2}\rfloor})\|_2^2\Big)^{\frac{1}{2}}.
$$

Plugging the above inequality into (H.9) then gives the stated bound. The proof is complete. □

We now apply Lemma H.5 to prove Corollary 13.

*Proof of Corollary 13.* By Lemma H.5 and the definition of $\kappa$, we know

$$
\mathbb{E}_\epsilon \sup_{\mathbf{w}\in\mathcal{W}_R} \Big\|\sum_{i=1}^{\lfloor\frac{n}{2}\rfloor} \epsilon_i\nabla f(\mathbf{w}; z_i, z_j)\Big\|_2 \leq \sqrt{n}\kappa\big(2L_\psi R\kappa + b'\big). \tag{H.10}
$$

According to the $L_\psi$-smoothness of $\psi$, the function $f$ is $(L_\psi\kappa^2)$-smooth

$$
\big\|\nabla f(\mathbf{w}; z, z') - \nabla f(\tilde{\mathbf{w}}; z, z')\big\|_2
$$
$$
= \big|\psi'(\langle \mathbf{w}, \phi(x, x')\rangle, \tau(y, y')) - \psi'(\langle \tilde{\mathbf{w}}, \phi(x, x')\rangle, \tau(y, y'))\big|\|\phi(x, x')\|_2
$$
$$
\leq L_\psi|\langle \mathbf{w} - \tilde{\mathbf{w}}, \phi(x, x')\rangle|\|\phi(x, x')\|_2 \leq L_\psi\kappa^2\|\mathbf{w} - \tilde{\mathbf{w}}\|_2.
$$

Therefore, Lemma H.2 holds with $L = L_\psi\kappa^2$. We can plug (H.10) into Lemma H.2 and get the stated bound. The proof is complete. □

# I  Proofs on Nonconvex Problems

In this section, we apply the uniform convergence of gradients to prove Theorem 14.

*Proof of Theorem 14.* By the elementary inequality $(a + b)^2 \leq 2(a^2 + b^2)$ and (D.6), we derive the following inequality with probability $1 - \delta/3$

$$
\sum_{t=1}^T \eta_t\|\nabla F(\mathbf{w}_t)\|_2^2 = \sum_{t=1}^T \eta_t\big\|\nabla F(\mathbf{w}_t) - \nabla F_S(\mathbf{w}_t) + \nabla F_S(\mathbf{w}_t)\big\|_2^2
$$

$$
\leq 2\sum_{t=1}^T \eta_t\big\|\nabla F(\mathbf{w}_t) - \nabla F_S(\mathbf{w}_t)\big\|_2^2 + 2\sum_{t=1}^T \eta_t\big\|\nabla F_S(\mathbf{w}_t)\big\|_2^2
$$

$$
\leq 2\sum_{t=1}^T \eta_t \max_{t=1,\ldots,T}\big\|\nabla F(\mathbf{w}_t) - \nabla F_S(\mathbf{w}_t)\big\|_2^2 + O\Big(\sum_{t=1}^T \eta_t^2 + \log(1/\delta)\Big).
$$

It then follows that

$$\frac{1}{T}\sum_{t=1}^{T}\|\nabla F(\mathbf{w}_t)\|_2^2 \leq 2 \max_{t=1,\ldots,T}\|\nabla F(\mathbf{w}_t) - \nabla F_S(\mathbf{w}_t)\|_2^2 + O(1)\Big(\sum_{t=1}^{T}\eta_t\Big)^{-1}\Big(\sum_{t=1}^{T}\eta_t^2 + \log(1/\delta)\Big)$$

$$= 2 \max_{t=1,\ldots,T}\|\nabla F(\mathbf{w}_t) - \nabla F_S(\mathbf{w}_t)\|_2^2 + O\Big(T^{-\frac{1}{2}}\log(1/\delta)\Big). \tag{I.1}$$

According to (D.7), with probability $1 - \delta/3$ we have the following inequality uniformly for all $t = 1, \ldots, T$

$$\|\mathbf{w}_t\|_2 \leq R_T := O\Big(T^{\frac{1}{4}}\log(1/\delta)\Big). \tag{I.2}$$

According to Corollary 12, the following inequality holds with probability $1 - \delta/3$ (we assume $R_T \geq 1$)

$$\sup_{\mathbf{w}\in\mathcal{W}_{R_T}}\|\nabla F(\mathbf{w}) - \nabla F_S(\mathbf{w})\|_2 = O\Big(R_T\sqrt{d + \log(1/\delta)}n^{-\frac{1}{2}}\Big). \tag{I.3}$$

Combining (I.1), (I.2) and (I.3) together, with probability $1 - \delta$ we derive the following inequality

$$\frac{1}{T}\sum_{t=1}^{T}\|\nabla F(\mathbf{w}_t)\|_2^2 = 2 \max_{t=1,\ldots,T}\|\nabla F(\mathbf{w}_t) - \nabla F_S(\mathbf{w}_t)\|_2^2 + O\Big(T^{-\frac{1}{2}}\log(1/\delta)\Big)$$

$$= O\Big(R_T^2(d + \log(1/\delta))n^{-1}\Big) + O\Big(T^{-\frac{1}{2}}\log(1/\delta)\Big)$$

$$= O\Big(\sqrt{T}\log^2(1/\delta)(d + \log(1/\delta))n^{-1}\Big) + O\Big(T^{-\frac{1}{2}}\log(1/\delta)\Big).$$

Therefore, we can choose $T \asymp nd^{-1}$ to derive the following inequality with probability $1 - \delta$

$$\frac{1}{T}\sum_{t=1}^{T}\|\nabla F(\mathbf{w}_t)\|_2^2 = O\Big(n^{-\frac{1}{2}}\log^2(1/\delta)(d + \log(1/\delta))^{\frac{1}{2}}\Big).$$

This gives the bound (5.4).

The proof of (5.5) is the same except using the uniform convergence of gradients established in Corollary 13 instead of Corollary 12. We omit the proof for simplicity. The proof is complete. $\square$

## J Proofs on Gradient Dominated Problems

In this section, we prove Theorem 15 on excess risk bounds for learning with gradient dominated problems. The following lemma is a simple extension of a similar result in [6].

**Lemma J.1.** *Assume for all $z, z'$, the function $\mathbf{w} \mapsto f(\mathbf{w}; z, z')$ is nonnegative and $G$-Lipschitz. Let $S = \{z_1, \ldots, z_n\}$ and $S' = \{z'_1, \ldots, z'_n\}$ be two datasets that differ by the first point. Let $\{\mathbf{w}_t\}, \{\mathbf{w}'_t\}$ be the sequence produced by SGD (Algorithm 1) w.r.t. $S$ and $S'$, respectively. Then for every $z, z' \in \mathcal{Z}$ and every $t_0 \in [n]$ we have*

$$\mathbb{E}\big[|f(\mathbf{w}_T; z, z') - f(\mathbf{w}'_T; z, z')|\big] \leq \frac{2Bt_0}{n}\sup_{\mathbf{w};z,z'}f(\mathbf{w}; z, z') + G\mathbb{E}\big[\|\mathbf{w}_T - \mathbf{w}'_T\|\|1 \notin I_{t_0}(A)\big]\Pr\{1 \notin I_{t_0}(A)\},$$

*where $I_t(A) := \{i_1, j_1, \ldots, i_t, j_t\}$ is the set of indices selected by $A$ in the first $t$ iterations.*

*Proof.* According to the law of total expectation, we know

$$\mathbb{E}\big[|f(\mathbf{w}_T; z, z') - f(\mathbf{w}'_T; z, z')|\big] = \mathbb{E}\big[|f(\mathbf{w}_T; z, z') - f(\mathbf{w}'_T; z, z')|\|1 \in I_{t_0}(A)\big]\Pr\{1 \in I_{t_0}(A)\}$$

$$+ \mathbb{E}\big[|f(\mathbf{w}_T; z, z') - f(\mathbf{w}'_T; z, z')|\|1 \notin I_{t_0}(A)\big]\Pr\{1 \notin I_{t_0}(A)\}.$$

According to the update rule, we know

$$\Pr\{1 \in I_{t_0}(A)\} \leq \sum_{t=1}^{t_0}\Pr\{i_t = 1 \text{ or } j_t = 1\} = \sum_{t=1}^{t_0}\frac{2(n-1)}{n(n-1)} = \frac{2t_0}{n}.$$

The stated bound then follows from the Lipschitz continuity of $f$. The proof is complete. $\square$

We follow the arguments in [6] to prove Theorem 15.

*Proof of Theorem 15.* We first give the stability bounds. Suppose $S$ and $S'$ differ by the first example. If $i_t \neq 1$ and $j_t \neq 1$, then

$$
\begin{aligned}
\|\mathbf{w}_{t+1} - \mathbf{w}'_{t+1}\|_2 &= \big\|\mathbf{w}_t - \eta_t \nabla f(\mathbf{w}_t; z_{i_t}, z_{j_t}) - \mathbf{w}'_t + \eta_t \nabla f(\mathbf{w}'_t; z'_{i_t}, z'_{j_t})\big\|_2 \\
&\leq \big\|\mathbf{w}_t - \mathbf{w}'_t\big\|_2 + \big\|\eta_t \nabla f(\mathbf{w}_t; z_{i_t}, z_{j_t}) - \eta_t \nabla f(\mathbf{w}'_t; z_{i_t}, z_{j_t})\big\|_2 \\
&\leq \big(1 + L\eta_t\big)\big\|\mathbf{w}_t - \mathbf{w}'_t\big\|_2.
\end{aligned}
$$

Otherwise, we have

$$
\|\mathbf{w}_{t+1} - \mathbf{w}'_{t+1}\|_2 \leq \|\mathbf{w}_t - \mathbf{w}'_t\|_2 + 2G\eta_t.
$$

It then follows that

$$
\begin{aligned}
\mathbb{E}_{(i_t, j_t)}\big[\|\mathbf{w}_{t+1} - \mathbf{w}'_{t+1}\|_2\big] \\
\leq \big(1 + L\eta_t\big)\big\|\mathbf{w}_t - \mathbf{w}'_t\big\|_2 \Pr\{i_t \neq 1 \text{ and } j_t \neq 1\} + \big(\|\mathbf{w}_t - \mathbf{w}'_t\|_2 + 2G\eta_t\big)\Pr\{i_t = 1 \text{ or } j_t = 1\} \\
= \frac{(n-2)\big(1 + L\eta_t\big)}{n}\big\|\mathbf{w}_t - \mathbf{w}'_t\big\|_2 + \frac{2}{n}\big(\|\mathbf{w}_t - \mathbf{w}'_t\|_2 + 2G\eta_t\big). \tag{J.1}
\end{aligned}
$$

Let $\triangle_t = \mathbb{E}[\|\mathbf{w}_t - \mathbf{w}'_t\|\, |1 \notin I_{t_0}(A)]$, where $I_{t_0}(A)$ is defined in Lemma J.1. Then it follows from (J.1) that

$$
\begin{aligned}
\triangle_{t+1} &\leq \frac{(n-2)\big(1 + L\eta_t\big)}{n}\triangle_t + \frac{2}{n}\big(\triangle_t + 2G\eta_t\big) \leq \big(1 + L(1 - 2/n)\eta_t\big)\triangle_t + \frac{4G\eta_t}{n} \\
&\leq \exp\big(L(1 - 2/n)\eta_t\big)\triangle_t + \frac{4G\eta_t}{n}.
\end{aligned}
$$

Since $\triangle_{t_0+1} = 0$, we can apply the above inequality repeatedly and get

$$
\begin{aligned}
\triangle_T &\leq \sum_{t=t_0+1}^{T} \prod_{k=t+1}^{T} \exp\big(L(1 - 2/n)\eta_k\big)\frac{4G\eta_t}{n} \leq \sum_{t=t_0+1}^{T} \prod_{k=t+1}^{T} \exp\big(Lc(1 - 2/n)/k\big)\frac{4Gc}{nt} \\
&\leq \sum_{t=t_0+1}^{T} \exp\Big(Lc(1 - 2/n)\sum_{k=t+1}^{T}\frac{1}{k}\Big)\frac{4Gc}{nt} \leq \sum_{t=t_0+1}^{T} \exp\Big(Lc(1 - 2/n)\log(T/t)\Big)\frac{4Gc}{nt} \\
&\leq \sum_{t=t_0+1}^{T} \Big(\frac{T}{t}\Big)^{Lc(1-2/n)}\frac{4Gc}{nt} = \frac{4Gc}{n}T^{Lc(1-2/n)}\sum_{t=t_0+1}^{T} t^{-Lc(1-2/n)-1} \\
&\leq \frac{4Gc}{n}T^{Lc(1-2/n)}\int_{t_0}^{T} x^{-Lc(1-2/n)-1}dx \leq \frac{1}{Lc(1-2/n)}\frac{4Gc}{n}\Big(\frac{T}{t_0}\Big)^{Lc(1-2/n)},
\end{aligned}
$$

where we have used

$$
\eta_t = \frac{2t+1}{2\beta(t+1)^2} \leq c/t, \quad c := 1/\beta.
$$

We can combine the above bound and Lemma J.1 together, and get

$$
\mathbb{E}\big[|f(\mathbf{w}_T; z, z') - f(\mathbf{w}'_T; z, z')|\big] = O\Big(\frac{t_0}{n} + \frac{G^2}{nL}\Big(\frac{T}{t_0}\Big)^{Lc}\Big).
$$

We can choose $t_0 \asymp T^{\frac{Lc}{Lc+1}}$ and get the following stability bounds

$$
\mathbb{E}\big[|f(\mathbf{w}_T; z, z') - f(\mathbf{w}'_T; z, z')|\big] = O\Big(\frac{T^{\frac{Lc}{Lc+1}}}{n}\Big).
$$

We can plug the above stability bounds into Part (a) of Theorem 1, and get the following generalization bounds

$$
\mathbb{E}\big[F(\mathbf{w}_T) - F_S(\mathbf{w}_T)\big] = O\Big(\frac{T^{\frac{L/\beta}{L/\beta+1}}}{n}\Big).
$$

Furthermore, according to (D.8) we have the following optimization error bounds

$$
\mathbb{E}_A[F_S(\mathbf{w}_T)] - \inf_{\mathbf{w}}[F_S(\mathbf{w})] = O\big(1/(T\beta^2)\big).
$$

We can plug the above generalization and optimization error bounds into (3.1), and get (5.7). The proof is complete. $\square$

## K    Examples of Pairwise Learning

In this section, we give some specific examples of pairwise learning: metric learning, ranking and AUC maximization. We denote $(t)_+ := \max(t, 0)$ and $x^\top$ the transpose of $x \in \mathbb{R}^d$. Let $\text{sign}(t)$ denote the sign of $t \in \mathbb{R}$.

**Supervised metric learning**. In supervised metric learning, we assume $\mathcal{Y} = \{\pm 1\}$ and aim to find a distance metric such that examples in the same class are similar while examples in different classes are apart from each other under this metric. A typical choice is the Mahalanobis metric of the form $h_{\mathbf{w}}(x_i, x_j) = \langle \mathbf{w}, (x_i - x_j)(x_i - x_j)^\top \rangle, \mathbf{w} \in \mathbb{S}^{d \times d}$, where $\mathbb{S}^{d \times d}$ denotes the set of positive semi-definite matrices in $\mathbb{R}^{d \times d}$. A common loss function in metric learning for $\mathbf{w}$ on $z = (x, y), z' = (x', y')$ takes the form [8]

$$f(\mathbf{w}; z, z') = g(yy'(1 - h_{\mathbf{w}}(x, x'))),$$

where $g : \mathbb{R} \to \mathbb{R}_+$ is a convex function for which some typical choices include the hinge loss $g(t) = (1 - t)_+$ and the logistic loss $g(t) = \log(1 + \exp(-t))$.

**Ranking**. For ranking problems, the output reflects the ordering between instances, i.e., the instance $x$ is considered to be better than $x'$ if $y > y'$. Our task is to predict the ordering between the objects based on observations by constructing ranking rules $h_{\mathbf{w}} : \mathcal{X} \times \mathcal{X} \to \mathbb{R}$, and predict $y > y'$ if $h_{\mathbf{w}}(x, x') > 0$ [3]. A common pairwise loss function used in ranking problems takes the form

$$f(\mathbf{w}; z, z') = g(\text{sign}(y - y')h_{\mathbf{w}}(x, x')),$$

where $g : \mathbb{R} \to \mathbb{R}_+$ is a convex function for which some typical choices are the exponential loss $g(t) = \exp(-t)$, the logistic loss $g(t) = \log(1 + \exp(-t))$ and the hinge loss $g(t) = (1 - t)_+$ [3].

**AUC maximization**. AUC is a widely used metric for measuring the performance of machine learning algorithms in imbalanced classification. If $\mathcal{Y} = \{\pm 1\}$, the AUC score of a model $h_{\mathbf{w}} : \mathcal{X} \mapsto \mathcal{Y}$ measures its probability of giving a larger value to a positive instance than to a negative instance. The problem of AUC maximization can be formulated as a pairwise learning problem with the following loss function [4, 22]

$$f(\mathbf{w}; z, z') = g(\mathbf{w}^\top (x - x'))\mathbb{I}_{[y=1, y'=-1]}, \tag{K.1}$$

where $g : \mathbb{R} \to \mathbb{R}_+$ is a convex function for which some typical choices are the least square loss $g(t) = (1 - t)^2$, the logistic loss $g(t) = \log(1 + \exp(-t))$ and the hinge loss $g(t) = (1 - t)_+$.

## L    Experimental Results

In this section, we present some experimental results to support our theory on the stability bounds. We consider AUC maximization with the loss function of the form of (K.1). We consider several datasets available at the LIBSVM site [2], whose information is summarized in Table L.1. We transform datasets with multiple class labels into datasets with binary class labels by grouping the first half of class labels into positive labels, and grouping the remaining class labels into negative labels. We randomly choose 80 percents of each dataset as the training set $S$, from which we perturb a single example in $S$ to create a neighboring dataset $S'$. We apply SGD (3.2) with the same parameters to $S$ and $S'$, and get two sequence of iterates $\{\mathbf{w}_t\}$ and $\{\mathbf{w}_t'\}$. We then calculate the Euclidean distance $\triangle_t = \|\mathbf{w}_t - \mathbf{w}_t'\|_2$ at each iteration to verify the stability of SGD. We consider step sizes of the form $\eta_t = \eta/\sqrt{T}$ with $\eta \in \{0.05, 0.25, 1, 4\}$, and report $\triangle_t$ as a function of the number of passes (the iteration number $t$ divided by the sample size $n$). We repeat the experiments 100 times and report the average as well as the standard deviation. Since we develop stability bounds for both smooth and nonsmooth loss functions, we consider two representative loss functions: the smooth logistic loss (i.e., Eq. (K.1) with $g(t) = \log(1 + \exp(-t))$) and the nonsmooth hinge loss (i.e., Eq. (K.1) with $g(t) = (1 - t)_+$).

In Figure L.1, we report the Euclidean distance $\triangle_t$ for AUC maximization with the hinge loss and the 4 stepsize sequences, while in Figure L.2, we report $\triangle_t$ for AUC maximization with the logistic loss. It is clear that $\triangle_t$ is an increasing function of both $t$ and $\eta$, which is consistent with our stability bounds in Theorem 3 and Theorem 6. It is also clear that the Euclidean distances for the logistic loss are significantly smaller than those for the hinge loss if we consider the same stepsize sequence, which is also consistent with Remark 4 on the comparison of stability bounds for SGD with smooth and nonsmooth problems.

Table L.1: Description of the datasets used in the experiments.

| datasets | # inst | # feat | datasets | # inst | # feat | datasets | # inst | # feat | datasets | # inst | # feat |
|----------|--------|--------|----------|--------|--------|----------|--------|--------|----------|--------|--------|
| a3a | 3185 | 122 | acoustic | 78823 | 50 | cifar10 | 50000 | 3072 | gisette | 7000 | 5000 |
| madelon | 2600 | 500 | mnist | 60000 | 780 | usps | 7291 | 256 | webspam_u | 350000 | 254 |

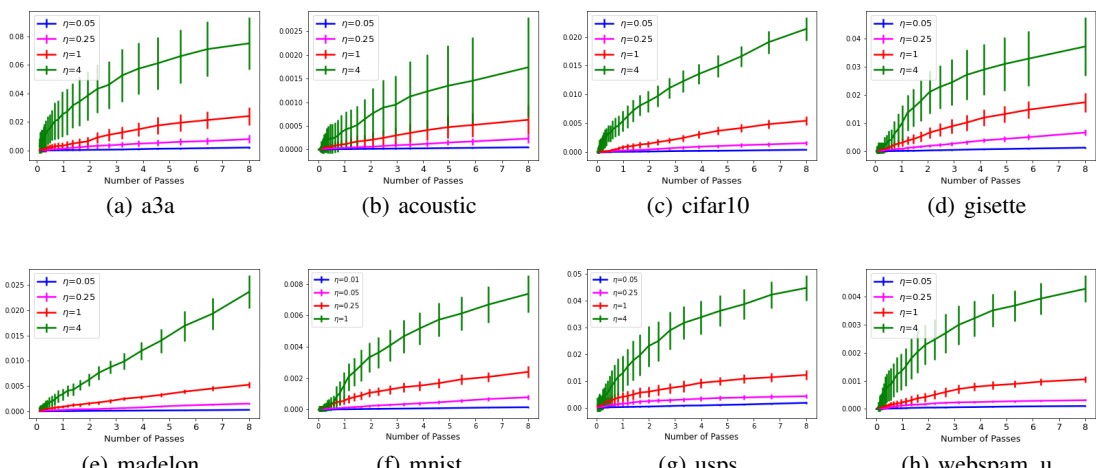

Figure L.1: Euclidean distance $\triangle_t$ as a function of the number of passes for the hinge loss.

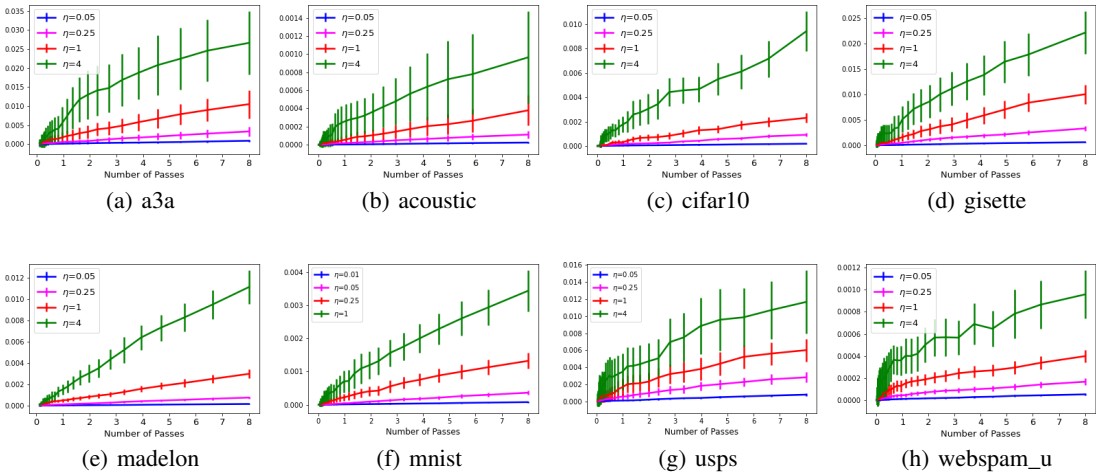

Figure L.2: Euclidean distance $\triangle_t$ as a function of the number of passes for the logistic loss.