# OpenReview forum: "Generalization Guarantee of SGD for Pairwise Learning"
_NeurIPS.cc/2021/Conference — NeurIPS 2021 Poster_

### Official Review · Reviewer_mXiM · 2021-07-07

**Rating:** 5
**Confidence:** 3

**Summary:**

The paper proposes generalization for pairwise learning, based on assumptions of algorithmic stability.
It shows different bounds that depend on assumptions on the smoothness of the pairwise function.

**Limitations And Societal Impact:**

Nothing to declare

**Main Review:**

I did not delve into the math in detail.
I think the most interesting part of the publication is how they can propose stronger assumptions to derive bounds in $O(n^{-1})$ for the SGD algorithm, without any noise assumption on the data distribution (as do generalization results in $O(n^{-1})$ for U-statistics do, see [13]), but only using assumptions on the pairwise function $f$. You say you incorporate the variance in the analysis to derive faster bounds, which is exactly the purpose of the Mammen-Tsybakov (sometimes known as Bernstein) conditions for instance based (not pairwise) learning use in generalization bounds. [13] shows that we can bound the variance at a lower cost when dealing with U-statistics, as the analysis relies on the second Hoeffding decomposition (writing a U-statistic as an standard empirical process and a degenerate U-statistic) and the Hajek projection used to obtain the standard empirical process is a variance-reducing transformation of the original U-statistic. Your results propose fast bounds that are derived from the shape of the function $f$ rather than by an assumption the data, which is novel in my opinion (I looked at [37] and they seem to only propose $O(n^{-1/2})$ and in my opinion is a good result to extend with your SGD analysis.
I think the paper seeks to cover too much ground for the format of the conference, which obscures the message, and should recenter the paper on one meaningful message.

Also, ref [a1] below is proposing guarantees for SGD for the minimization of U-statistics (pairwise functions), which is exactly what you are doing. Their analysis assumes only convexity (l. 41-42) and a VC-assumption. I believe that your contributions should be compared to theirs.

[a1] SGD Algorithms based on Incomplete U -statistics: Large-Scale Minimization of Empirical Risk, Papa et al, NeurIPS 2015

----- AFTER READING THE REVIEWS AND REBUTTAL

I thank the authors for a serious response to the concerns of the reviewers.
I am very familiar with uniform convergence generalization bounds but not that familiar with algorithmic stability.
While my reject rating was motivated by the fact that the paper overlaps with existing work, the authors have successfully underlined the differences between their work and [a1]. For these reasons, I raised my rating to weak reject, as I would accept the paper if its results were put in perspective with already existing pairwise generalization bounds. I did not raise it further as integrating them is a significant undertaking.

**Time Spent Reviewing:**

2

---

> ### Author Response · Authors · 2021-08-08
> **Response to Reviewer mXiM**
>
> First of all, thank you for your constructive suggestions and comments.
>
> **Q: Discussion with related work [13, a1]**
>
> **A:** Thank you for indicating the interesting work on the generalization analysis of SGD [a1] which we are not aware of. We will cite this paper and add detailed discussions in the revised version. Here we sketch the difference between our work and [13, a1].
>
> The work [13] focuses on the exact solution of empirical risk minimizer for pairwise learning, where the optimization error is not considered there. As a comparison, we study the excess risk of SGD and we consider the tradeoff between optimization and generalization. Furthermore, [13] studies the generalization gap from the perspective of uniform convergence, i.e., bounding the uniform deviation between population and empirical risks over a hypothesis space. As a comparison, we consider two approaches: an algorithmic stability approach for convex problems and an uniform convergence approach for nonconvex problems. These two approaches are orthogonal and each one cannot dominate the other. For example, the uniform convergence has the superiority of being able to handle nonconvex problems, while the convexity assumption is often required for algorithmic stability to get meaningful bounds [25]. On the other hand, the uniform convergence requires a complexity assumption on the hypothesis space. As a comparison, algorithmic stability always yields complexity-free bounds and is appealing for learning with large hypothesis spaces [5]. Therefore, our stability analysis and the uniform convergence in [13] are complementary to each other.
>
> The work [a1] considers a very general problem setting for SGD with $K$-sample U-statistic of degrees $(d_1,\ldots,d_K)$, which includes our algorithm as a special case with $K=1$ and $d_1=2$. It shows the advantage of reducing variances using gradient estimates through incomplete U-statistics over that through complete U-statistics based on subsamples. We sketch the difference as follows. First, the generalization analysis in [a1] requires smoothness and strong convexity assumptions. As a comparison, we also consider nonsmooth problems (Section 4.2) and nonconvex problems (Section 5). Second, the work [a1] studies generalization via the uniform convergence approach and requires a complexity assumption. As a comparison, we also study generalization via a fundamentally different algorithmic stability approach.  It would be very interesting to see whether our stability analysis can be extended to SGD with general $K$-sample U-statistic in [a1]. We will leave it as a research problem for further investigation.
>
> **Q: I think the paper seeks to cover too much ground for the format of the conference, which obscures the message, and should recenter the paper on one meaningful message.**
>
> **A:** Thank you for the constructive comment. Our main message is to significantly improve the existing stability analysis by developing fast and high-probability rates for SGD with a cheap computational complexity. For example, the state-of-the-art stability analysis in [37] can only yield O(1/sqrt{n}) bounds and require a smoothness assumption. As a comparison, our stability analysis can yield O(1/n) bounds and apply to nonsmooth problems. We will clarify this in the revised version.

---

### Official Review · Reviewer_xcch · 2021-07-15

**Rating:** 6
**Confidence:** 3

**Summary:**

The paper seeks to develop new generalization and excess risk bounds for pairwise learning using SGD. The goal is to achieve error rates with near-optimal dependence in the number of gradient evaluations. To this end, stability properties of SGD are analysed and used to obtain the generalization/excess risk bounds.

**Limitations And Societal Impact:**

No negative social impact

**Main Review:**

When it comes to deriving generalization and excess-risk bounds, the authors argue that pairwise learning is more difficult than point-wise learning due to the non-i.i.d. nature of the training examples of the former. This then motivates the approach taken by the authors, which involves studying the stability properties of SGD and applying some decoupling tricks (to overcome issues that come with the non-i.i.d. nature of the problem).

It seems to me that the authors have missed some highly relevant work in point-wise learning that should lead to optimal excess-risk rates (both in expectation and high probability) when applied to pair-wise learning. In fact, modern online-to-batch-conversion techniques---see e.g. Theorem 1 in http://proceedings.mlr.press/v97/cutkosky19a.html---reduce the problem of deriving excess risk bounds to that of getting regret bounds in an online learning setting. The data need not be i.i.d., and so this is applicable to pair-wise learning. Furthermore, SGD is essentially just OGD in the online learning setting, for which regret bounds are well studied for non-smooth/smooth and convex/strongly convex cases. Therefore, applying Theorem 1 in http://proceedings.mlr.press/v97/cutkosky19a.htm to the setting of pair-wise learning, and plugging in the regret bound of OGD, should recover, or even improve the excess-risk result of the current paper.

The current paper might still have some novel contributions when it comes to generalization bound (not excess-risk bounds) since I do not really know how these can be obtained using the online learning framework.

The paper is a little dense and still contains many inconsistencies and typos. I now mention some of them:
- It seems that the authors talk about generalization bounds when they mean excess-risk bound (see e.g. Theorem 7). Please fix this.
- The use of 'linear time' is not correct. A linear time rate in n is something line exp(-O(n)).
- Lines 102-104, you mention a randomized algorithm A, but you are not taking the expectation when you write F(A(S))
- In thm1, it should be F(A(S)) and not F(S)
- In thm3, mention that A is given a training dataset of size n
- The statement of Thm 4 is a little strange. How do you define a w that is independent of the algorithm A yet dependent on S? Furthermore, to call the result of Thm 4 a generalization bound, you technically need w = A(S). But the current theorem statement does not allow this.
- Another instance of confusing the concepts of generalization and excess-risk is in Remark 3: you start talking about generalization, then switch to talking about F(w_\star)
- In lines 246-247: the claim that a log n^2/\sqrt{n} bound has not been derived before for point-wise learning needs more explanation. Can you point to the best-known rate for this case?
- In Thm 9, what is w_s^*? Define it explicitly.
- Missing bracket in F_S(A(S) in thm 10.


**Time Spent Reviewing:**

3 hours

---

> ### Author Response · Authors · 2021-08-08
> **Response to Reviewer xcch**
>
> First of all, thank you for your constructive comments and suggestions.
>
> **Q: Applying Theorem 1 in Cutkosky (2019) should recover or even improve the excess-risk result of the current paper.**
>
> **A:** Thank you for pointing out the appealing work on modern online-to-batch-conversion techniques which we will cite and discuss in detail in the revised version.
>
> We would like to clarify that we consider a fundamentally different problem setting from the one in Cutkosky (2019), and our excess risk bounds cannot be recovered from an application of Theorem 1 in Cutkosky (2019). Indeed, the excess risk bounds in Cutkosky (2019) were derived under the assumption that $g_t$ is an unbiased estimator of the gradient of the population risk. This is different from our problem setting where we first draw a dataset $S$ from the true probability measure, and then apply SGD to minimize $F_S$. For our SGD, $\nabla f(w_t;z_{i_t},z_{j_t})$ is an unbiased gradient estimator of the empirical risk $F_S$ instead of the population risk $F$. Therefore, directly applying Theorem 1 in Cutkosky (2019) to our setting yields actually optimization error bounds $F_S(w_t)-\inf_{w}F_S(w)$. To bound the excess population risk $F(w_t)-\inf_{w}F(w)$ in our setting, as we detailed in Section 3.1, we need to consider another term called the generalization gap $F(w_t)-F_S(w_t)$. This term appears due to the discrepancy between the empirical probability measure associated with SGD and the true probability measure for sampling $S$, and is challenging to control since $w_t$ depends on $S$ in a highly nonlinear manner. Our main focus is to study the generalization gap by leveraging algorithmic stability, which is a fundamental concept in statistical learning theory. Furthermore, the analysis in Cutkosky (2019) requires a convexity assumption, while we also consider nonconvex problems in Section 5. In summary, our results cannot be directly recovered from the analysis in Cutkosky (2019) as we need to handle the challenging problem of controlling the generalization gap, which is not discussed in Cutkosky (2019). We will clarify the similarity and difference between Cutkosky (2019) and ours in the revised version.
>
> **Q:  The paper is a little dense and still contains many inconsistencies and typos.**
>
> **A:** Thank you for your careful reading and kindly pointing out the typos. We will address them in the revised version and would particularly pay attention to the difference between generalization bounds and excess risk bounds.
>
> **Q:** The statement of Thm 4 is a little strange. How do you define a $w$ that is independent of the algorithm A yet dependent on S? Furthermore, to call the result of Thm 4 a generalization bound, you technically need $w = A(S)$. But the current theorem statement does not allow this.
>
> **A:** Sorry for the confusion. Theorem 4 should be an excess risk bound instead of a generalization bound. We will not apply Theorem 4 with $w=A(S)$. Actually, we apply Theorem 4 with $w=w_S^*$ ($w_S^*$ is defined in line 102) and $w=w^*$ to get excess risk bounds (please see our discussions between line 195 to 202). Please also note $w_S^*$ depends on $S$ but not on $A$ as it is the empirical risk minimizer. We will change the title of Theorem 4 from generalization bound to excess risk bound in the revised version to avoid confusion.
>
> **Q:** In lines 246-247: the claim that a  $\log n^2/\sqrt{n}$ bound has not been derived before for point-wise learning needs more explanation. Can you point to the best-known rate for this case?
>
> **A:** Thank you for the careful reading. The best high-probability excess risk bound for SGD with nonsmooth loss functions is $\log n^2/\sqrt{n}$ and was derived in [3]. However, their algorithm requires $O(n^2)$ gradient computations, which are expensive for large-scale applications. We significantly improve it by deriving the same excess risk bound with the much less $O(n)$ gradient computations. We will make the statement of the claim more clearly in the revised version.
>
> **Q:** In Thm 9, what is $w_s^*$? Define it explicitly.
>
> **A:** It is defined in Line 102. We will add a table of notations in the revised version to make the notations clear.

---

> > ### Comment · Reviewer_xcch · 2021-08-25
> > **Further clarification**
> >
> > Thanks for your initial answers to my questions.
> >
> > If you let $G_{t-1}$ be the $\sigma$-algebra generated by the past randomness up to round $s=t-1$, isn't it the case that $\mathbb{E}[\nabla f(w_t;z_{I_t}, z_{J_t})  \mid  G_{t-1}] = \nabla F(w_t)$, where $F$ is the population risk? Note that $w_t$ is $G_{t-1}$-measurable, i.e. it is a function of the past. If you agree that the above equality holds, then the result of Cutkosky (2019) should be enough to get a bound on $F(w_t) -\min_w F(w)$ (i.e. no need for a generalization bound).

---

> > > ### Author Response · Authors · 2021-08-25
> > > **Response**
> > >
> > > Thank you for your further query. We consider an offline setting where the dataset $S$ is drawn before running the algorithm and each training example can be used several times in the optimization process. In this setting, the inequality $\mathbb{E}[\nabla f(\mathbf{w}_t;z_\{i_t\},z_\{j_t\})|G_\{t-1}]=\nabla F(\mathbf{w}_t)$ no longer holds true. The underlying reason is that $\mathbf{w}_\{t\}$ can depend on $(z_\{i_t\},z_\{j_t\})$ because either $z_\{i_t\}$ or $z_\{j_t\}$ may have already been encountered in the previous iterations (for example, it is possible that $i_t=i_1$ and in this case $z_\{i_t\}$ has already been used in getting $\mathbf{w}_t$). This dependency does not allow us to take the conditional expectation w.r.t. $(z_\{i_t\},z_\{j_t\})$ to get $\nabla F(\mathbf{w}_t)$. In our setting, at each iteration we draw an index pair $(i_t,j_t)$ which is independent of $\mathbf{w}_t$. Therefore, we can only take the conditional expectation w.r.t. $(i_t,j_t)$ and get $\mathbb{E}_\{i_t,j_t\}[f(\mathbf{w}_t;z_\{i_t\},z_\{j_t\})]=F_S(\mathbf{w}_t)$. Then, we have to use either the stability analysis or the uniform convergence analysis to control the generalization gap between $F_S(\mathbf{w}_t)$ and $F(\mathbf{w}_t)$, which is main aim of this paper. Please note that the online setting considered in the paper (Cutkosky, 2019) assumes an unbiased gradient estimate of $\nabla F(\mathbf{w}_t)$ and therefore there is no need to study the generalization gap there. We do not have this unbiased gradient estimate of $\nabla F(\mathbf{w}_t)$ in our offline setting.

---

> > > > ### Comment · Reviewer_xcch · 2021-08-25
> > > > **Thanks for clarifying**
> > > >
> > > > Thanks for the clarification. I am satisfied with your answer and will increase my score accordingly.

---

### Official Review · Reviewer_NxFE · 2021-07-17

**Rating:** 7
**Confidence:** 4

**Summary:**

The paper analyzes the generalization ability of SGD for pairwise learning problems. The authors prove better generalization error bounds incorporating some variance term or the optimal error. They also provide a new SGD with O(n) computation per trial whose generalization bound is state-of-the-art. Furthermore, they show results for strongly convex losses and non-convex losses.


**Main Review:**

The writing of the paper is excellent in general. In particular, every theorem comes with a remark, which is helpful to understand the technical improvement to related work.
The paper is technically solid.

My concern is about concrete examples of loss functions and motivating problems. For example, AUC maximization is closely related to the paper. However, generalization error of AUC would involve conditional distributions on positive and negative instances and the generalization error defined in this paper seems not appropriate to the AUC problem. Then I wonder what concrete problems are subjects of the formulation of the paper.

Corollary 5 (b) assumes that F(w^*) is O(1/n), but this assumption seems not natural since F(w^*) is independent of the sample size and does not decrease when sample size n increases. So I feel that statement (b) is not meaningful. Furthermore, in Remark 3, the authors insist on the advantage of Theorem 4 to previous work [37][53] based on the corollary. I feel the statement is not fair as it is based on a weird assumption and should be removed.

The technical results of Theorem 8 (improving n^2 to n in computation time per trial) and Corollary 13 (extension of the previous approach to pairwise non-convex learning) are non-trivial and impressive. In particular, Coro. 13 needs to overcome the difficulty caused by dependent pairs in the analysis.

Overall, my impression of the paper is rather positive and the paper is better evaluated if my concerns raised above are resolved.

Minor Comments:
Line 145: Does the expectation is also taken w.r.t. the randomized algorithm A?


After reading other reviews and author responses, I raised my score a bit since the raised issues are resolved.


**Time Spent Reviewing:**

3 hours

---

> ### Author Response · Authors · 2021-08-08
> **Response to Reviewer NxFE**
>
> First of all, thank you for your constructive comments and suggestions.
>
> **Q: I wonder what concrete problems are subjects of the formulation of the paper.**
>
> **A:** Thank you for the constructive comment. Two notable examples of pairwise learning include ranking and supervised metric learning. For ranking, we build a function $h_w:\mathcal{X}\to \mathcal{Y}$ to rank instances in a way consistent with the outputs, i.e., $h_w(x)<h_w(x')$ if $y<y'$ for two example pairs $z=(x,y),z'=(x',y')$ [13]. Then we can formulate ranking as a pairwise learning problem with $f(w;z,z')=\psi(sgn(y-y')(h_{w}(x)-h_w(x')))$, where $sgn$ is the sign function and $\psi$ can be either the hinge loss $\psi(t)=\max\\{1-t,0\\}$ or the logistic loss $\psi(t)=\log(1+\exp(-t))$. For supervised metric learning with $\mathcal{Y}=\\{-1,+1\\}$, we find a distance function under which examples with the same label are similar while examples with different labels are apart from each other [29]. A popular distance function takes the form $h_w(x,x')=\langle w,(x-x')(x-x')^\top\rangle$, where $w\in \mathbb{R}^{d\times d}$ is positive definite. We can formulate supervised metric learning as pairwise learning with $f(w;z,z')=\psi(\tau(y,y')(1-h_w(x,x')))$, where $\tau(y,y')=1$ if $y=y'$ and $-1$ otherwise.
>
> Since ranking and supervised metric learning are instanations of pairwise learning problems, our generalization analysis directly applies and yields meaningful excess risk bounds. We will add more applications of our results to specific problems in the revised version.
>
> **Q: Corollary 5 (b) and Remark 3**
>
> **A:** Thank you for the careful observation and constructive suggestions. The main message we want to convey regarding Theorem 4 is that we can take $w=w^*$ there and get an upper bound involving $F(w^*)$, which improves if $F(w^*)$ is small. In the literature, people refer to bounds of this property as optimistic bounds [54], where the case $F(w^*)=0$ was considered to derive fast excess risk bounds [54]. We follow the discussion in [54] by showing our analysis can derive fast rates by exploiting the property of small $F(w^*)$ in the low-noise case, while the existing stability analysis [37, 53] cannot yield fast rates in the low-noise case. We agree assuming $F(w^*)=O(1/n)$ is a bit confusing since the left-hand side is independent of sample size. Following your suggestion, we will modify the statement of Corollary 5 and Remark 3 in the revised version, and add more discussions regarding optimistic bounds.
>
> **Q:  Line 145: Does the expectation is also taken w.r.t. the randomized algorithm A?**
>
> **A:** Thank you for the careful reading. You are right. We will modify it in the revised version.

---

### Author Response · Authors · 2021-08-29
**Thank you for the update**

We thank all the reviewers for updating their reviews and scores. It is highly appreciated.

---

### Decision · Program_Chairs · 2021-09-27

**Decision:**

Accept (Poster)

**Comment:**

The paper makes a good contribution to pairwise learning problems by providing better generalization error bounds for stochastic gradient descent (SGD) applied to these problems, and providing analysis for both convex and non-convex problems.